# Solving Stochastic Variational Inequalities
# without the Bounded Variance Assumption

**Ahmet Alacaoglu** [1]   **Jun-Hyun Kim** [1]

## Abstract

We analyze algorithms for solving stochastic variational inequalities (VI) without the bounded variance or bounded domain assumptions, where our main focus is min-max optimization with possibly unbounded constraint sets. We focus on two classes of problems: monotone VIs; and structured nonmonotone VIs that admit a solution to the *weak Minty VI*. The latter assumption allows us to solve structured nonconvex-nonconcave min-max problems. For both classes of VIs, to make the expected residual norm less than $\varepsilon$, we show an oracle complexity of $\widetilde{O}(\varepsilon^{-4})$, which is the best-known for constrained VIs. In our setting, this complexity had been obtained with the bounded variance assumption in the literature, which is not even satisfied for bilinear min-max problems with an unbounded domain. We obtain this complexity for stochastic oracles whose variance can grow as fast as the squared norm of the optimization variable.

## 1. Introduction

In this work, we focus on stochastic variational inequalities (SVI) where the aim is to

$$\text{find } \mathbf{z}^\star \text{ s.t. } \langle G(\mathbf{z}^\star), \mathbf{z} - \mathbf{z}^\star \rangle + r(\mathbf{z}) - r(\mathbf{z}^\star) \geq 0 \quad \forall \mathbf{z}, \quad (1.1)$$

where $r \colon \mathbb{R}^m \to \mathbb{R} \cup \{+\infty\}$ is a proper, convex and closed function and $G \colon \mathbb{R}^m \to \mathbb{R}^m$ is an operator. When $r$ is equal to the indicator function of a convex and closed set $C \subseteq \mathbb{R}^m$, (1.1) reduces to a more well-studied SVI problem with set constraints. In the stochastic case, we assume that we have an *unbiased* oracle $\widetilde{G}$ such that

$$\mathbb{E}[\widetilde{G}(\mathbf{z})] = G(\mathbf{z}). \quad (1.2)$$

[1]University of British Columbia, Vancouver, Canada. Correspondence to: Ahmet Alacaoglu <ahmet.alacaoglu@ubc.ca>, Jun-Hyun Kim <junhyun@student.ubc.ca>.

*Proceedings of the 43$^{rd}$ International Conference on Machine Learning*, Seoul, South Korea. PMLR 306, 2026. Copyright 2026 by the author(s).

One common application of SVI is stochastic min-max optimization, formulated as,

$$\min_{\mathbf{x} \in \mathbb{R}^d} \max_{\mathbf{y} \in \mathbb{R}^n} f(\mathbf{x}, \mathbf{y}) + h_1(\mathbf{x}) - h_2(\mathbf{y}), \quad (1.3)$$

where $h_1, h_2$ are given *regularizers*. Taking $h_1, h_2$ as indicator functions results in constrained min-max optimization. This problem maps to (1.1) with $\mathbf{z} = \begin{pmatrix} \mathbf{x} \\ \mathbf{y} \end{pmatrix}$,

$$G(\mathbf{z}) = \begin{pmatrix} \nabla_{\mathbf{x}} f(\mathbf{x}, \mathbf{y}) \\ -\nabla_{\mathbf{y}} f(\mathbf{x}, \mathbf{y}) \end{pmatrix} \text{ and } r(\mathbf{z}) = h_1(\mathbf{x}) + h_2(\mathbf{y}). \quad (1.4)$$

The problem (1.3) gained significant interest in machine learning recently due to applications in adversarial and robust learning, as well as generative adversarial networks (Duchi & Namkoong, 2021; Madry et al., 2018; Goodfellow et al., 2014). Moreover, (1.3) is a classical framework for solving constrained optimization problems (Bertsekas, 2014). For example, given a nonlinear programming problem

$$\min_{\mathbf{x} \in X} p(\mathbf{x}) \text{ subject to } q(\mathbf{x}) \leq 0, \quad (1.5)$$

the standard approach is to use the Lagrangian duality framework to reformulate this problem as

$$\min_{\mathbf{x} \in X} \max_{\mathbf{y} \geq 0} p(\mathbf{x}) + \langle \mathbf{y}, q(\mathbf{x}) \rangle. \quad (1.6)$$

It is now easy to see that this problem corresponds to (1.3), which itself is a special case of (1.1). In addition to min-max optimization, SVI is also a common framework to model problems arising in game theory, see for example (Mertikopoulos, 2019, Section 2.3).

**Variance assumptions.** In our stochastic setup, a common assumption in addition to (1.2) is that of bounded variance (Nemirovski et al., 2009; Robbins & Monro, 1951). In particular, $\widetilde{G}$ satisfies this assumption if the following holds for a finite $\sigma^2$:

$$\mathbb{E}\|\widetilde{G}(\mathbf{z}) - G(\mathbf{z})\|^2 \leq \sigma^2. \quad (\text{BV})$$

Despite being standard, it is well-known that this is a restrictive assumption. In particular, let us consider a simple bilinear min-max problem, in view of (1.3). In this case, the operator $G$ from (1.1) is linear (see (1.4)), and hence (BV)

fails with common sampling schemes to obtain stochastic oracles, such as row/column sampling, unless the domains of $h_1, h_2$ are bounded. Boundedness of the domains of $h_1, h_2$ is unfortunately unrealistic, which, for example, can be seen by considering (1.5) with linear $p, q$ (which indeed gives us the classical linear programming problem), where the resulting min-max problem (1.6) has unbounded domains. We also refer to (Iusem et al., 2017, Example 1) for a discussion about this assumption in our setting.

In this work, we focus on algorithms and convergence analyses for solving SVI under the weaker assumption

$$\mathbb{E}\|\widetilde{G}(\mathbf{z}) - G(\mathbf{z})\|^2 \leq B^2\|\mathbf{z} - \mathbf{z}^\star\|^2 + \sigma^2, \qquad (1.7)$$

and its *equivalent* variant[1]

$$\mathbb{E}\|\widetilde{G}(\mathbf{z}) - G(\mathbf{z})\|^2 \leq B^2\|\mathbf{z} - \mathbf{z}_0\|^2 + \sigma^2, \qquad (1.8)$$

which are used for example in (Gladyshev, 1965; Iusem et al., 2017; Choudhury et al., 2023; Alacaoglu et al., 2025; Kotsalis et al., 2022). It is simple to see that this assumption is trivially satisfied when $G$ is linear, which corresponds to bilinear objectives in (1.3). For standard row/column sampling in bilinear problems, the noise scales with the iterate, so bounded variance can fail on unbounded domains; See Example A.2 in App. A for a concrete example. To our knowledge, this is currently the most relaxed variance assumption under which one gets optimal convergence rate and complexity guarantees even for convex minimization, see for example (Alacaoglu et al., 2025; Neu & Okolo, 2024). In particular, this assumption implies the other relaxations of (BV), such as the ones appearing in (Khaled et al., 2023; Khaled & Richtárik, 2023) or (Mishchenko et al., 2020), see Fact A.3.

**Notation.** The subdifferential of the convex function $r$ in (1.1) is denoted as $\partial r$. With this, for notational convenience, we write (1.1) as an *inclusion* problem:

$$\text{find } \mathbf{z}^\star \text{ such that } 0 \in (G + \partial r)\mathbf{z}^\star. \qquad (1.9)$$

Classical references containing the background for this formulation include (Bauschke & Combettes, 2017; Rockafellar, 1976). Recall the definition of the proximal operator: $\text{prox}_r(\mathbf{x}) = \arg\min_{\mathbf{u}} r(\mathbf{u}) + \frac{1}{2}\|\mathbf{u} - \mathbf{x}\|^2$.

**Residual.** Despite the ubiquity of SVI and the well-known limitations of the bounded variance assumption (BV), analysis of algorithms for SVI under (1.7), especially for non-monotone problems, remained mostly unexplored. This is relevant especially in the case when we do not have bounded domains (since a bounded domain would reduce (1.7) to

(BV)). Unbounded domains are common in min-max optimization, as mentioned earlier. In this setting, we will focus on complexity guarantees on the *residual*, given as

$$\text{res}(\mathbf{z}_k) := \text{dist}(0, (G + \partial r)\mathbf{z}_k) = \min_{\mathbf{u} \in (G + \partial r)\mathbf{z}_k} \|\mathbf{u}\|, \tag{1.10}$$

which is the generalization of the gradient (or operator) norm optimality measure (which is for unconstrained problems). Indeed, one can observe that when $r \equiv 0$, this reduces to $\|G(\mathbf{z}_k)\|$. For the min-max case, this is nothing but $\sqrt{\|\nabla_{\mathbf{x}} f(\mathbf{x}_k, \mathbf{y}_k)\|^2 + \|\nabla_{\mathbf{y}} f(\mathbf{x}_k, \mathbf{y}_k)\|^2}$.

The reason for our focus on the residual is the following: When we have nonconvex-nonconcave problems, the duality gap cannot be used. Even with convex-concave problems, in the case of unbounded domains, the duality gap, which is commonly used for VIs (see (Nemirovski et al., 2009; Facchinei & Pang, 2003)), is not applicable, since it would require taking a maximum over the domain of $r$ which is unbounded.

A common workaround is the *so-called* restricted duality gap (Nesterov, 2007). The restricted gap is also not completely satisfactory because for the restricted gap to be an optimality measure, one needs the knowledge of the norm of the iterates that the algorithm generates (Nesterov, 2007, Lemma 1). This is neither known in advance, nor easily controllable in the stochastic case since stochastic algorithms generate sequences that are not uniformly bounded, in general.

**Role of convexity and monotonicity.** Let us consider the case of convex-concave min-max optimization, which is the problem we have when $f(\cdot, \mathbf{y})$ is convex and $f(\mathbf{x}, \cdot)$ is concave. This leads to a monotone SVI, that is, the operator $G$ defined by the gradients of $f$ (see (1.4)) is a monotone operator:

$$\langle G(\mathbf{x}) - G(\mathbf{y}), \mathbf{x} - \mathbf{y} \rangle \geq 0.$$

To go beyond the convex-concave case, we will consider a common assumption from the literature, which requires the existence of a solution $\mathbf{x}^\star$ to the ($\rho \geq 0$)-weak Minty Variational Inequality (MVI), that is,

$$\langle \mathbf{u}, \mathbf{x} - \mathbf{x}^\star \rangle \geq -\rho\|\mathbf{u}\|^2, \text{ where}$$
$$(\mathbf{x}, \mathbf{u}) \in \text{gra}(G + \partial r) = \{(\mathbf{x}, \mathbf{u}) : \mathbf{u} \in (G + \partial r)\mathbf{x}\}. \tag{wMVI}$$

When $r \equiv 0$, this assumption was proposed by (Diakonikolas et al., 2021) which generalized the comonotonicity assumption of (Combettes & Pennanen, 2004; Bauschke et al., 2021) (see also (Pethick et al., 2022) for the constrained version) which is shown to hold for the so-called *interaction dominant* min-max problems (Grimmer et al., 2023) and others (Pethick et al., 2022). For brevity, we sometimes refer to this as the *weak MVI assumption* in the sequel. This

---

[1]The equivalence is trivial to see by using Young's inequality on the right-hand side and adjusting the definitions of constants $B^2, \sigma^2$. See Appendix A (Fact A.1) for details.

defines a class of *nonmonotone* problems since monotonicity of $G$ is not required. The level of nonmonotonicity (or, nonconvex-nonconcavity for the special case of min-max problems in (1.3)) is set by the parameter $\rho \geq 0$.

When $\rho = 0$, this reduces to the well-known assumption of the existence of a solution to the Minty VI (Facchinei & Pang, 2003), which is also sometimes referred to as the *coherence*, see, e.g., (Mertikopoulos et al., 2019). For policy optimization in reinforcement learning, a variation of this assumption with $\rho = 0$ holds (Lan, 2023). Some applications, including those with $\rho > 0$ are included in (Pethick et al., 2022; Alacaoglu et al., 2024; Lee & Kim, 2021).

The assumption (wMVI) is among the weakest-known requirement under which complexity results have been shown for nonconvex-nonconcave problems. Some alternative assumptions require Polyak-Łojasiewicz (PŁ) or Kurdyka-Łojasiewicz (KŁ)-type properties to hold for the dual variable, see, e.g., (Li et al., 2025). The relationship between the latter class of problems and problems satisfying (wMVI) is not well-understood and the algorithm development between these two classes of problems have been largely independent of each other. We focus on analyzing algorithms under (wMVI).

### 1.1. Assumptions

For all our results except Section 4, we require a standard Lipschitzness assumption for the operator $G$, which is defined as

$$\|G(\mathbf{x}) - G(\mathbf{y})\| \leq L\|\mathbf{x} - \mathbf{y}\|.$$

We now collect our assumptions used in the sequel.

**Assumption 1.** Let $G$ be $L$-Lipschitz and let $G + \partial r$ satisfy (wMVI) with a solution $\mathbf{z}^\star$ and $\rho \geq 0$.

In Section 4, we will use the stronger *expected Lipschitzness* assumption that we discuss more in the sequel.

**Assumption 2.** We can get unbiased samples $\widetilde{G}(\mathbf{x}, \xi)$ and $\widetilde{G}(\mathbf{y}, \xi)$ with the same random seed $\xi \sim P$ where $P$ is a fixed distribution. Let $G$ be $L$-expected Lipschitz, that is:

$$\mathbb{E}_\xi \|\widetilde{G}(\mathbf{x}, \xi) - \widetilde{G}(\mathbf{y}, \xi)\|^2 \leq L_{\exp}\|\mathbf{x} - \mathbf{y}\|^2,$$

where $\mathbb{E}[\widetilde{G}(\mathbf{x}, \xi)] = G(\mathbf{x})$.

We finally formalize our assumptions concerning $G$.

**Assumption 3.** At each iteration, we sample $\xi \sim P$ where $P$ is a fixed distribution and set $\widetilde{G}$ such that

$$\mathbb{E}[\widetilde{G}(\mathbf{z}, \xi)] = G(\mathbf{z}) \quad \text{and}$$
$$\mathbb{E}\|\widetilde{G}(\mathbf{z}, \xi) - G(\mathbf{z})\|^2 \leq B^2\|\mathbf{z} - \mathbf{z}_0\|^2 + \sigma^2.$$

Clearly, this assumption can also be restated to hold only at $\mathbf{z} = \mathbf{z}_t$. As mentioned before, this remains the weakest

variance assumption under which optimal complexity guarantees are shown for convex minimization or monotone VIs (Alacaoglu et al., 2025). In view of the last two assumptions, let us point out that we overload the notation $\widetilde{G}$ and use it without the argument $\xi$ when the context is suitable, that is, we often use the notation $\widetilde{G}(\mathbf{x}) := \widetilde{G}(\mathbf{x}, \xi)$.

### 1.2. Contributions and Comparisons

In the sequel, we first state the algorithms which are either existing in the literature or are simple modifications over the existing algorithms. Our main contribution is *the analysis of these methods without the bounded variance or bounded domain assumptions*. We next highlight our main complexity results (by which we mean the number of stochastic first-order oracles —*sfo*— used by an algorithm) for obtaining an output $\mathbf{z}$ for which we have $\mathbb{E}[\mathrm{res}(\mathbf{z})] \leq \varepsilon$ (see (1.10)).

- We show, in Section 2, that for $\rho < \frac{1}{12L}$, a forward-backward-forward algorithm with mini-batching achieves the stochastic oracle complexity $\widetilde{O}(\varepsilon^{-4})$.

- We show, in Section 3, that for $\rho < 1/L$ (which is the tightest-known upper bound for the nonmonotonicity parameter $\rho$), an inexact fixed-point algorithm equipped with multilevel Monte Carlo (MLMC) estimator achieves the (expected) complexity $\widetilde{O}(\varepsilon^{-4})$.

- We show, in Section 4, that for $\rho < \frac{1}{16L}$ (see Thm. 4.1 for the details), a variance reduced forward-backward-forward method with Halpern anchoring achieves the complexity $\widetilde{O}(\varepsilon^{-4})$. This method is single-loop and only uses 3 stochastic oracles for $G$ at every iteration, not requiring any large mini-batch sizes.

- In Section 5, we test the numerical performance of our algorithms and illustrate two main points: *(i)* as predicted by theory, Algorithm 1 converges for a wider range of $\rho$ compared to earlier works, for a problem where even the *deterministic* extragradient algorithm, without noise, diverges (Gorbunov et al., 2023), *(ii)* introduction of Halpern anchoring in Algorithm 4 results in a more robust behavior with respect to tuning of the initial step size compared to the benchmark method in (Pethick et al., 2023).

To our knowledge, we provide the first complexity results where nonmonotonicity in view of (wMVI) can be tolerated for constrained min-max problems without bounded variance. Table 1 contains a comparison between our results and the existing results for SVI without bounded variance. Our results match the complexity results known with bounded variance for Halpern-based or variance reduced methods, see (Lee & Kim, 2021; Pethick et al., 2023). That is, the

stochastic first-order complexity $\widetilde{O}(\varepsilon^{-4})$ for the residual for solving weak MVI problems is the best-known even with the bounded variance assumption.

In fact, even for *constrained* monotone VIs with bounded variance, this complexity is the best-known for the residual guarantees (Iusem et al., 2017; Alacaoglu et al., 2025). Better complexity results for the residual for monotone problems exist only in the *unconstrained* case (where the residual norm becomes the operator norm) with bounded variance, see (Cai et al., 2022; Chen & Luo, 2024).

As we will discuss further in the sequel, three main results we prove are complementary: that is, they each extend the state-of-the-art in different directions and none of the proposed methods uniformly improve over the others. We refer to Table 1 for a summary.

**Discussion about Table 1.** In the nonmonotone case ($\rho > 0$ in Table 1), our main contributions are two-fold, *(i)* we provide complexity results for constrained SVI without bounded variance assumptions *(ii)* even in the unconstrained case, we improve the existing result of (Choudhury et al., 2023) because we show the results with the best-known range on $\rho$ and also without the knowledge of hard-to-compute quantities about the solution, that were required in (Choudhury et al., 2023, Thm. 4.5) for setting the mini-batch size.

In the monotone case, which is implied by the results with $\rho = 0$ in Table 1, our contributions are the following: compared to (Kotsalis et al., 2022; Iusem et al., 2017), we provide guarantees without large mini-batch sizes. Compared to (Alacaoglu et al., 2025), we provide guarantees under Assumption 1 which is weaker than Assumption 2 which requires a multi-point access to the oracle with a stronger Lipschitzness assumption.

A simple example of a $G$ that satisfies Assumption 1 but not Assumption 2 (see (Alacaoglu et al., 2024)) is $G(x) = G_1(x) + G_2(x)$ where $G_1(x) = x^2$ and $G_2(x) = -x^2$ and $\widetilde{G}$ is selected uniformly at random between $G_1, G_2$. However, Assumption 2 remains needed for single-loop algorithms even with bounded variance (Pethick et al., 2023).

**Other related results under relaxed noise assumptions.** Several related stochastic extragradient-type results also consider noise assumptions weaker than uniformly bounded variance. The guarantee in (Mishchenko et al., 2020, Theorem 2) without uniformly bounded variance is obtained in a strongly monotone/strongly convex setting. In the merely monotone case, the guarantee in (Mishchenko et al., 2020, Theorem 3) is stated for a different optimality measure and involves a supremum of the variance over the domain, which requires a uniform variance bound on unbounded domain. In Fact A.3, we prove that the relaxed noise assumptions in

this work imply ours.

The work of (Hsieh et al., 2020) allows distance dependent noise growth, but its rate guarantees rely on an error bound condition, which is stronger than monotonicity, and are stated for an unconstrained case. The related result of (Gorbunov et al., 2022) also considers relaxed variance assumptions, but their main convergence rates rely on quasi-strong monotonicity and uniqueness of the solution, and are also stated for unconstrained VIs. These assumptions are stronger than monotonicity and they are not satisfied by general monotone linear operators such as bilinear examples.

## 2. Results with Minibatching

We start with the most straightforward approach one may take to address our problem: an algorithm with large minibatches. On a high level, when the mini-batch size becomes large enough, the algorithm behaves more and more like a *full-gradient* algorithm. As a warm-up, we start with analyzing such an algorithm. In fact, this simple approach has complementary advantages to the other approaches we consider in the sequel. In particular, we require the weakest set of assumptions in this result and unlike our results in Section 3, the number of oracles used at each iteration is *deterministic* rather than *expected*. We discuss this further in Section 3.

### 2.1. Algorithmic Ideas

An idea that we use throughout the paper is the forward-backward-forward (FBF) algorithm of (Tseng, 2000), which iterates as

$$\mathbf{z}_{k+1/2} = \mathrm{prox}_{\eta_k r}(\mathbf{z}_k - \eta_k G(\mathbf{z}_k))$$
$$\mathbf{z}_{k+1} = \mathbf{z}_{k+1/2} - \eta_k \left( G(\mathbf{z}_{k+1/2}) - G(\mathbf{z}_k) \right).$$

One can easily extend this method to the stochastic case:

$$\mathbf{z}_{k+1/2} = \mathrm{prox}_{\eta_k r}(\mathbf{z}_k - \eta_k \widehat{G}(\mathbf{z}_k))$$
$$\mathbf{z}_{k+1} = \mathbf{z}_{k+1/2} - \eta_k \left( \widehat{G}(\mathbf{z}_{k+1/2}) - \widehat{G}(\mathbf{z}_k) \right), \tag{2.1}$$

where

$$\widehat{G}(\mathbf{z}_k) = \frac{1}{b_k} \sum_{i=1}^{b_k} \widetilde{G}(\mathbf{z}_k, \xi_k^i) \text{ and} \tag{2.2}$$

$$\widehat{G}(\mathbf{z}_{k+1/2}) = \frac{1}{b_k} \sum_{i=1}^{b_k} \widetilde{G}(\mathbf{z}_{k+1/2}, \xi_{k+1/2}^i), \tag{2.3}$$

for i.i.d. samples $\xi_k^i \sim P$ and $\xi_{k+1/2}^i \sim P$ for $i = 1, \ldots, b_k$.

**Remark 2.1** (Feasibility of the iterates). When $r = \delta_C$ is the indicator of a constraint set $C$, the prox step guarantees

| | Constraint | Need to know[†] | Range of $\rho$ | Complexity | Lipschitz | Single loop & MB[*]-free |
|---|---|---|---|---|---|---|
| (Choudhury et al., 2023) | × | $L, B, \mathbb{E}\|\widetilde{F}(\mathbf{x}^\star)\|^2,$ $\|\mathbf{z}_0 - \mathbf{z}^\star\|^2$ | $\rho < \frac{1}{2L}$ | $O(\varepsilon^{-4})$ | Asp. 1 | × (large MB) |
| (Kotsalis et al., 2022) (Boţ et al., 2021) | ✓ | $L, B$ | $\rho = 0$ | $O(\varepsilon^{-4})$ | Asp. 1 | × (large MB) |
| (Iusem et al., 2017) | ✓ | $L, B$ | $\rho = 0$ | $\widetilde{O}(\varepsilon^{-4})$ | Asp. 1 | × (large MB) |
| (Alacaoglu et al., 2025) | ✓ | $L, B$ | $\rho = 0$ | $\widetilde{O}(\varepsilon^{-4})$ | Asp. 2 | ✓ |
| Thm. 2.2 | ✓ | $L, B$ | $\rho < \frac{1}{12L}$ | $\widetilde{O}(\varepsilon^{-4})$ | Asp. 1 | × (incr. MB) |
| Thm. 3.1 | ✓ | $L, B$ | $\rho < \frac{1}{L}$ | $\widetilde{O}(\varepsilon^{-4})$ | Asp. 1 | × (loops) |
| Thm. 4.1 | ✓ | $L, B$ | $\rho < \frac{1}{16L}$[‡] | $\widetilde{O}(\varepsilon^{-4})$ | Asp. 2 | ✓ |

*Table 1.* Existing results without bounded variance assumption. [*]MB: mini-batch. [†]Constants that algorithms need to set parameters.[‡] The upper bound of $\rho$ in this case converges to $\frac{1}{16L}$ as a step size parameter gets smaller, see Theorem D.4 for details.

that $\mathbf{z}_{k+1/2} \in C$. Since our output is selected from these points, the output is feasible. The point $\mathbf{z}_{k+1}$ is an auxiliary point used to continue the algorithm and may lie outside $C$. This is allowed in our setting because $G$ is defined on $\mathbb{R}^m$. If $G$ is only defined on a restricted set, such as in (Farzin et al., 2025), then our algorithm may not be applicable.

This algorithmic construction is certainly not new, see, for example (Böhm et al., 2022) who analyzed a similar stochastic FBF under bounded variance, for complexity results on the expected gap function. Another related work by (Iusem et al., 2017) analyzed an extragradient method á la (Korpelevich, 1976) under Assumption 3 with mini-batching, in the case where $\rho = 0$. A similar result for FBF is studied in (Boţ et al., 2021).

### 2.2. Complexity Analysis

We start with our main complexity result of this section.

**Theorem 2.2** (See Theorem B.2 for the detailed parameter choices and proof). *Let Assumptions 1 and 3 hold and suppose that*

$$\rho < \frac{1}{12L}.$$

*For algorithm (2.1) with estimators computed as (2.2) with $b_k = \Theta(k \log(k+1))$ and $\eta_k = \Theta(1/L)$, we have*

$$\mathbb{E}[\mathrm{res}(\mathbf{z}^{out})] \leq \varepsilon \text{ with sfo complexity } \widetilde{O}(\varepsilon^{-4}),$$

*where $\mathbf{z}^{out}$ is generated by selecting an index $\hat{k}$ uniformly at random after running $K$ iterations and letting $\mathbf{z}^{out} = \mathbf{z}_{\hat{k}+1/2}$.*

**Remark 2.3.** The main limitation of this scheme is that it requires increasing mini-batch sizes and the upper bound for

$\rho$ is suboptimal (which we did not fully optimize). However, its strength lies in the fact that the number of stochastic oracles used at each iteration is deterministic. Compared to (Choudhury et al., 2023, Thm. 4.5), we can handle constrained problems and our batch sizes do not require any knowledge about the solution[2]. Compared to (Kotsalis et al., 2022; Iusem et al., 2017), we can handle $\rho > 0$.

**Proof sketch.** The main idea of this proof is similar to (Kotsalis et al., 2022) and (Iusem et al., 2017) with the exception that our analysis can tolerate a nonzero $\rho$. In particular, by using the specific form of the mini-batch size, one can first show that the iterates stay bounded in expectation:

$$\mathbb{E}\|\mathbf{z}_k - \mathbf{z}^\star\|^2 \leq C^2,$$

for a constant $C$, see Theorem B.2.

Then, one can use this bound on (1.7) to further upper bound the variance along our iterates, with a term depending on $C$. After this, the main analysis of FBF goes through. The main reason that we can tolerate a nonzero $\rho$ is the following chain of identities that follow from the definitions of $\mathbf{z}_{k+1/2}$, $\mathbf{z}_{k+1}$ in (2.1):

$$\mathbf{z}_{k+1/2} = \arg\min_{\mathbf{z}} r(\mathbf{z}) + \frac{1}{2\eta_k}\|\mathbf{z} - (\mathbf{z}_k - \eta_k \widehat{G}(\mathbf{z}_k))\|^2$$

$$\iff \mathbf{z}_{k+1/2} + \eta_k \partial r(\mathbf{z}_{k+1/2}) \ni \mathbf{z}_k - \eta_k \widehat{G}(\mathbf{z}_k)$$

$$\iff G(\mathbf{z}_{k+1/2}) + \partial r(\mathbf{z}_{k+1/2}) \ni \eta_k^{-1}(\mathbf{z}_k - \mathbf{z}_{k+1})$$
$$+ G(\mathbf{z}_{k+1/2}) - \widehat{G}(\mathbf{z}_{k+1/2}).$$

---

[2]Parameter $b_k$, given explicitly in Thm. B.2 depends on $L, \rho, B$ whereas the batch size in (Choudhury et al., 2023, Thm. 4.5) also depends on hard-to-compute $\|\mathbf{x}_0 - \mathbf{x}^\star\|^2$ and $\sigma_\star^2 = \mathbb{E}\|F_i(\mathbf{x}^\star)\|^2$.

Then, in view of the assumption (wMVI), we have

$$\langle \eta_k^{-1}(\mathbf{z}_k - \mathbf{z}_{k+1}), \mathbf{z}_{k+1/2} - \mathbf{z}^\star \rangle$$
$$+ \langle G(\mathbf{z}_{k+1/2}) - \widehat{G}(\mathbf{z}_{k+1/2}), \mathbf{z}_{k+1/2} - \mathbf{z}^\star \rangle$$
$$\geq -\rho \| \eta_k^{-1}(\mathbf{z}_k - \mathbf{z}_{k+1}) + G(\mathbf{z}_{k+1/2}) - \widehat{G}(\mathbf{z}_{k+1/2}) \|^2,$$

as $(\mathbf{z}_{k+1/2}, \eta_k^{-1}(\mathbf{z}_k - \mathbf{z}_{k+1}) + G(\mathbf{z}_{k+1/2}) - \widehat{G}(\mathbf{z}_{k+1/2})) \in$ gra$(G + \partial r)$.

Analyzing FBF in the standard way from this, one has

$$\mathbb{E}\|\mathbf{z}_{k+1} - \mathbf{z}^\star\|^2 \leq \mathbb{E}\|\mathbf{z}_k - \mathbf{z}^\star\|^2$$
$$+ \frac{2\rho}{\eta_k} \mathbb{E}\|\mathbf{z}_k - \mathbf{z}_{k+1} + \eta_k(G(\mathbf{z}_{k+1/2}) - \widehat{G}(\mathbf{z}_{k+1/2}))\|^2$$
$$+ \mathbb{E}\|\mathbf{z}_{k+1} - \mathbf{z}_{k+1/2}\|^2 - \mathbb{E}\|\mathbf{z}_k - \mathbf{z}_{k+1/2}\|^2. \qquad (2.4)$$

It is also easy to see by Young's inequality that

$$\frac{\rho}{\eta_k} \|\mathbf{z}_k - \mathbf{z}_{k+1} + \eta_k(G(\mathbf{z}_{k+1/2}) - \widehat{G}(\mathbf{z}_{k+1/2}))\|^2$$
$$= O\Big( \frac{\rho}{\eta_k} \big[ \|\mathbf{z}_k - \mathbf{z}_{k+1/2}\|^2 + \|\mathbf{z}_{k+1/2} - \mathbf{z}_{k+1}\|^2$$
$$+ \eta_k^2 \|\widehat{G}(\mathbf{z}_{k+1/2}) - G(\mathbf{z}_{k+1/2})\|^2 \big] \Big),$$

and

$$\|\mathbf{z}_{k+1} - \mathbf{z}_{k+1/2}\|^2$$
$$= O(\eta_k^2(L^2 \|\mathbf{z}_k - \mathbf{z}_{k+1/2}\|^2 + \|\widehat{G}(\mathbf{z}_k) - G(\mathbf{z}_k)\|^2$$
$$+ \|\widehat{G}(\mathbf{z}_{k+1/2}) - G(\mathbf{z}_{k+1/2})\|^2)).$$

Due to the last two bounds, with sufficiently small $\rho$, $\eta_k$ and large enough $b_k$ in (2.4), one can cancel the error terms coming from nonmonotonicity and also coming from $\|\mathbf{z}_{k+1} - \mathbf{z}_{k+1/2}\|^2$. The rest of the error terms can be proven to be small since they are controlled by the mini-batch sizes.

## 3. Results with MLMC

As mentioned earlier, the parameter $\rho$ determines the level of nonmonotonicity (or nonconvex-nonconcavity for min-max problems) in view of (wMVI). The largest-known upper bound for $\rho$ that can be tolerated by first-order methods is recently established by Alacaoglu et al. (2024) who showed the best-known first-order complexity results under $\rho < 1/L$. This work only focused on analyzing their method under bounded variance. In this section, we show how to generalize their analysis under Assumption 3 which relaxes the bounded variance assumption.

### 3.1. Algorithmic Ideas

To describe the algorithmic ideas, let us recall the definition of nonexpansiveness, which asks for the operator $T \colon \mathbb{R}^m \to \mathbb{R}^m$ to satisfy $\|T\mathbf{x} - T\mathbf{y}\| \leq \|\mathbf{x} - \mathbf{y}\|$. We next recall the

resolvent operator, which is a generalization of the proximal operator. In particular, we define the resolvent of $G + \partial r$ as

$$J_{\eta(G+\partial r)} = (\mathrm{Id} + \eta(G + \partial r))^{-1}, \qquad (3.1)$$

where we used Id to denote the identity operator.

In the special case of min-max problems, we have

$$J_{\eta(G+\partial r)}(\mathbf{z}_k) = \arg\min_{\mathbf{u}} \max_{\mathbf{v}} f(\mathbf{u}, \mathbf{v}) + h_1(\mathbf{u}) - h_2(\mathbf{v})$$
$$+ \frac{1}{2\eta} \|\mathbf{u} - \mathbf{u}_k\|^2 - \frac{1}{2\eta} \|\mathbf{v} - \mathbf{v}_k\|^2, \qquad (3.2)$$

which is a strongly convex-strongly concave optimization problem. To make this connection explicit, we will refer to the estimation of approximate solutions of (3.1) as the *proximal subproblem*. Approximate solutions to (3.1) or (3.2) can be found efficiently with stochastic gradients of $f$ or stochastic evaluations of $G$, see for example (Kotsalis et al., 2022).

Then, the algorithm in (Alacaoglu et al., 2024) proceeds by applying the classical inexact Krasnoselskii-Mann (KM) iteration to the *conically* quasi-nonexpansive operator (in view of (Bauschke et al., 2021)) $J_{\eta(G+\partial r)}$, where this property of $J_{\eta(G+\partial r)}$ under (wMVI) follows from the developments in (Bauschke et al., 2021), see (Alacaoglu et al., 2024) for the details.

As motivated in (Alacaoglu et al., 2024), to get the best complexity with this scheme, one needs a strong control over the bias of the estimation of $J_{\eta(G+\partial r)}$. Multilevel Monte-Carlo technique is a natural choice because it helps trade-off the bias and variance of the estimator. The primitive used in MLMC is an algorithm that can solve the strongly monotone proximal subproblem (see (3.1), (3.2)) with an optimal complexity. This is indeed where bounded variance was required in the work of (Alacaoglu et al., 2024). Hence we have precisely the same algorithm, but we show how it can be analyzed under more general Assumption 3.

### 3.2. Complexity Analysis

The main complexity result in this section is summarized in the following theorem.

**Theorem 3.1** (See Theorem C.5 for the detailed parameter choices and proof). *Let Assumptions 1, 3 hold and suppose that $\rho < 1/L$.*

*Then, for Alg. 1 with $\eta \leq \frac{1}{L}$ and $\alpha_k = \frac{\alpha}{\sqrt{k+2}\log(k+3)}$ (where the expressions for $N_k, M_k$ are in App. C), we have*

$$\mathbb{E}\|\mathbf{z}_{\hat{k}} - J_{\eta(G+\partial r)}(\mathbf{z}_{\hat{k}})\| \leq \varepsilon$$
*with expected sfo complexity* $\widetilde{O}(\varepsilon^{-4})$,

*where $\mathbf{z}_{\hat{k}}$ is selected uniformly at random after running the algorithm for $K$ iterations.*

---

**Algorithm 1** Inexact KM iteration (see (Alacaoglu et al., 2024))

---

**Input:** Parameters $\eta, N_k, M_k, \alpha = 1 - \frac{\rho}{\eta}, \alpha_k = \frac{\alpha}{\sqrt{k+2}\log(k+3)}$, $\mathbf{z}_0$, subroutine MLMC-FBF given in Algorithm 2

**for** $k = 0, 1, 2, \ldots, K-1$ **do**
$\quad \widetilde{J}_{\eta(G+\partial r)}^{(m)}(\mathbf{z}_k) = \text{MLMC-FBF}\left(\mathbf{z}_k, N_k, \eta\partial r, \text{Id} + \eta\widetilde{G}, 1 + \eta L\right)$ independently $m = 1, \ldots, M_k$
$\quad \widetilde{J}_{\eta(G+\partial r)}(\mathbf{z}_k) = \frac{1}{M_k}\sum_{i=1}^{M_k} \widetilde{J}_{\eta(G+\partial r)}^{(i)}(\mathbf{z}_k)$
$\quad \mathbf{z}_{k+1} = (1 - \alpha_k)\mathbf{z}_k + \alpha_k\widetilde{J}_{\eta(G+\partial r)}(\mathbf{z}_k)$

---

**Algorithm 2** MLMC-FBF$(\mathbf{x}_0, N, A, B, L_B)$ (see (Alacaoglu et al., 2024))

---

**Input:** Initial iterate $\mathbf{x}_0$

Define $\mathbf{y}^i = \text{FBF}(\mathbf{x}_0, 2^i, \widetilde{B}, A, L_B)$ for any $i \geq 0$. Draw $I \sim \text{Geom}(1/2)$

**Output:** $\mathbf{y}^{\text{out}} = \mathbf{y}^0 + 2^I(\mathbf{y}^I - \mathbf{y}^{I-1})$ if $2^I \leq N$, otherwise $\mathbf{y}^{\text{out}} = \mathbf{y}^0$.

---

**Remark 3.2.** One can translate this result to

$$\mathbb{E}[\text{res}(\mathbf{z}^{\text{out}})] \leq \varepsilon \text{ with expected sfo complexity } \widetilde{O}(\varepsilon^{-4}),$$

where $\mathbf{z}^{\text{out}}$ is generated by applying one step of (2.1) for problem (3.2), starting from $\mathbf{z}_{\hat{k}}$. A similar argument appears in (Cai et al., 2024, Lemma C.4).

**Remark 3.3.** From a theoretical point-of-view, this theorem gives us the strongest result because we obtain the best-known complexity results under (wMVI) with the best-known range on $\rho$ without bounded variance. On the other hand, a theoretical limitation of this approach is that the number of stochastic oracle calls at each iteration is random, causing the final complexity result to be on the *expected* number of oracle calls (which is shared by approaches relying on MLMC). A practical drawback is that the algorithm is not single-loop.

**Remark 3.4.** Similar to the discussion in Remark 2.1, our subsolver Algorithm 3 in this section generates iterates $\mathbf{x}_{k+1}$ which are not necessarily feasible. For cases where $G$ is not defined on the whole space, one can replace Algorithm 3 with a stochastic variant of the extragradient method (Korpelevich, 1976) and analyze it using the same ideas we used for analyzing Algorithm 3. For this variant, all the iterates where the operator $G$ is evaluated would be feasible.

We start with a lemma from (Alacaoglu et al., 2024) which only uses Assumption 1 and gives a bound on how close $\mathbf{z}_k$ is to be a fixed-point of the resolvent $J_{\eta(G+\partial r)}$. We use the notation $\mathbb{E}_k$ to denote the conditional expectation where we condition on the $\sigma$-algebra generated by the iterates until $\mathbf{z}_k$.

**Lemma 3.5.** *(See (Alacaoglu et al., 2024, Lemma C.8)) Let*

---

**Algorithm 3** FBF$(z_0, T, A, \widetilde{B}_{\text{in}}, L_B)$ from (Tseng, 2000) – Stochastic

---

**Input:** Initial iterate $\mathbf{x}_0$, $\widetilde{B}(\cdot) = \widetilde{B}_{\text{in}}(\cdot) - \mathbf{x}_0$

**for** $t = 0, 1, 2, \ldots, T-1$ **do**
$\quad \mathbf{x}_{t+1/2} = \text{prox}_{\tau_t r}(\mathbf{x}_t - \tau_t\widetilde{B}(\mathbf{x}_t))$
$\quad \mathbf{x}_{t+1} = \mathbf{x}_{t+1/2} + \tau_t\widetilde{B}(\mathbf{x}_t) - \tau_t\widetilde{B}(\mathbf{x}_{t+1/2})$

---

*Assumption 1 hold and $\rho < \eta$. We then have*

$$\sum_{k=0}^{K-1} \alpha_k\mathbb{E}\|(\text{Id} - J_{\eta(G+\partial r)})(\mathbf{z}_k)\|^2 = O\Big(\|\mathbf{z}_0 - \mathbf{z}^\star\|^2$$
$$+ \mathbb{E}\sum_{k=0}^{K-1}\Big[\alpha_k^2\|J_{\eta(G+\partial r)}(\mathbf{z}_k) - \widetilde{J}_{\eta(G+\partial r)}(\mathbf{z}_k)\|^2$$
$$+ \alpha_k\|\mathbf{z}_k - \mathbf{z}^\star\|\|J_{\eta(G+\partial r)}(\mathbf{z}_k) - \mathbb{E}_k[\widetilde{J}_{\eta(G+\partial r)}(\mathbf{z}_k)]\|\Big]\Big).$$

The main message of this lemma is that the error in the form of *bias*, that is $\|J_{\eta(G+\partial r)}(\mathbf{z}_k) - \mathbb{E}_k[\widetilde{J}_{\eta(G+\partial r)}(\mathbf{z}_k)]\|$ is only multiplied by $\alpha_k$ whereas the error in the form of *variance* $\mathbb{E}\|J_{\eta(G+\partial r)}(\mathbf{z}_k) - \widetilde{J}_{\eta(G+\partial r)}(\mathbf{z}_k)\|^2$ is multiplied by $\alpha_k^2$. Since $\alpha_k$ is small, this means that the analysis has the former error as the bottleneck.

This is where MLMC comes into play and the primitive in this estimator is a subsolver for the proximal subproblem. As mentioned before, this is the only place where (Alacaoglu et al., 2024) needed the bounded variance assumption. We next show that the bounded variance is in fact not needed: by a slightly different choice of step size, one can incorporate Assumption 3. Similar results appeared in (Kotsalis et al., 2022; Wang & Bertsekas, 2015) for different algorithms. We provide our analysis for simplicity and to be self-contained.

**Lemma 3.6.** *Let Assumptions 1, 3 hold. Let $\tau_t = \Theta\left(\frac{1}{t\mu + L^2/\mu}\right)$ and $\rho < \eta$. We have for the output of Alg. 3 that*

$$\mathbb{E}\|\mathbf{x}_T - J_{\eta(G+\partial r)}(\mathbf{z}_k)\|^2 = O(T^{-1}).$$

This lemma tells us how close $\mathbf{x}_T$ is to being a solution of the proximal subproblem, see (3.1) and (3.2). The main idea is extremely simple and can be found in standard textbooks for

---

**Algorithm 4** Variance reduced FBF with Halpern anchoring

---

**Input:** Initial iterate $\mathbf{z}_0$ and parameter $\rho \geq 0$.
**for** $k = 0, \ldots$ **do**
$\quad \bar{\mathbf{z}}_k = \beta_k \mathbf{z}_0 + (1 - \beta_k)\mathbf{z}_k$
$\quad \mathbf{z}_{k+1/2} = \text{prox}_{\gamma_k r}(\bar{\mathbf{z}}_k - \gamma_k \mathbf{g}_k)$
$\quad$ Set $\widetilde{G}_{k+1/2} = \widetilde{G}(\mathbf{z}_{k+1/2}, \xi_{k+1/2})$
$\quad \mathbf{z}_{k+1} = \bar{\mathbf{z}}_k - \tau_k(\bar{\mathbf{z}}_k - \mathbf{z}_{k+1/2} - \gamma_k \mathbf{g}_k + \gamma_k \widetilde{G}_{k+1/2})$
$\quad \mathbf{g}_{k+1} = \widetilde{G}(\mathbf{z}_{k+1}, \xi_{k+1}) + (1 - \alpha_k)(\mathbf{g}_k - \widetilde{G}(\mathbf{z}_k, \xi_{k+1}))$

---

the minimization case, see (Wright & Recht, 2022, Section 5.4.3). In particular, once we have strong convexity (or strong convexity-strong concavity), handling the additional error terms coming from Assumption 3 is straightforward, because the negative term that the strong convexity gives can be used to cancel these when the step size is chosen properly.

Let us provide the sketch of the argument here. Without the explicit constants, the main recursion for the stochastic FBF under bounded variance is the following

$$\mathbb{E}\|\mathbf{z}_{t+1} - \mathbf{z}^\star\|^2 \leq (1 - \tau_t \mu)\mathbb{E}\|\mathbf{z}_t - \mathbf{z}^\star\|^2 + O(\tau_t^2),$$

which one can use to get a rate $O(1/t)$, by induction. Under Assumption 3, the recursion becomes

$$\mathbb{E}\|\mathbf{z}_{t+1} - \mathbf{z}^\star\|^2 \leq (1 - \tau_t \mu + \tau_t^2 B^2)\mathbb{E}\|\mathbf{z}_t - \mathbf{z}^\star\|^2 + O(\tau_t^2).$$

Focusing on the coefficient of the first term on the right-hand side, the new error term scales as $\tau_t^2$. This can be absorbed in the term $-\tau_t \mu$ since for small $\tau_t$, we have $\tau_t^2 < \tau_t$. Hence, even though our problem is, strictly speaking, more general, the resulting recursion is the same as SGD and the same ideas as (Wright & Recht, 2022, Section 5.4.3) can be used.

Equipped with this result, which gives us the desired behavior from the subsolver under Assumption 3, the analysis follows the same steps as (Alacaoglu et al., 2024).

# 4. Variance Reduction

In this section, we design a single-loop algorithm that does not rely on large mini-batch sizes, that uses 3 unbiased samples of $G$ at every iteration.

## 4.1. Algorithmic Ideas

The three main components of this algorithm are *(i)* variance reduction via STORM estimator (Cutkosky & Orabona, 2019), *(ii)* the FBF method (Tseng, 2000) and *(iii)* the Halpern anchoring (Halpern, 1967; Yoon & Ryu, 2021). This algorithm and our analysis in this section are combining the ideas from (Pethick et al., 2023; Alacaoglu et al., 2025) to combine the best-of-both-worlds in each case. Compared to the former work, we show guarantees without bounded

variance and compared to the latter, our results can accommodate $\rho > 0$, all with the same complexity. An earlier work on the FBF variant used in Alg. 4 is (Giselsson, 2021).

In particular, when $\beta_k \equiv 0$ in Alg. 4, one can see that this algorithm reduces to the one in (Pethick et al., 2023). When $\tau = 1$, one can also notice the similarity between this algorithm and the classical FBF algorithm of (Tseng, 2000). Indeed, FBF is the same as Alg. 4 when $\beta_k \equiv 0, \tau = 1$ and when the estimators are replaced with the full operator evaluations.

As recently observed by Neu & Okolo (2024) for convex-concave min-max problems, including a nonzero $\beta_k$ is important to handle Assumption 3. With the selection of $\beta_k = \Theta(1/k)$, the anchoring in the algorithm corresponds to that of the classical Halpern iteration (Halpern, 1967; Yoon & Ryu, 2021).

## 4.2. Complexity Analysis

We state the main result and then the proof ideas. The complete proof is rather technical and is provided in App. D. See Theorem D.4 for a precise statement.

**Theorem 4.1** (See Theorem D.4 for detailed parameter choices and proof)**.** *Let Assumptions 1, 2, 3 hold and let $\rho \leq f(\bar{\tau})$ for a function $f$ where $f(\bar{\tau}) \to 1/(16L)$ as $\bar{\tau} \to 0$ (the precise form of $f(\bar{\tau})$ appears in Theorem D.4).*

*Then, for the output of Alg. 4 with $\beta_k = \Theta(1/k)$, $\gamma_k = \Theta(1/L_{\exp})$, $\tau_k = \Theta(1/\sqrt{k})$, we have*

$$\mathbb{E}[\text{res}(\mathbf{z}^{out})] \leq \varepsilon \text{ with sfo complexity } \widetilde{O}(\varepsilon^{-4}),$$

*where* $\Pr(\hat{k} = k) = \frac{\tau_k(k+3)}{\sum_{i=0}^{K-1} \tau_i(i+3)}$ *and* $\mathbf{z}^{out} = \mathbf{z}_{\hat{k}+1/2}$*.*

**Remark 4.2.** This result gives us the simplest algorithmic construction compared to earlier sections. We neither need large mini-batch sizes nor inner loops in this method. The cost is the need for the slightly stronger oracle and the Lipschitzness assumption given in Assumption 2, which, in fact, is common for variance reduction (Arjevani et al., 2023; Pethick et al., 2023). Another limitation compared to Sec. 3 is that the upper bound for $\rho$ is suboptimal (which is not optimized fully).

**Proof sketch.** One critical property of this FBF-based method in Algorithm 4 (compared to the algorithm considered in (Alacaoglu et al., 2025)) is the following property (similar to Section 2.2): Notice that $\frac{1}{\gamma_k}(\bar{\mathbf{z}}_k - \mathbf{z}_{k+1/2}) - \mathbf{g}_k + G(\mathbf{z}_{k+1/2}) \in (G + \partial r)(\mathbf{z}_{k+1/2})$ and hence using (wMVI):

$$\gamma_k^{-1}\langle \bar{\mathbf{z}}_k - \mathbf{z}_{k+1/2} - \gamma_k \mathbf{g}_k, \mathbf{z}_{k+1/2} - \mathbf{z}^\star \rangle$$
$$+ \langle G(\mathbf{z}_{k+1/2}), \mathbf{z}_{k+1/2} - \mathbf{z}^\star \rangle$$
$$\geq -\rho \|\gamma_k^{-1}(\bar{\mathbf{z}}_k - \mathbf{z}_{k+1/2}) - \mathbf{g}_k + G(\mathbf{z}_{k+1/2})\|^2.$$

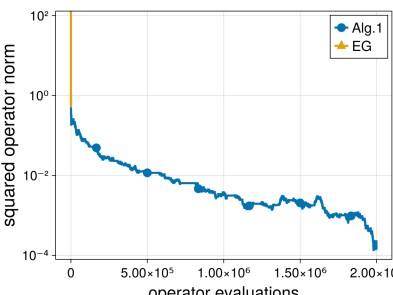 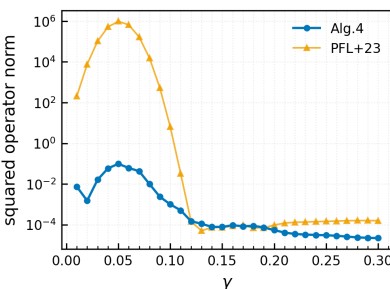 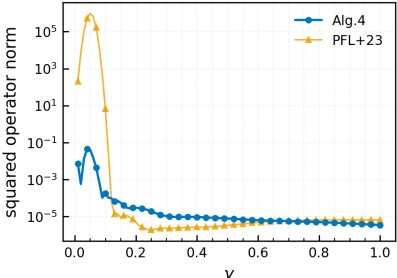

*Figure 1.* Left: Alg. 1 and EG for the counter-example problem. Middle and right: Alg. 4 and the algorithm of (Pethick et al., 2023) for an unconstrained problem with noise distributed as Student's t and Laplace-distribution.

On the other hand, analyzing the inner products on the left-hand side gives a recursion of the form

$$\mathbb{E}\|\mathbf{z}^\star - \mathbf{z}_{k+1}\|^2$$
$$\leq (1 - \beta_k)\mathbb{E}\|\mathbf{z}^\star - \mathbf{z}_k\|^2 + \beta_k\|\mathbf{z}^\star - \mathbf{z}_0\|^2$$
$$- \mathsf{Good}_k + \mathsf{Bad}_k + O(\tau_k^2)$$

where

$$\mathsf{Good}_k = \Theta(\tau_k)\mathbb{E}\|\mathbf{z}_k - \mathbf{z}_{k+1/2}\|^2 + \Theta(1)\mathbb{E}\|\mathbf{z}_k - \mathbf{z}_{k+1}\|^2$$
$$+ \Theta(\tau_k\beta_k)\mathbb{E}\|\mathbf{z}_0 - \mathbf{z}_{k+1/2}\|^2 + \Theta(\beta_k)\mathbb{E}\|\mathbf{z}_0 - \mathbf{z}_{k+1}\|^2$$

and

$$\mathsf{Bad}_k = \Theta(\tau_k)\mathbb{E}\|\mathbf{g}_k - G(\mathbf{z}_k)\|^2$$
$$+ \Theta\left(\frac{\rho\tau_k}{\gamma_k}\right)\mathbb{E}\|\bar{\mathbf{z}}_k - \mathbf{z}_{k+1/2} - \gamma_k(\mathbf{g}_k - G(\mathbf{z}_{k+1/2}))\|^2.$$

On a high level, one can see

$$\frac{\rho\tau_k}{\gamma_k}\mathbb{E}\|\bar{\mathbf{z}}_k - \mathbf{z}_{k+1/2} - \gamma_k(\mathbf{g}_k - G(\mathbf{z}_{k+1/2}))\|^2$$
$$\leq \Theta\left(\frac{\rho\tau_k}{\gamma_k} + \rho\tau_k\gamma_k L^2\right)\mathbb{E}\|\mathbf{z}_k - \mathbf{z}_{k+1/2}\|^2$$
$$+ \Theta\left(\frac{\rho\tau_k\beta_k}{\gamma_k}\right)\mathbb{E}\|\mathbf{z}_0 - \mathbf{z}_{k+1/2}\|^2$$
$$+ \Theta\left(\rho\tau_k\gamma_k\right)\mathbb{E}\|\mathbf{g}_k - G(\mathbf{z}_k)\|^2$$

By selecting the parameters accordingly, when $\rho$ is small enough (see Thm. 4.1) and the orders of the parameters are $\tau_k = \Theta(1/\sqrt{k})$, $\beta_k = \Theta(1/k)$, $\gamma_k = \Theta(1/L)$, the error terms can be cancelled.

The final error will be due to the variance error $\mathbb{E}\|\mathbf{g}_k - G(\mathbf{z}_k)\|^2$. The control for this comes from the standard bounds on STORM estimator (Cutkosky & Orabona, 2019), see Lemma D.3. Routine calculations help us finish the proof.

## 5. Numerical Results

Our numerical experiments evaluate robustness across three benchmarks. (*Left*) We reproduce the counter-example from (Gorbunov et al., 2023, Thm. 4.3) with operator $F(x) = LAx$ where $A$ is a rotation matrix. We set $\rho = \frac{1}{2L}$, a regime in which EG is known to lose stability on this instance. In this setting, EG diverges as expected, whereas Alg. 1 converges. (*Middle/Right*) Here, we compare Alg. 4 with (Pethick et al., 2023, Alg. 1) on the unconstrained quadratic Example 2 of (Pethick et al., 2023) (see (E.2)) with noisy oracle: the middle figure uses noise with Student's $t$-distribution with parameter $\nu = 2$ and the right figure uses Laplace noise. For each step size $\gamma$ on a fixed grid, we run 7 independent seeds per method and plot the mean of the results. As $\gamma$ decreases, Alg. 4 remains stable and convergent over a broad range, whereas (Pethick et al., 2023, Alg. 1) diverges for small $\gamma$, demonstrating stronger robustness of Alg. 4 to noisy oracles. See App. E for details. Code for reproducing all numerical experiments is available at https://github.com/JunHyun-K/stochastic-vi-without-bounded-variance.

## 6. Conclusions

We provided three different algorithms that all achieve complexity $\widetilde{O}(\varepsilon^{-4})$ for solving nonmonotone stochastic variational inequalities without uniformly bounded variance (which is the best-known complexity, up-to $\log$ terms, even for constrained monotone SVIs with bounded variance). These algorithms provide distinct advantages compared to each other and hence are complementary. An open question is to design and analyze a unified algorithm that can achieve the *best-of-three-worlds*: a single-loop algorithm with the best range for $\rho$ and without large mini-batches. Moreover, for problems satisfying (wMVI), it is still unknown whether it is possible to improve $O(\varepsilon^{-4})$ for getting $\mathbb{E}[\mathrm{res}(\mathbf{z})] \leq \varepsilon$, even with bounded variance and no constraints.

## Acknowledgments

Ahmet Alacaoglu acknowledges the support of the Natural Sciences and Engineering Research Council of Canada (NSERC), [funding reference number RGPIN-2025-06634].

## Impact Statement

"This paper presents work whose goal is to advance the field of Machine Learning. There are many potential societal consequences of our work, none which we feel must be specifically highlighted here."

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

# A. Preliminaries

**Notation.** In the sequel, we use the notation $\mathbb{E}_k$ to denote the conditional expectation where we condition on the $\sigma$-algebra generated by all the iterates until $\mathbf{z}_k$. The notation $\mathbb{E}_{k+1/2}$ is defined similarly.

**Fact A.1.** *The inequalities* (1.7) *and* (1.8) *are equivalent up to a redefinition of constants.*

*Proof.* Young's inequality gives $\|\mathbf{z} - \mathbf{z}^\star\|^2 \leq 2\|\mathbf{z} - \mathbf{z}_0\|^2 + 2\|\mathbf{z}_0 - \mathbf{z}^\star\|^2$. Substituting this into (1.7) gives

$$\mathbb{E}\|\widetilde{G}(\mathbf{z}) - G(\mathbf{z})\|^2 \leq 2B^2\|\mathbf{z} - \mathbf{z}_0\|^2 + (2B^2\|\mathbf{z}_0 - \mathbf{z}^\star\|^2 + \sigma^2),$$

which matches (1.8) after renaming constants since $\|\mathbf{z}_0 - \mathbf{z}^\star\|^2$ is a constant.

The reverse direction follows similarly by using $\|\mathbf{z} - \mathbf{z}_0\|^2 \leq 2\|\mathbf{z} - \mathbf{z}^\star\|^2 + 2\|\mathbf{z}^\star - \mathbf{z}_0\|^2$. $\qquad\square$

**Example A.2** (A concrete example where bounded variance fails but (1.7) holds). We give a simple example illustrating why (BV) can fail on unbounded domains, while (1.7) holds naturally. Consider the unconstrained stochastic bilinear saddle point problem.

$$\min_{\mathbf{x}\in\mathbb{R}^n} \max_{\mathbf{y}\in\mathbb{R}^m} \langle \mathbf{y}, A\mathbf{x}\rangle,$$

where $A \in \mathbb{R}^{m\times n}$. Let $\mathbf{z} = \binom{\mathbf{x}}{\mathbf{y}} \in \mathbb{R}^{n+m}$, and let $\mathbf{a}_i^\top$ and $\mathbf{b}_j$ denote the $i$th row and $j$th column of $A$, respectively. The associated operator is

$$G(\mathbf{x}, \mathbf{y}) = \begin{pmatrix} A^\top \mathbf{y} \\ -A\mathbf{x} \end{pmatrix}.$$

Let $\xi = (i, j)$, where $i$ and $j$ are sampled uniformly from $\{1, \ldots, m\}$ and $\{1, \ldots, n\}$. Define the stochastic operator

$$\widetilde{G}(\mathbf{z}, \xi) = \begin{pmatrix} n(\mathbf{b}_j^\top \mathbf{y})\mathbf{e}_j \\ -m(\mathbf{a}_i^\top \mathbf{x})\mathbf{e}_i \end{pmatrix}$$

where $\mathbf{e}_j \in \mathbb{R}^n$ and $\mathbf{e}_i \in \mathbb{R}^m$ are standard basis vectors. Then

$$\mathbb{E}_\xi[\widetilde{G}(\mathbf{z}, \xi)] = \begin{pmatrix} \sum_{j=1}^n (\mathbf{b}_j^\top \mathbf{y})\mathbf{e}_j \\ -\sum_{i=1}^m (\mathbf{a}_i^\top \mathbf{x})\mathbf{e}_i \end{pmatrix} = \begin{pmatrix} A^\top \mathbf{y} \\ -A\mathbf{x} \end{pmatrix} = G(\mathbf{z}).$$

Note that we have $\mathbf{z}^\star = \binom{\mathbf{0}}{\mathbf{0}}$.

Since $\widetilde{G}(\mathbf{z}, \xi)$ is unbiased and $\mathbb{E}_\xi\|\widetilde{G}(\mathbf{z}, \xi)\|^2 = n\sum_{j=1}^n (\mathbf{b}_j^\top \mathbf{y})^2 + m\sum_{i=1}^m (\mathbf{a}_i^\top \mathbf{x})^2 = n\|A^\top \mathbf{y}\|^2 + m\|A\mathbf{x}\|^2$, we have

$$\mathbb{E}_\xi\|\widetilde{G}(\mathbf{z}, \xi) - G(\mathbf{z})\|^2 = \mathbb{E}_\xi\|\widetilde{G}(\mathbf{z}, \xi)\|^2 - \|G(\mathbf{z})\|^2 = (n-1)\|A^\top \mathbf{y}\|^2 + (m-1)\|A\mathbf{x}\|^2.$$

This quantity is not uniformly bounded on an unbounded domain in general. For example, if $A \neq 0$ and $n > 1$, choose $\mathbf{y}$ such that $A^\top \mathbf{y} \neq 0$ and take $\mathbf{z}_t = \binom{\mathbf{0}}{t\mathbf{y}}$. Then

$$\mathbb{E}_\xi\|\widetilde{G}(\mathbf{z}_t, \xi) - G(\mathbf{z}_t)\|^2 = t^2(n-1)\|A^\top \mathbf{y}\|^2 \to \infty \quad \text{as} \quad t \to \infty.$$

Hence the bounded variance assumption (BV) fails on unbounded domains.

On the other hand, (1.7) holds immediately. Indeed,

$$\mathbb{E}_\xi\|\widetilde{G}(\mathbf{z}, \xi) - G(\mathbf{z})\|^2 \leq \max\{m-1, n-1\}\|A\|^2(\|\mathbf{x}\|^2 + \|\mathbf{y}\|^2) = B^2\|\mathbf{z} - \mathbf{z}^\star\|^2,$$

where one can take $B^2 = \max\{m-1, n-1\}\|A\|^2$. Thus (1.7) holds with $\sigma = 0$, while bounded variance assumption does not hold in general.

**Fact A.3.** *If $\widetilde{G}(\cdot, \xi)$ is Lipschitz and $\mathbb{E}\|\widetilde{G}(\mathbf{x}^\star, \xi)\|^2 \leq \sigma_\star^2$ where $\mathbf{x}^\star$ is the solution, then the BG assumption holds for $\widetilde{G}$.*

*Proof.* We use Lipschitzness of each component to derive

$$\mathbb{E}\|G(\mathbf{x}, \xi) - G(\mathbf{x})\|^2 \leq \mathbb{E}\|G(\mathbf{x}, \xi)\|^2 \leq 2\mathbb{E}\|G(\mathbf{x}, \xi) - G(\mathbf{x}^\star, \xi)\|^2 + 2\mathbb{E}\|G(\mathbf{x}^\star, \xi)\|^2 \leq 2L^2\|\mathbf{x} - \mathbf{x}^\star\|^2 + 2\sigma_\star^2,$$

where the first step is by $\mathbb{E}\|X - \mathbb{E}X\|^2 \leq \mathbb{E}\|X\|^2$. $\qquad\square$

This fact implies that Assumptions 2 and 4 in the work of Mishchenko et al. (2020) are stronger than our assumption since it implies Assumption 3.

## B. Proofs for Section 2

We start by analyzing one iteration of the algorithm given in Equation (2.1). This is a rather standard analysis in the monotone case, see for example (Böhm et al., 2022). What we have in this lemma is a straightforward generalization that includes the (wMVI) assumption instead of monotonicity.

**Lemma B.1.** *Let Assumption 1 hold. Then we have*

$$\mathbb{E}\|\mathbf{z}^\star - \mathbf{z}_{k+1}\|^2 \le \mathbb{E}\|\mathbf{z}^\star - \mathbf{z}_k\|^2 + \left(\frac{100\rho}{49\eta_k}(1 + \eta_k L)^2 + \frac{50}{49}\eta_k^2 L^2 - 1\right)\mathbb{E}\|\mathbf{z}_k - \mathbf{z}_{k+1/2}\|^2$$
$$+ 50\eta_k^2 \mathbb{E}\|\widehat{G}(\mathbf{z}_{k+1/2}) - G(\mathbf{z}_{k+1/2})\|^2$$
$$+ \left(100\rho\eta_k + 50\eta_k^2\right)\mathbb{E}\|\widehat{G}(\mathbf{z}_k) - G(\mathbf{z}_k)\|^2.$$

*Proof.* The definitions of $\mathbf{z}_{k+1/2}$ and the proximal operator give us that

$$\mathbf{z}_{k+1/2} = \arg\min_{\mathbf{z}} r(\mathbf{z}) + \frac{1}{2\eta_k}\|\mathbf{z} - (\mathbf{z}_k - \eta_k\widehat{G}(\mathbf{z}_k))\|^2$$
$$\iff \frac{1}{\eta_k}(\mathbf{z}_{k+1/2} - \mathbf{z}_k) + \widehat{G}(\mathbf{z}_k) + \partial r(\mathbf{z}_{k+1/2}) \ni 0$$
$$\iff G(\mathbf{z}_{k+1/2}) + \partial r(\mathbf{z}_{k+1/2}) \ni \frac{1}{\eta_k}(\mathbf{z}_k - \mathbf{z}_{k+1/2}) + G(\mathbf{z}_{k+1/2}) - \widehat{G}(\mathbf{z}_k)$$
$$\iff G(\mathbf{z}_{k+1/2}) + \partial r(\mathbf{z}_{k+1/2}) \ni \frac{1}{\eta_k}(\mathbf{z}_k - \mathbf{z}_{k+1}) + G(\mathbf{z}_{k+1/2}) - \widehat{G}(\mathbf{z}_{k+1/2}),$$

where the last line used the definition of $\mathbf{z}_{k+1}$.

Then, in view of the assumption (wMVI), we have

$$\langle\eta_k^{-1}(\mathbf{z}_k - \mathbf{z}_{k+1}) + G(\mathbf{z}_{k+1/2}) - \widehat{G}(\mathbf{z}_{k+1/2}), \mathbf{z}_{k+1/2} - \mathbf{z}^\star\rangle$$
$$\ge -\rho\|\eta_k^{-1}(\mathbf{z}_k - \mathbf{z}_{k+1}) + G(\mathbf{z}_{k+1/2}) - \widehat{G}(\mathbf{z}_{k+1/2})\|^2, \tag{B.1}$$

because $(\mathbf{z}_{k+1/2}, \eta_k^{-1}(\mathbf{z}_k - \mathbf{z}_{k+1}) + G(\mathbf{z}_{k+1/2}) - \widehat{G}(\mathbf{z}_{k+1/2})) \in \text{gra}(G + \partial r)$.

We now continue with the standard analysis of FBF-type methods (Tseng, 2000). Let us first note

$$2\langle\mathbf{z}_k - \mathbf{z}_{k+1}, \mathbf{z}_{k+1/2} - \mathbf{z}^\star\rangle$$
$$= 2\langle\mathbf{z}_k - \mathbf{z}_{k+1}, \mathbf{z}_{k+1} - \mathbf{z}^\star\rangle + 2\langle\mathbf{z}_k - \mathbf{z}_{k+1}, \mathbf{z}_{k+1/2} - \mathbf{z}_{k+1}\rangle$$
$$= \|\mathbf{z}_k - \mathbf{z}^\star\|^2 - \|\mathbf{z}_{k+1} - \mathbf{z}^\star\|^2 + \|\mathbf{z}_{k+1} - \mathbf{z}_{k+1/2}\|^2 - \|\mathbf{z}_k - \mathbf{z}_{k+1/2}\|^2.$$

Moreover, by $\mathbf{z}_{k+1/2}$ being deterministic when we condition on the history up to and including $\mathbf{z}_{k+1/2}$ and also the unbiasedness of $\widehat{G}$, we have

$$\mathbb{E}_{k+1/2}\langle G(\mathbf{z}_{k+1/2}) - \widehat{G}(\mathbf{z}_{k+1/2}), \mathbf{z}_{k+1/2} - \mathbf{z}^\star\rangle$$
$$= \langle\mathbb{E}_{k+1/2}[G(\mathbf{z}_{k+1/2}) - \widehat{G}(\mathbf{z}_{k+1/2})], \mathbf{z}_{k+1/2} - \mathbf{z}^\star\rangle$$
$$= 0.$$

We now multiply both sides of (B.1) by $2\eta_k$, take expectation, use tower property, and then use the last two estimates in (B.1) to get

$$\mathbb{E}\|\mathbf{z}_{k+1} - \mathbf{z}^\star\|^2 \le \mathbb{E}\|\mathbf{z}_k - \mathbf{z}^\star\|^2 + \mathbb{E}\|\mathbf{z}_{k+1} - \mathbf{z}_{k+1/2}\|^2 - \mathbb{E}\|\mathbf{z}_k - \mathbf{z}_{k+1/2}\|^2$$
$$+ \frac{2\rho}{\eta_k}\mathbb{E}\|\mathbf{z}_k - \mathbf{z}_{k+1} + \eta_k(G(\mathbf{z}_{k+1/2}) - \widehat{G}(\mathbf{z}_{k+1/2}))\|^2. \tag{B.2}$$

Young's inequality gives, for any $c_1 > 0$,

$$\|\mathbf{z}_k - \mathbf{z}_{k+1} + \eta_k(G(\mathbf{z}_{k+1/2}) - \widehat{G}(\mathbf{z}_{k+1/2}))\|^2$$
$$= \|\mathbf{z}_k - \mathbf{z}_{k+1/2} + \eta_k(G(\mathbf{z}_{k+1/2}) - \widehat{G}(\mathbf{z}_k))\|^2$$
$$\leq (1 + c_1)\|\mathbf{z}_k - \mathbf{z}_{k+1/2} + \eta_k(G(\mathbf{z}_{k+1/2}) - G(\mathbf{z}_k))\|^2 + \left(1 + \frac{1}{c_1}\right)\eta_k^2\|\widehat{G}(\mathbf{z}_k) - G(\mathbf{z}_k)\|^2$$
$$\leq (1 + c_1)(1 + \eta_k L)^2\|\mathbf{z}_k - \mathbf{z}_{k+1/2}\|^2 + \left(1 + \frac{1}{c_1}\right)\eta_k^2\|\widehat{G}(\mathbf{z}_k) - G(\mathbf{z}_k)\|^2.$$

We next use Young's inequality and Lipschitzness of $G$ to obtain, for any $d_1 > 0$,

$$\mathbb{E}\|\mathbf{z}_{k+1} - \mathbf{z}_{k+1/2}\|^2$$
$$= \eta_k^2\mathbb{E}\|\widehat{G}(\mathbf{z}_k) - \widehat{G}(\mathbf{z}_{k+1/2})\|^2$$
$$\leq \eta_k^2((1 + d_1)L^2\mathbb{E}\|\mathbf{z}_k - \mathbf{z}_{k+1/2}\|^2 + \left(1 + \frac{1}{d_1}\right)\mathbb{E}\|\widehat{G}(\mathbf{z}_k) - G(\mathbf{z}_k) + G(\mathbf{z}_{k+1/2}) - \widehat{G}(\mathbf{z}_{k+1/2})\|^2)$$
$$\leq \eta_k^2((1 + d_1)L^2\mathbb{E}\|\mathbf{z}_k - \mathbf{z}_{k+1/2}\|^2 + \left(1 + \frac{1}{d_1}\right)(\mathbb{E}\|\widehat{G}(\mathbf{z}_k) - G(\mathbf{z}_k)\|^2 + \mathbb{E}\|G(\mathbf{z}_{k+1/2}) - \widehat{G}(\mathbf{z}_{k+1/2})\|^2)),$$

because,

$$\mathbb{E}\|\widehat{G}(\mathbf{z}_k) - G(\mathbf{z}_k) + G(\mathbf{z}_{k+1/2}) - \widehat{G}(\mathbf{z}_{k+1/2})\|^2 = \mathbb{E}\|\widehat{G}(\mathbf{z}_k) - G(\mathbf{z}_k)\|^2 + \mathbb{E}\|G(\mathbf{z}_{k+1/2}) - \widehat{G}(\mathbf{z}_{k+1/2})\|^2$$
$$+ 2\mathbb{E}\langle\widehat{G}(\mathbf{z}_k) - G(\mathbf{z}_k), G(\mathbf{z}_{k+1/2}) - \widehat{G}(\mathbf{z}_{k+1/2})\rangle$$

by conditioning on the history up to and including $\mathbf{z}_{k+1/2}$, we have

$$\mathbb{E}\langle\widehat{G}(\mathbf{z}_k) - G(\mathbf{z}_k), G(\mathbf{z}_{k+1/2}) - \widehat{G}(\mathbf{z}_{k+1/2})\rangle$$
$$= \mathbb{E}\langle\widehat{G}(\mathbf{z}_k) - G(\mathbf{z}_k), \mathbb{E}_{k+1/2}[G(\mathbf{z}_{k+1/2}) - \widehat{G}(\mathbf{z}_{k+1/2})]\rangle$$
$$= 0.$$

Plugging in the last two estimates into (B.2) and combining the like terms,

$$\mathbb{E}\|\mathbf{z}_{k+1} - \mathbf{z}^\star\|^2 \leq \mathbb{E}\|\mathbf{z}_k - \mathbf{z}^\star\|^2 + \left(\frac{2\rho}{\eta_k}(1 + c_1)(1 + \eta_k L)^2 + \eta_k^2(1 + d_1)L^2 - 1\right)\mathbb{E}\|\mathbf{z}_k - \mathbf{z}_{k+1/2}\|^2$$
$$+ \left(2\rho\eta_k\left(1 + \frac{1}{c_1}\right) + \eta_k^2\left(1 + \frac{1}{d_1}\right)\right)\mathbb{E}\|\widehat{G}(\mathbf{z}_k) - G(\mathbf{z}_k)\|^2$$
$$+ \eta_k^2\left(1 + \frac{1}{d_1}\right)\mathbb{E}\|\widehat{G}(\mathbf{z}_{k+1/2}) - G(\mathbf{z}_{k+1/2})\|^2.$$

Let us pick

$$c_1 = d_1 = \frac{1}{49}.$$

Using these values conclude the proof. $\qquad\square$

Let us restate Theorem 2.2 and the provide its proof.

**Theorem B.2.** *(Detailed restatement of Theorem 2.2) Let Assumptions 1 and 3 hold and suppose that*

$$\rho < \frac{41}{490L} \approx \frac{1}{12L}, \quad \eta_k = \frac{2}{5L}, \quad \delta = 1 - \frac{50}{49}\eta_k^2 L^2 - \frac{100\rho}{49\eta_k}(1 + \eta_k L)^2$$

*where $\delta > 0$ by definition, and*

$$b_k = \bar{b}(k+1)\log^2(k+3), \ where \ \bar{b} = \frac{48B^2}{L^2\delta}.$$

*Then, the algorithm in* (2.1) *with gradient estimators computed as* (2.2) *and parameters as above outputs* $\mathbf{z}^{out}$ *such that*

$$\mathbb{E}[\mathrm{res}(\mathbf{z}^{out})] \leq \varepsilon \ with \ stochastic \ oracle \ complexity \ \widetilde{O}(\varepsilon^{-4}),$$

*where* $\mathbf{z}^{out} = \mathbf{z}_{\hat{k}+1/2}$ *and* $\hat{k}$ *is selected uniformly at random from* $\{0, \dots, K-1\}$.

**Remark B.3.** The range of $\rho$ obtained in this result is indeed rather pessimistic. We did not aim to optimize this constant and it can be improved by using, for example, the two step variant from (Pethick et al., 2022). That is, one can analyze the algorithm

$$\text{Define } \widehat{F}(\mathbf{z}_k) := \mathbf{z}_k - \eta\widehat{G}(\mathbf{z}_k), \widehat{F}(\mathbf{z}_{k+1/2}) := \mathbf{z}_{k+1/2} - \eta\widehat{G}(\mathbf{z}_{k+1/2})$$

$$\mathbf{z}_{k+1/2} = J_{\eta\partial r}(\widehat{F}(\mathbf{z}_k))$$

$$\mathbf{z}_{k+1/2} = \mathbf{z}_k - \tau(\widehat{F}(\mathbf{z}_k) - \widehat{F}(\mathbf{z}_{k+1/2})),$$

where $\tau$ is smaller than $\eta$ proportionally to obtain a better range for $\rho$, by using the insights from (Pethick et al., 2023; 2022). In either case, the resulting range for $\rho$ will be smaller than the best-known range given in Section 3. As a result, we skip this variant for brevity.

*Proof.* Let us pick

$$\eta_k^2 = \frac{4}{25L^2},$$

where we assume that $\rho < \frac{41}{490L} \approx \frac{1}{12L}$. This implies that

$$\delta = 1 - \frac{50}{49}\eta_k^2 L^2 - \frac{100\rho}{49\eta_k}(1+\eta_k L)^2 = \frac{41}{49} - 10\rho L > 0,$$

and then the inequality in Lemma B.1 becomes

$$\delta\mathbb{E}\|\mathbf{z}_k - \mathbf{z}_{k+1/2}\|^2 \leq \mathbb{E}\|\mathbf{z}^\star - \mathbf{z}_k\|^2 - \mathbb{E}\|\mathbf{z}^\star - \mathbf{z}_{k+1}\|^2$$

$$+ \frac{8}{L^2}\mathbb{E}\|\widehat{G}(\mathbf{z}_{k+1/2}) - G(\mathbf{z}_{k+1/2})\|^2$$

$$+ \frac{8+40\rho L}{L^2}\mathbb{E}\|\widehat{G}(\mathbf{z}_k) - G(\mathbf{z}_k)\|^2. \tag{B.3}$$

We next bound the terms in the last two lines. By using the fact that $\widehat{G}$ is a mini-batch estimator that averages unbiased i.i.d. samples, we have, by standard estimations (see e.g. (Kotsalis et al., 2022, Eq. (3.45))) that

$$\mathbb{E}\|\widehat{G}(\mathbf{z}_{k+1/2}) - G(\mathbf{z}_{k+1/2})\|^2 \leq \frac{1}{b_k}\mathbb{E}\|\widetilde{G}(\mathbf{z}_{k+1/2}, \xi_{k+1/2}^1) - G(\mathbf{z}_{k+1/2})\|^2$$

$$\leq \frac{1}{b_k}\left(B^2\mathbb{E}\|\mathbf{z}_{k+1/2} - \mathbf{z}_0\|^2 + \sigma^2\right)$$

$$\leq \frac{3B^2}{b_k}\mathbb{E}\left[\|\mathbf{z}_{k+1/2} - \mathbf{z}_k\|^2 + \|\mathbf{z}_k - \mathbf{z}^\star\|^2 + \|\mathbf{z}^\star - \mathbf{z}_0\|^2\right] + \frac{\sigma^2}{b_k}$$

and similarly,

$$\mathbb{E}\|\widehat{G}(\mathbf{z}_k) - G(\mathbf{z}_k)\|^2 \leq \frac{2B^2}{b_k}\mathbb{E}\left[\|\mathbf{z}_k - \mathbf{z}^\star\|^2 + \|\mathbf{z}^\star - \mathbf{z}_0\|^2\right] + \frac{\sigma^2}{b_k}.$$

After plugging in the last two estimates into (B.3) and using

$$\bar{b} = \frac{48B^2}{L^2\delta} \text{ and } b_k = \bar{b}(k+1)\log^2(k+3),$$

we obtain

$$\frac{\delta}{2}\mathbb{E}\|\mathbf{z}_k - \mathbf{z}_{k+1/2}\|^2 \leq \left(1 + \frac{\delta}{(k+1)\log^2(k+3)}\right)\mathbb{E}\|\mathbf{z}^\star - \mathbf{z}_k\|^2 - \mathbb{E}\|\mathbf{z}^\star - \mathbf{z}_{k+1}\|^2$$
$$+ \frac{1}{(k+1)\log^2(k+3)}\left(\delta\|\mathbf{z}^\star - \mathbf{z}_0\|^2 + \frac{20\sigma^2}{L^2\bar{b}}\right). \tag{B.4}$$

First, we discard the nonnegative term on the left-hand side and obtain

$$\mathbb{E}\|\mathbf{z}^\star - \mathbf{z}_{k+1}\|^2 \leq \left(1 + \frac{\delta}{(k+1)\log^2(k+3)}\right)\mathbb{E}\|\mathbf{z}^\star - \mathbf{z}_k\|^2$$
$$+ \frac{1}{(k+1)\log^2(k+3)}\left(\delta\|\mathbf{z}^\star - \mathbf{z}_0\|^2 + \frac{20\sigma^2}{L^2\bar{b}}\right).$$

Note how this recursion is of the same form as (Bauschke & Combettes, 2017, Lemma 5.31) because $\sum_{k=0}^\infty \frac{1}{(k+1)\log^2(k+3)} < 2$, hence arguing in the same way gives us that

$$\mathbb{E}\|\mathbf{z}^\star - \mathbf{z}_k\|^2 \leq C < +\infty. \tag{B.5}$$

for an easily computable $C$. Then plugging in this uniform upper bound on $\mathbb{E}\|\mathbf{z}^\star - \mathbf{z}_k\|^2$ into (B.4) gives

$$\frac{1}{K}\sum_{k=0}^{K-1}\mathbb{E}\|\mathbf{z}_k - \mathbf{z}_{k+1/2}\|^2 = O\left(\frac{1}{K}\right), \tag{B.6}$$

since $\sum_{k=0}^\infty \frac{1}{(k+1)\log^2(k+3)} < 2$.

We finally bound the quantity $\frac{1}{K}\sum_{k=1}^K \mathbb{E}[\mathrm{res}^2(\mathbf{z}_{k+1/2})] = \frac{1}{K}\sum_{k=1}^K \mathbb{E}[\min_{\mathbf{u}\in(G+\partial r)\mathbf{z}_{k+1/2}}\|\mathbf{u}\|^2]$. For this, note that

$$\mathbf{z}_{k+1/2} = \arg\min_{\mathbf{z}} r(\mathbf{z}) + \frac{1}{2\eta_k}\|\mathbf{z} - (\mathbf{z}_k - \eta_k\widehat{G}(\mathbf{z}_k))\|^2$$
$$\iff \mathbf{z}_{k+1/2} + \eta_k\partial r(\mathbf{z}_{k+1/2}) \ni \mathbf{z}_k - \eta_k\widehat{G}(\mathbf{z}_k)$$
$$\iff G(\mathbf{z}_{k+1/2}) + \partial r(\mathbf{z}_{k+1/2}) \ni \eta_k^{-1}(\mathbf{z}_k - \mathbf{z}_{k+1/2}) + G(\mathbf{z}_{k+1/2}) - \widehat{G}(\mathbf{z}_k),$$

which implies by Young's inequalities that

$$\mathbb{E}[\mathrm{res}^2(\mathbf{z}_{k+1/2})] = \mathbb{E}[\min_{\mathbf{u}\in(G+\partial r)\mathbf{z}_{k+1/2}}\|\mathbf{u}\|^2] \leq 3(\eta_k^{-2} + L^2)\|\mathbf{z}_k - \mathbf{z}_{k+1/2}\|^2 + 3\|G(\mathbf{z}_k) - \widehat{G}(\mathbf{z}_k)\|^2$$
$$\leq 3(\eta_k^{-2} + L^2)\|\mathbf{z}_k - \mathbf{z}_{k+1/2}\|^2 + \frac{3}{\bar{b}(k+1)\log^2(k+3)}\left(B^2\|\mathbf{z}_k - \mathbf{z}_0\|^2 + \sigma^2\right).$$

This gives that

$$\frac{1}{K}\sum_{k=0}^{K-1}\mathbb{E}[\mathrm{res}^2(\mathbf{z}_{k+1/2})]$$
$$\leq \frac{1}{K}\sum_{k=0}^{K-1}\left(3(\eta_k^{-2} + L^2)\|\mathbf{z}_k - \mathbf{z}_{k+1/2}\|^2 + \frac{3}{\bar{b}(k+1)\log^2(k+3)}\left(B^2\|\mathbf{z}_k - \mathbf{z}_0\|^2 + \sigma^2\right)\right).$$

We now plug in (B.5) and (B.6) here (after applying Young's inequality) and use $\sum_{k=0}^\infty \frac{1}{(k+1)\log^2(k+3)} < +\infty$ to obtain

$$\frac{1}{K}\sum_{k=0}^{K-1}\mathbb{E}[\mathrm{res}^2(\mathbf{z}_{k+1/2})] = O\left(\frac{1}{K}\right).$$

Hence, the number of iterations $K$ to make the left-hand side less than $\varepsilon$ is of the order $\varepsilon^{-2}$. Moreover, the mini-batch sizes over $\{0, \dots, K-1\}$ are upper bounded by $b_k \leq \bar{b} K \log(K+2)$, giving the stochastic oracle complexity $\widetilde{O}(\varepsilon^{-4})$ for getting

$$\mathbb{E}[\mathrm{res}^2(\mathbf{z}_{\hat{k}+1/2})] \leq \varepsilon^2,$$

where $\hat{k}$ is selected uniformly at random from $\{0, \dots, K-1\}$. Note also that we use Jensen's inequality to conclude the assertion in the theorem statement since $\mathbb{E}[\mathrm{res}(\mathbf{z}_{\hat{k}+1/2})] \leq \sqrt{\mathbb{E}[\mathrm{res}^2(\mathbf{z}_{\hat{k}+1/2})]}$ where $\mathbf{z}^{\mathrm{out}} = \mathbf{z}_{\hat{k}+1/2}$. $\qquad\square$

# C. Proofs for Section 3

In this case, the main departure from the work of (Alacaoglu et al., 2024) is the realization that a bounded variance assumption is not required for their construction to go through. That is, the only place that bounded variance is needed in this paper is for the inner solver used in the MLMC estimator. This inner solver is an operator splitting algorithm applied to a strongly monotone problem. Thanks to strong monotonicity, it is rather straightforward to handle Assumption 3 which replaces the bounded variance assumption, with a small change on the parameter choices. Then we will see that we can use this analysis for the inner solver under Assumption 3 and obtain the same complexity guarantees as (Alacaoglu et al., 2024), which also allows us to handle problems where $\rho$ has the best-known upper bound.

We start with the analysis of the inner solver. It is worth noting that different operator splitting methods are already analyzed under Assumption 3 and strong monotonicity, see for example (Iusem et al., 2017) and (Kotsalis et al., 2022). We provide a proof for FBF under this case for being self-contained and then we show how to use this result in the construction of (Alacaoglu et al., 2024) for getting the final result.

In particular, in the innermost loop of our algorithm, we are solving a strongly monotone inclusion problem. Hence let us write down the abstract problem (and we will see later how to map this back to our original setting)

$$0 \in (A + \mathsf{B})\mathbf{x}^\star, \tag{C.1}$$

where we have access to $\widetilde{B}$ such that $\mathbb{E}[\widetilde{B}(\mathbf{x})] = \mathsf{B}(\mathbf{x})$. For terms such as strong monotonicity or maximal monotonicity, we refer to the textbook (Bauschke & Combettes, 2017). What matters for our purposes is that our subproblem, that is, estimation of $J_{\eta(G+\partial r)}(\mathbf{z}_k)$ satisfies these assumptions.

**Lemma C.1.** *(Detailed restatement of Lemma 3.6) Let $A$ in (C.1) be maximally monotone and $\mathsf{B}$ be $L_{\mathsf{B}}$-Lipschitz and $\mu$-strongly monotone. Assume that $\mathbb{E}\|\widetilde{B}(\mathbf{x}) - \mathsf{B}(\mathbf{x})\|^2 \leq B^2\|\mathbf{x} - \mathbf{x}_0\|^2 + \sigma^2$. Let $\tau_t = \frac{4}{(t+1)\mu + 144M^2/\mu}$. Then, we have for the output of Algorithm 3 that*

$$\mathbb{E}\|\mathbf{x}^\star - \mathbf{x}_T\|^2 \leq \frac{3\kappa\|\mathbf{x}^\star - \mathbf{x}_0\|^2 + 282\sigma^2/\mu^2}{T + \kappa},$$

*where $\kappa = 144M^2/\mu^2$ with $M = \max(L_{\mathsf{B}}, B)$.*

**Remark C.2.** Note that the above lemma is written for an arbitrary problem of finding $\mathbf{x}^\star$ such that $0 \in (A + \mathsf{B})\mathbf{x}^\star$. However, in our particular case, our subproblem (that is, finding the resolvent $J_{\eta(G+\partial r)}(\mathbf{z}_k)$) is finding $\mathbf{x}$ such that $0 \in (\mathrm{Id} + \eta G)\mathbf{x} + \eta \partial r(\mathbf{x}) - \mathbf{z}_k$, hence in the notation of the above statement, we have $\mathbf{x}^\star = J_{\eta(G+\partial r)}(\mathbf{z}_k)$ and then the left-hand side of the statement becomes $\mathbb{E}\|\mathbf{x}_T - J_{\eta(G+\partial r)}(\mathbf{z}_k)\|^2$. The operators are mapped as $A = \eta \partial r$ and $\mathsf{B}(\cdot) = (\mathrm{Id} + \eta G)(\cdot) - \mathbf{z}_k$.

*Proof.* For brevity, let us denote $L = L_{\mathsf{B}}$ in this proof. We proceed as (Alacaoglu et al., 2024, Theorem C.1, until Equation (50)). By using strong monotonicity of $G$ and monotonicity of $\partial r$, one obtains

$$\left(\frac{1}{2\tau_t} + \frac{\mu}{2}\right)\mathbb{E}\|\mathbf{x}^\star - \mathbf{x}_{t+1}\|^2 \leq \frac{1}{2\tau_t}\mathbb{E}\|\mathbf{x}^\star - \mathbf{x}_t\|^2 + \frac{19}{36\tau_t}\mathbb{E}\|\mathbf{x}_{t+1} - \mathbf{x}_{t+1/2}\|^2$$
$$- \frac{1}{2\tau_t}\mathbb{E}\|\mathbf{x}_t - \mathbf{x}_{t+1/2}\|^2, \tag{C.2}$$

since $\tau_t \mu < 1/36$.

We estimate using Young's inequalities, Lipschitzness of $\mathsf{B}$, and Assumption 3 to obtain

$$\mathbb{E}\|\mathbf{x}_{t+1} - \mathbf{x}_{t+1/2}\|^2 \leq \tau_t^2 \|\widetilde{B}(\mathbf{x}_t) - \widetilde{B}(\mathbf{x}_{t+1/2})\|^2$$
$$\leq 3\tau_t^2 \mathbb{E}\left(\|\widetilde{B}(\mathbf{x}_t) - \mathsf{B}(\mathbf{x}_t)\|^2 + \|\widetilde{B}(\mathbf{x}_{t+1/2}) - \mathsf{B}(\mathbf{x}_{t+1/2})\|^2 + \|\mathsf{B}(\mathbf{x}_t) - \mathsf{B}(\mathbf{x}_{t+1/2})\|^2\right)$$
$$\leq 3\tau_t^2 B^2 \mathbb{E}\|\mathbf{x}_t - \mathbf{x}_0\|^2 + 3\tau_t^2 B^2 \mathbb{E}\|\mathbf{x}_{t+1/2} - \mathbf{x}_0\|^2 + 6\tau_t^2\sigma^2 + 3\tau_t^2 L^2 \mathbb{E}\|\mathbf{x}_t - \mathbf{x}_{t+1/2}\|^2$$
$$\leq 6\tau_t^2 B^2 \left(\mathbb{E}\|\mathbf{x}_t - \mathbf{x}^\star\|^2 + \|\mathbf{x}_0 - \mathbf{x}^\star\|^2\right)$$
$$+ 9\tau_t^2 B^2 \left(\mathbb{E}\|\mathbf{x}_t - \mathbf{x}^\star\|^2 + \mathbb{E}\|\mathbf{x}_t - \mathbf{x}_{t+1/2}\|^2 + \|\mathbf{x}_0 - \mathbf{x}^\star\|^2\right)$$
$$+ 3\tau_t^2 L^2 \mathbb{E}\|\mathbf{x}_t - \mathbf{x}_{t+1/2}\|^2 + 6\tau_t^2\sigma^2.$$

We plug this into (C.2), use $\frac{19}{36\tau_t} \times 15\tau_t^2 B^2 < 8\tau_t B^2$, and the notation $M = \max(L, B)$ to get

$$\left(\frac{1}{2\tau_t} + \frac{\mu}{2}\right) \mathbb{E}\|\mathbf{x}^\star - \mathbf{x}_{t+1}\|^2 \leq \left(\frac{1}{2\tau_t} + 8\tau_t B^2\right) \mathbb{E}\|\mathbf{x}^\star - \mathbf{x}_t\|^2 + 8\tau_t B^2 \|\mathbf{x}_0 - \mathbf{x}^\star\|^2$$
$$+ \left(10\tau_t M^2 - \frac{1}{2\tau_t}\right) \mathbb{E}\|\mathbf{x}_t - \mathbf{x}_{t+1/2}\|^2 + 5\tau_t \sigma^2.$$

To simplify, let us assume that $8\tau_t B^2 \leq \frac{\mu}{4}$ and define

$$\frac{1}{2\gamma_t} = \frac{1}{2\tau_t} + \frac{\mu}{4}. \tag{C.3}$$

Then, we can equivalently write the previous recursion as

$$\left(\frac{1}{2\gamma_t} + \frac{\mu}{4}\right) \|\mathbf{x}^\star - \mathbf{x}_{t+1}\|^2 \leq \frac{1}{2\gamma_t} \|\mathbf{x}^\star - \mathbf{x}_t\|^2 + 8\eta_t B^2 \|\mathbf{x}_0 - \mathbf{x}^\star\|^2$$
$$+ \left(10\tau_t M^2 - \frac{1}{2\eta_t}\right) \|\mathbf{x}_t - \mathbf{x}_{t+1/2}\|^2 + 5\tau_t \sigma^2. \tag{C.4}$$

Hence, the main recursion we have is very similar to (Alacaoglu et al., 2024, Eq. (51)).

Assume now

$$10\tau_t M^2 \leq \frac{1}{2\tau_t} \iff 20\tau_t^2 M^2 \leq 1.$$

Let us now set

$$\gamma_t = \frac{4}{(t+3)\mu + 144M^2/\mu}, \ \frac{1}{2\gamma_t} = \frac{(t+3)\mu + 144M^2/\mu}{8} \ \text{and} \ \frac{1}{2\gamma_t} + \frac{\mu}{4} = \frac{(t+5)\mu + 144M^2/\mu}{8}.$$

Let us note that this also gives the following for $\tau_t$:

$$\frac{1}{2\tau_t} = \frac{(t+1)\mu + 144M^2/\mu}{8} \iff \tau_t = \frac{4}{(t+1)\mu + 144M^2/\mu}.$$

Notice how both of the following requirements are satisfied with this choice of $\tau_t$:

$$20\tau_t^2 M^2 \leq 1 \quad \text{and} \quad 8\tau_t B^2 \leq \frac{\mu}{4}.$$

Next, multiply (C.4) by $\left(\frac{1}{2\gamma_t} + \mu/4\right)^{-1} = \frac{8}{(t+5)\mu + 144M^2/\mu}$ and get

$$\|\mathbf{x}^\star - \mathbf{x}_{t+1}\|^2 \leq \frac{(t+3)\mu + 144M^2/\mu}{(t+5)\mu + 144M^2/\mu} \|\mathbf{x}^\star - \mathbf{x}_t\|^2 + \frac{64\eta_t M^2}{(t+5)\mu + 144M^2/\mu} \|\mathbf{x}_0 - \mathbf{x}^\star\|^2$$
$$+ \frac{64}{(t+5)\mu + 144M^2/\mu} \eta_t \sigma^2.$$

Plugging in $\tau_t$ gives

$$\|\mathbf{x}^\star - \mathbf{x}_{t+1}\|^2 \leq \frac{(t+3)\mu + 144M^2/\mu}{(t+5)\mu + 144M^2/\mu} \|\mathbf{x}^\star - \mathbf{x}_t\|^2$$
$$+ \frac{256M^2}{((t+5)\mu + 144M^2/\mu)((t+1)\mu + 144M^2/\mu)} \|\mathbf{x}_0 - \mathbf{x}^\star\|^2$$
$$+ \frac{256}{((t+5)\mu + 144M^2/\mu)((t+1)\mu + 144M^2/\mu)} \sigma^2.$$

An equivalent way to write this, by letting $\kappa = 144M^2/\mu^2$ is

$$\|\mathbf{x}^\star - \mathbf{x}_{t+1}\|^2 \leq \frac{(t+3) + \kappa}{(t+5) + \kappa} \|\mathbf{x}^\star - \mathbf{x}_t\|^2$$
$$+ \frac{2\kappa}{((t+5) + \kappa)((t+1) + \kappa)} \|\mathbf{x}_0 - \mathbf{x}^\star\|^2$$
$$+ \frac{256\sigma^2/\mu^2}{((t+5) + \kappa)((t+1) + \kappa)}.$$

We now prove by induction that

$$\mathbb{E}\|\mathbf{x}^\star - \mathbf{x}_t\|^2 \leq \frac{a\kappa\|\mathbf{x}^\star - \mathbf{x}_0\|^2 + bG^2/\mu^2}{t + \kappa},$$

with $a = 3$ and $b = 282$.

The base case $t = 0$ holds trivially when $a \geq 1$. We want to show that

$$\|\mathbf{x}^\star - \mathbf{x}_{T+1}\|^2 \leq \frac{T+3+\kappa}{T+5+\kappa} \frac{a\kappa\|\mathbf{x}^\star - \mathbf{x}_0\|^2 + b\sigma^2/\mu^2}{T+\kappa} + \frac{2\kappa}{(T+5+\kappa)(T+1+\kappa)} \|\mathbf{x}^\star - \mathbf{x}_0\|^2$$
$$+ \frac{256\sigma^2/\mu^2}{(T+5+\kappa)(T+1+\kappa)}.$$

For some constant $d > 1$, let us have $\frac{2}{a} \leq \frac{1}{d}$ and $\frac{256}{b} \leq \frac{1}{d}$, and then we have

$$\|\mathbf{x}^\star - \mathbf{x}_{T+1}\|^2 \leq \frac{1}{T+5+\kappa} \left( \frac{T+3+\kappa}{T+\kappa} + \frac{1}{d(T+1+\kappa)} \right) \left( a\kappa\|\mathbf{x}^\star - \mathbf{x}_0\|^2 + b\sigma^2/\mu^2 \right)$$

and then we wish to find $d$ such that (after setting $A = T + \kappa = T + 144M^2/\mu^2$ and where $A \geq 144$)

$$\frac{A+3}{A(A+5)} + \frac{1}{d(A+1)(A+5)} \leq \frac{1}{A+1} \iff d(A+1)(A+3) + A \leq dA(A+5)$$
$$\iff 4Ad + 3d + A \leq 5Ad \iff 3d + A \leq dA \iff 3d \leq (d-1)A,$$

which is satisfied when $d = 1.1$ since this would require $3.3 \leq 0.1A \iff 33 \leq A$ which is true because $A = T + \kappa \geq \kappa \geq 144$. We then use the bounds for $a, b$ as

$$a \geq 2.2 \quad \text{and} \quad b \geq 256 \times 1.1$$

The proof is completed. $\qquad\qquad\qquad\qquad\qquad\qquad\qquad\qquad\qquad\qquad\qquad\qquad\qquad\qquad$ $\square$

## C.1. Results for the MLMC estimator

We now continue with the results related to the MLMC estimator, taken from (Alacaoglu et al., 2024). The only difference will be the change in the complexity analysis inner solver used for estimating the resolvent (that is, the proximal subproblem). As a result, the only change in the lemmas we cite below are the constants.

For brevity, let us denote the upper bound in Lemma C.1 as

$$C = 3\kappa\|\mathbf{x}^\star - \mathbf{x}_0\|^2 + 282\sigma^2/\mu^2, \quad C_1 = 3\kappa, \quad C_2 = 282/\mu^2. \tag{C.5}$$

where $\kappa = 144M^2/\mu^2$. The lemmas below have identical proofs to the corresponding results we cite from (Alacaoglu et al., 2024) with the minor differences of having the constants in Lemma C.1 instead of (Alacaoglu et al., 2024, Theorem C.1). Hence, we do not repeat their proofs and just refer to (Alacaoglu et al., 2024).

**Lemma C.3.** *(Alacaoglu et al., 2024, Lemma C.9) Under the setting of Lemma C.1, we have, for the output of Algorithm 2, that*

$$\|\mathbb{E}[\mathbf{y}^{out}] - \mathbf{y}^\star\|^2 \leq \frac{2C}{N},$$
$$\mathbb{E}\|\mathbf{y}^{out} - \mathbf{y}^\star\|^2 \leq 14C \log_2 N.$$

*where $C$ is as defined in (C.5) and the number of calls to $\widetilde{B}$ is $O(\log_2 N)$.*

**Lemma C.4.** *(Alacaoglu et al., 2024, Corollary C.10) Let $\widetilde{J}_{\eta(G+\partial r)}(\mathbf{z}_k)$ be as defined in Algorithm 1. Under the setting of Lemma C.1, we have for any $b_k, v$ that*

$$\|\mathbb{E}[\widetilde{J}_{\eta(G+\partial r)}(\mathbf{z}_k)] - J_{\eta(G+\partial r)}(\mathbf{z}_k)\|^2 \le b_k^2(\|(\mathrm{Id} - J_{\eta(G+\partial r)})(\mathbf{z}_k)\|^2 + \sigma^2),$$

$$\mathbb{E}\|\widetilde{J}_{\eta(G+\partial r)}(\mathbf{z}_k) - J_{\eta(G+\partial r)}(\mathbf{z}_k)\|^2 \le v^2(\|(\mathrm{Id} - J_{\eta(G+\partial r)})(\mathbf{z}_k)\|^2 + \sigma^2),$$

*where*

$$N_k = \left\lceil \frac{\max\{2C_1, 2C_2\}}{\min\{b_k^2, v^2/2\}} \right\rceil, \text{ and } M_k = \left\lceil \frac{28\max\{C_1, C_2\}\log_2 N_k}{v^2} \right\rceil \tag{C.6}$$

*where $C_1, C_2$ are as defined in (C.5). The number of calls to the stochastic first-order oracle at each iteration is $O((\log N_k) \cdot M_k)$ in expectation.*

Finally, the complexity analysis will be the combination of these three results. Most of the derivation is the same as (Alacaoglu et al., 2024) up to the change of the inner solver, as a result, we only sketch the differences in the proof, compared to (Alacaoglu et al., 2024, Theorem C.11).

**Theorem C.5.** *(Detailed restatement of Theorem 3.1) Let Assumptions 1 and 3 hold and suppose that $\rho < 1/L$. Then, for Algorithm 1 with $\eta \le \frac{1}{L}$ and $\alpha = 1 - \rho/\eta$, $\alpha_k = \frac{\alpha}{\sqrt{k+2}\log(k+3)}$; and $N_k, M_k$ given in (C.6) with $b_k^2 = \frac{\alpha_k}{120\alpha(k+1)}, v^2 = \frac{1}{60}$, we have*

$$\mathbb{E}\|\mathbf{z}_{\hat{k}} - J_{\eta(G+\partial r)}(\mathbf{z}_{\hat{k}})\| \le \varepsilon \text{ with expected sfo complexity } \widetilde{O}(\varepsilon^{-4}),$$

*where $\mathbf{z}_{\hat{k}}$ is selected uniformly at random after running the algorithm for $K$ iterations.*

*Proof.* The proof of this theorem mirrors that of (Alacaoglu et al., 2024, Theorem C.11) which is in fact oblivious to the inner solver until the last paragraph of the proof, particularly the calculation of the stochastic oracle complexity of each iteration. In our case, $N_k, M_k$ depend on $C_1, C_2$ which are slightly different than (Alacaoglu et al., 2024), due to us using Lemma C.1 which requires only Assumption 3, unlike the result of (Alacaoglu et al., 2024) that used the bounded variance assumption. However, this only changes the absolute constants for the number of expected calls to the stochastic oracle and the complexity result is hence the same. □

## D. Proofs for Section 4

Let us provide a further intuition for Algorithm 4. Note that by the definition of $\mathbf{z}_{k+1/2}$, we have

$$\Longleftrightarrow 0 \in \partial r(\mathbf{z}_{k+1/2}) + \frac{1}{\gamma_k}\left(\mathbf{z}_{k+1/2} - \bar{\mathbf{z}}_k + \gamma_k\mathbf{g}_k\right)$$

$$\Longleftrightarrow G(\mathbf{z}_{k+1/2}) + \partial r(\mathbf{z}_{k+1/2}) \ni \frac{1}{\gamma_k}\left(\bar{\mathbf{z}}_k - \mathbf{z}_{k+1/2} - \gamma_k\mathbf{g}_k + \gamma_k G(\mathbf{z}_{k+1/2})\right).$$

Hence, the second step, that is, the update of $\mathbf{z}_{k+1}$ is a subgradient descent-like step by using the unbiased estimate of a particular *subgradient* (indeed for a min-max problem, $G$ contains the gradients of the coupling function) from $(G + \partial r)\mathbf{z}_{k+1/2}$ with step size $\tau_k$ (which absorbs the $1/\gamma_k$ appearing above). In this update, we move from the anchored iterate $\bar{\mathbf{z}}_k$ to follow the idea of Halpern anchoring.

We now analyze one iteration of this method.

**Lemma D.1.** *Let Assumptions 1, 2, and 3 hold. Then we have, for Algorithm 4, that*

$$\mathbb{E}\|\mathbf{z}^\star - \mathbf{z}_{k+1}\|^2 \le (1 - \beta_k)\mathbb{E}\|\mathbf{z}^\star - \mathbf{z}_k\|^2 + \beta_k\|\mathbf{z}^\star - \mathbf{z}_0\|^2 - \frac{\beta_k(1 - \tau_k)}{1 + \tau_k}\mathbb{E}\|\mathbf{z}_0 - \mathbf{z}_{k+1}\|^2$$
$$+ \mathcal{C}_{1,k}\mathbb{E}\|\mathbf{z}_k - \mathbf{z}_{k+1/2}\|^2 + \mathcal{C}_{2,k}\mathbb{E}\|\mathbf{z}_k - \mathbf{z}_{k+1}\|^2$$
$$+ \mathcal{C}_{3,k}\mathbb{E}\|\mathbf{z}_0 - \mathbf{z}_{k+1/2}\|^2 + \mathcal{C}_{4,k}\mathbb{E}\|\mathbf{g}_k - G(\mathbf{z}_k)\|^2 + \mathcal{E}_k. \tag{D.1}$$

*where we define*

$$\mathcal{E}_k = \frac{72\tau_k^2\gamma_k^2\sigma^2}{1 + \tau_k},$$
$$\mathcal{C}_{1,k} = \frac{\tau_k(1/4 + (9/2)\gamma_k^2L^2 + 72\tau_k\gamma_k^2L^2 - 2(1 - \beta_k))}{1 + \tau_k} + \frac{5\rho\tau_k(1 - \beta_k)}{2\gamma_k} + 11\rho\tau_kL^2\gamma_k,$$
$$\mathcal{C}_{2,k} = \frac{1}{24(1 + \tau_k)} - \frac{(1 - \tau_k)(1 - \beta_k)}{1 + \tau_k}, \tag{D.2}$$
$$\mathcal{C}_{3,k} = \frac{5\rho\tau_k\beta_k}{2\gamma_k} + \frac{2\tau_k(36\tau_k\gamma_k^2B^2 - \beta_k)}{1 + \tau_k},$$
$$\mathcal{C}_{4,k} = 110\rho\tau_k\gamma_k + \frac{4\tau_k\gamma_k^2(9 + 18\tau_k)}{1 + \tau_k}.$$

**Remark D.2.** The bound on the right-hand side of (D.1) is rather complicated. The main intuition is that for each of the error terms independent of $\mathbf{z}^\star$, we have negative coefficients that we can use to cancel the positive coefficients (after picking the parameters accordingly), except the last terms on the right-hand side of (D.1). For the second term of the last line, we will use the classical bound of STORM variance reduced estimator of (Cutkosky & Orabona, 2019). The last term in the last line will be sufficiently small due to the choice of $\tau_k^2$.

*Proof of Lemma D.1.* The definitions of $\mathbf{z}_{k+1/2}$ and the proximal operator give

$$\mathbf{z}_{k+1/2} + \gamma_k\partial r(\mathbf{z}_{k+1/2}) \ni \bar{\mathbf{z}}_k - \gamma_k\mathbf{g}_k$$
$$\Longleftrightarrow G(\mathbf{z}_{k+1/2}) + \partial r(\mathbf{z}_{k+1/2}) \ni \frac{1}{\gamma_k}\left(\bar{\mathbf{z}}_k - \mathbf{z}_{k+1/2}\right) - \mathbf{g}_k + G(\mathbf{z}_{k+1/2}).$$

Then, using (wMVI) gives

$$\left\langle \frac{1}{\gamma_k}(\bar{\mathbf{z}}_k - \mathbf{z}_{k+1/2}) - \mathbf{g}_k + G(\mathbf{z}_{k+1/2}), \mathbf{z}_{k+1/2} - \mathbf{z}^\star \right\rangle$$
$$\ge -\rho\left\|\frac{1}{\gamma_k}(\bar{\mathbf{z}}_k - \mathbf{z}_{k+1/2}) - \mathbf{g}_k + G(\mathbf{z}_{k+1/2})\right\|^2. \tag{D.3}$$

After multiplying both sides by $\tau_k \gamma_k$, we get

$$\tau_k \langle \bar{\mathbf{z}}_k - \mathbf{z}_{k+1/2} - \gamma_k \mathbf{g}_k, \mathbf{z}_{k+1/2} - \mathbf{z}^\star \rangle + \tau_k \gamma_k \langle G(\mathbf{z}_{k+1/2}), \mathbf{z}_{k+1/2} - \mathbf{z}^\star \rangle$$
$$\geq -\frac{\rho \tau_k}{\gamma_k} \| \bar{\mathbf{z}}_k - \mathbf{z}_{k+1/2} - \gamma_k \mathbf{g}_k + \gamma_k G(\mathbf{z}_{k+1/2}) \|^2. \tag{D.4}$$

We estimate the first inner product. The definition of $\mathbf{z}_{k+1}$, after rearranging and dividing each side by $\tau_k$ yields

$$\mathbf{z}_{k+1/2} - \bar{\mathbf{z}}_k = \frac{1}{\tau_k} \left( \mathbf{z}_{k+1} - \bar{\mathbf{z}}_k \right) - \gamma_k \mathbf{g}_k + \gamma_k \widetilde{G}(\mathbf{z}_{k+1/2}, \xi_{k+1/2}).$$

As a result, we have for the first inner product in (D.4) that

$$\tau_k \langle \mathbf{z}_{k+1/2} - \bar{\mathbf{z}}_k + \gamma_k \mathbf{g}_k, \mathbf{z}^\star - \mathbf{z}_{k+1/2} \rangle = \langle \mathbf{z}_{k+1} - \bar{\mathbf{z}}_k + \tau_k \gamma_k \widetilde{G}(\mathbf{z}_{k+1/2}, \xi_{k+1/2}), \mathbf{z}^\star - \mathbf{z}_{k+1/2} \rangle.$$

This gives in (D.4) that

$$\langle \mathbf{z}_{k+1} - \bar{\mathbf{z}}_k, \mathbf{z}^\star - \mathbf{z}_{k+1/2} \rangle + \tau_k \gamma_k \langle G(\mathbf{z}_{k+1/2}) - \widetilde{G}(\mathbf{z}_{k+1/2}, \xi_{k+1/2}), \mathbf{z}_{k+1/2} - \mathbf{z}^\star \rangle$$
$$\geq -\frac{\rho \tau_k}{\gamma_k} \| \bar{\mathbf{z}}_k - \mathbf{z}_{k+1/2} - \gamma_k \mathbf{g}_k + \gamma_k G(\mathbf{z}_{k+1/2}) \|^2.$$

After taking expectation, using the tower property, and the assumption of $\widetilde{G}(\mathbf{z}_{k+1/2}, \xi_{k+1/2})$ being unbiased, we obtain

$$\mathbb{E} \langle \mathbf{z}_{k+1} - \bar{\mathbf{z}}_k, \mathbf{z}^\star - \mathbf{z}_{k+1/2} \rangle \geq -\frac{\rho \tau_k}{\gamma_k} \mathbb{E} \| \bar{\mathbf{z}}_k - \mathbf{z}_{k+1/2} - \gamma_k \mathbf{g}_k + \gamma_k G(\mathbf{z}_{k+1/2}) \|^2. \tag{D.5}$$

We rewrite the inner product using squared norms to derive

$$2 \langle \mathbf{z}_{k+1} - \bar{\mathbf{z}}_k, \mathbf{z}^\star - \mathbf{z}_{k+1/2} \rangle = 2 \langle \mathbf{z}_{k+1} - \mathbf{z}_{k+1/2}, \mathbf{z}^\star - \mathbf{z}_{k+1/2} \rangle + 2 \langle \mathbf{z}_{k+1/2} - \bar{\mathbf{z}}_k, \mathbf{z}^\star - \mathbf{z}_{k+1/2} \rangle$$
$$= \| \mathbf{z}_{k+1} - \mathbf{z}_{k+1/2} \|^2 - \| \mathbf{z}^\star - \mathbf{z}_{k+1} \|^2 - \| \mathbf{z}_{k+1/2} - \bar{\mathbf{z}}_k \|^2 + \| \mathbf{z}^\star - \bar{\mathbf{z}}_k \|^2. \tag{D.6}$$

We continue to estimate the main error term $\| \mathbf{z}_{k+1} - \mathbf{z}_{k+1/2} \|^2$ as

$$\| \mathbf{z}_{k+1} - \mathbf{z}_{k+1/2} \|^2$$
$$= \langle \mathbf{z}_{k+1} - \mathbf{z}_{k+1/2}, \mathbf{z}_{k+1} - \mathbf{z}_{k+1/2} \rangle$$
$$= (1 - \tau_k) \langle \bar{\mathbf{z}}_k - \mathbf{z}_{k+1/2}, \mathbf{z}_{k+1} - \mathbf{z}_{k+1/2} \rangle + \tau_k \gamma_k \langle \mathbf{g}_k - \widetilde{G}(\mathbf{z}_{k+1/2}, \xi_{k+1/2}), \mathbf{z}_{k+1} - \mathbf{z}_{k+1/2} \rangle$$
$$= \frac{1 - \tau_k}{2} \left( \| \bar{\mathbf{z}}_k - \mathbf{z}_{k+1/2} \|^2 + \| \mathbf{z}_{k+1} - \mathbf{z}_{k+1/2} \|^2 - \| \mathbf{z}_{k+1} - \bar{\mathbf{z}}_k \|^2 \right)$$
$$+ \tau_k \gamma_k \langle \mathbf{g}_k - \widetilde{G}(\mathbf{z}_{k+1/2}, \xi_{k+1/2}), \mathbf{z}_{k+1} - \mathbf{z}_{k+1/2} \rangle, \tag{D.7}$$

where the second equality used the definition of $\mathbf{z}_{k+1}$ as

$$\mathbf{z}_{k+1} - \mathbf{z}_{k+1/2} = \bar{\mathbf{z}}_k - \mathbf{z}_{k+1/2} - \tau_k (\bar{\mathbf{z}}_k - \mathbf{z}_{k+1/2} - \gamma_k \mathbf{g}_k + \gamma_k \widetilde{G}(\mathbf{z}_{k+1/2}, \xi_{k+1/2}))$$
$$= (1 - \tau_k)(\bar{\mathbf{z}}_k - \mathbf{z}_{k+1/2}) + \tau_k \gamma_k (\mathbf{g}_k - \widetilde{G}(\mathbf{z}_{k+1/2}, \xi_{k+1/2})).$$

The last equality in (D.7) used the elementary identity $\langle \mathbf{a}, \mathbf{b} \rangle = \frac{1}{2} \left( \|\mathbf{a}\|^2 + \|\mathbf{b}\|^2 - \|\mathbf{a} - \mathbf{b}\|^2 \right)$.

Rearranging (D.7) leads to

$$\frac{1 + \tau_k}{2} \| \mathbf{z}_{k+1} - \mathbf{z}_{k+1/2} \|^2$$
$$= \frac{1 - \tau_k}{2} \left( \| \bar{\mathbf{z}}_k - \mathbf{z}_{k+1/2} \|^2 - \| \mathbf{z}_{k+1} - \bar{\mathbf{z}}_k \|^2 \right) + \tau_k \gamma_k \langle \mathbf{g}_k - \widetilde{G}(\mathbf{z}_{k+1/2}, \xi_{k+1/2}), \mathbf{z}_{k+1} - \mathbf{z}_{k+1/2} \rangle.$$

Multiplying both sides of $\frac{2}{1 + \tau_k}$ gives us

$$\| \mathbf{z}_{k+1} - \mathbf{z}_{k+1/2} \|^2 = \frac{1 - \tau_k}{1 + \tau_k} \left( \| \bar{\mathbf{z}}_k - \mathbf{z}_{k+1/2} \|^2 - \| \mathbf{z}_{k+1} - \bar{\mathbf{z}}_k \|^2 \right)$$
$$+ \frac{2 \tau_k \gamma_k}{1 + \tau_k} \langle \mathbf{g}_k - \widetilde{G}(\mathbf{z}_{k+1/2}, \xi_{k+1/2}), \mathbf{z}_{k+1} - \mathbf{z}_{k+1/2} \rangle.$$

With this, we estimate (D.6) as

$$
\begin{aligned}
2\langle \mathbf{z}_{k+1} - \bar{\mathbf{z}}_k, \mathbf{z}^\star - \mathbf{z}_{k+1/2}\rangle \\
= \frac{-2\tau_k}{1+\tau_k}\|\bar{\mathbf{z}}_k - \mathbf{z}_{k+1/2}\|^2 - \frac{1-\tau_k}{1+\tau_k}\|\mathbf{z}_{k+1} - \bar{\mathbf{z}}_k\|^2 - \|\mathbf{z}^\star - \mathbf{z}_{k+1}\|^2 + \|\mathbf{z}^\star - \bar{\mathbf{z}}_k\|^2 \\
+ \frac{2\tau_k\gamma_k}{1+\tau_k}\langle \mathbf{g}_k - \widetilde{G}(\mathbf{z}_{k+1/2}, \xi_{k+1/2}), \mathbf{z}_{k+1} - \mathbf{z}_{k+1/2}\rangle.
\end{aligned}
\tag{D.8}
$$

Next, we use the definition of $\bar{\mathbf{z}}_k$ for estimating the first, second and fourth terms on the right-hand side of (D.8), to derive

$$
\begin{aligned}
2\langle \mathbf{z}_{k+1} - \bar{\mathbf{z}}_k, \mathbf{z}^\star - \mathbf{z}_{k+1/2}\rangle = {} & \beta_k\|\mathbf{z}^\star - \mathbf{z}_0\|^2 + (1-\beta_k)\|\mathbf{z}^\star - \mathbf{z}_k\|^2 - \|\mathbf{z}^\star - \mathbf{z}_{k+1}\|^2 \\
& - \frac{2\tau_k}{1+\tau_k}\left(\beta_k\|\mathbf{z}_0 - \mathbf{z}_{k+1/2}\|^2 + (1-\beta_k)\|\mathbf{z}_k - \mathbf{z}_{k+1/2}\|^2\right) \\
& - \frac{\beta_k(1-\tau_k)}{1+\tau_k}\|\mathbf{z}_0 - \mathbf{z}_{k+1}\|^2 - \frac{(1-\beta_k)(1-\tau_k)}{1+\tau_k}\|\mathbf{z}_k - \mathbf{z}_{k+1}\|^2 \\
& + \frac{2\tau_k\gamma_k}{1+\tau_k}\langle \mathbf{g}_k - \widetilde{G}(\mathbf{z}_{k+1/2}, \xi_{k+1/2}), \mathbf{z}_{k+1} - \mathbf{z}_{k+1/2}\rangle,
\end{aligned}
\tag{D.9}
$$

by using the identity $\|\beta\mathbf{a} + (1-\beta)\mathbf{b}\|^2 = \beta\|\mathbf{a}\|^2 + (1-\beta)\|\mathbf{b}\|^2 - \beta(1-\beta)\|\mathbf{a}-\mathbf{b}\|^2$ three times for all the terms involving $\bar{\mathbf{z}}_k$ in (D.8).

Same as the derivation of (Alacaoglu et al., 2025, Eq. (4.27)), one can estimate by Young's inequality that

$$
\begin{aligned}
\mathbb{E}&\left[\frac{2\tau_k\gamma_k}{1+\tau_k}\langle \mathbf{g}_k - \widetilde{G}(\mathbf{z}_{k+1/2}, \xi_{k+1/2}), \mathbf{z}_{k+1} - \mathbf{z}_{k+1/2}\rangle\right] \\
& \leq \frac{\tau_k}{(1+\tau_k)c_1}\mathbb{E}\|\mathbf{z}_k - \mathbf{z}_{k+1/2}\|^2 + \frac{\tau_k\gamma_k^2 c_1}{1+\tau_k}\mathbb{E}\|\mathbf{g}_k - G(\mathbf{z}_{k+1/2})\|^2 \\
& \quad + \frac{1}{(1+\tau_k)c_2}\mathbb{E}\|\mathbf{z}_{k+1} - \mathbf{z}_k\|^2 + \frac{\tau_k^2\gamma_k^2 c_2}{(1+\tau_k)}\mathbb{E}\|\mathbf{g}_k - \widetilde{G}(\mathbf{z}_{k+1/2}, \xi_{k+1/2})\|^2,
\end{aligned}
\tag{D.10}
$$

where we used

$$
\begin{aligned}
\mathbb{E}_{k+1/2}&\langle \mathbf{g}_k - \widetilde{G}(\mathbf{z}_{k+1/2}, \xi_{k+1/2}), \mathbf{z}_{k+1} - \mathbf{z}_{k+1/2}\rangle \\
&= \mathbb{E}_{k+1/2}[\langle \mathbf{g}_k - \widetilde{G}(\mathbf{z}_{k+1/2}, \xi_{k+1/2}), \mathbf{z}_k - \mathbf{z}_{k+1/2}\rangle + \langle \mathbf{g}_k - \widetilde{G}(\mathbf{z}_{k+1/2}, \xi_{k+1/2}), \mathbf{z}_{k+1} - \mathbf{z}_k\rangle] \\
&= \langle \mathbf{g}_k - G(\mathbf{z}_{k+1/2}), \mathbf{z}_k - \mathbf{z}_{k+1/2}\rangle + \mathbb{E}_{k+1/2}\langle \mathbf{g}_k - \widetilde{G}(\mathbf{z}_{k+1/2}, \xi_{k+1/2}), \mathbf{z}_{k+1} - \mathbf{z}_k\rangle].
\end{aligned}
$$

On (D.10), after Young's inequality and Lipschitzness of $G$, we estimate as

$$
\begin{aligned}
\mathbb{E}&\left[\frac{2\tau_k\gamma_k}{1+\tau_k}\langle \mathbf{g}_k - \widetilde{G}(\mathbf{z}_{k+1/2}, \xi_{k+1/2}), \mathbf{z}_{k+1} - \mathbf{z}_{k+1/2}\rangle\right] \\
& \leq \frac{\tau_k}{(1+\tau_k)c_1}\mathbb{E}\|\mathbf{z}_k - \mathbf{z}_{k+1/2}\|^2 + \frac{A_1 L^2 \tau_k\gamma_k^2 c_1}{1+\tau_k}\mathbb{E}\|\mathbf{z}_k - \mathbf{z}_{k+1/2}\|^2 + \frac{A_2\tau_k\gamma_k^2 c_1}{1+\tau_k}\mathbb{E}\|\mathbf{g}_k - G(\mathbf{z}_k)\|^2 \\
& \quad + \frac{1}{(1+\tau_k)c_2}\mathbb{E}\|\mathbf{z}_{k+1} - \mathbf{z}_k\|^2 + \frac{3\tau_k^2\gamma_k^2 c_2}{(1+\tau_k)}\mathbb{E}\|\mathbf{g}_k - G(\mathbf{z}_k)\|^2 \\
& \quad + \frac{3L^2\tau_k^2\gamma_k^2 c_2}{(1+\tau_k)}\mathbb{E}\|\mathbf{z}_k - \mathbf{z}_{k+1/2}\|^2 + \frac{3\tau_k^2\gamma_k^2 c_2}{(1+\tau_k)}\mathbb{E}\|G(\mathbf{z}_{k+1/2}) - \widetilde{G}(\mathbf{z}_{k+1/2}, \xi_{k+1/2})\|^2.
\end{aligned}
\tag{D.11}
$$

where $A_1 = (1 + a_1)$, $A_2 = (1 + \frac{1}{a_1})$ for any positive $a_1$.

Applying Assumption 3 results in the estimate

$$
\mathbb{E}\|G(\mathbf{z}_{k+1/2}) - \widetilde{G}(\mathbf{z}_{k+1/2}, \xi_{k+1/2})\|^2 \leq B^2\mathbb{E}\|\mathbf{z}_{k+1/2} - \mathbf{z}_0\|^2 + \sigma^2.
\tag{D.12}
$$

We also have by Young's inequalities that

$$
\begin{aligned}
\frac{\rho\tau_k}{\gamma_k}&\|\bar{\mathbf{z}}_k - \mathbf{z}_{k+1/2} - \gamma_k\mathbf{g}_k + \gamma_k G(\mathbf{z}_{k+1/2})\|^2 \\
&\leq \left(\frac{E_1\rho\tau_k(1-\beta_k)}{\gamma_k} + E_3 L^2\rho\tau_k\gamma_k\right)\|\mathbf{z}_k - \mathbf{z}_{k+1/2}\|^2 \\
&\quad + \frac{E_1\rho\tau_k\beta_k}{\gamma_k}\|\mathbf{z}_0 - \mathbf{z}_{k+1/2}\|^2 + E_2\rho\tau_k\gamma_k\|\mathbf{g}_k - G(\mathbf{z}_k)\|^2,
\end{aligned}
\tag{D.13}
$$

where we use the property

$$
\begin{aligned}
\|\bar{\mathbf{z}}_k &- \mathbf{z}_{k+1/2}\|^2 \\
&= \|\beta_k\mathbf{z}_0 + (1-\beta_k)\mathbf{z}_k - \mathbf{z}_{k+1/2}\|^2 \leq \beta_k\|\mathbf{z}_0 - \mathbf{z}_{k+1/2}\|^2 + (1-\beta_k)\|\mathbf{z}_k - \mathbf{z}_{k+1/2}\|^2.
\end{aligned}
$$

On (D.13), we have $E_1 = (1 + e_1)$, $E_2 = (1 + \frac{1}{e_1})(1 + e_2)$, and $E_3 = (1 + \frac{1}{e_1})(1 + \frac{1}{e_2})$ for any positive $e_1, e_2$.

Finally, we set

$$
c_1 = 4, \quad c_2 = 24, \quad a_1 = \frac{1}{8}, \quad e_1 = \frac{1}{4}, \quad e_2 = 10.
$$

With these choices, we have

$$
A_1 = \frac{9}{8}, \quad A_2 = 9, \quad E_1 = \frac{5}{4}, \quad E_2 = 55, \quad E_3 = \frac{11}{2}.
$$

Next, we multiply both sides of (D.5) by 2, then plug in the expectation of (D.9), (D.11), (D.12), and (D.13) to get the result. $\qquad\square$

We now continue with the recursion of the STORM estimator of (Cutkosky & Orabona, 2019) that is used to update $\mathbf{g}_k$. This lemma is taken from (Alacaoglu et al., 2025, Lemma 6.1) (which we refer to for the proof of this precise statement) that followed the idea of (Cutkosky & Orabona, 2019) with the minor change of using the Assumption 3 rather than the uniformly bounded variance.

**Lemma D.3.** *(See (Cutkosky & Orabona, 2019) for the original idea and (Alacaoglu et al., 2025, Lemma 6.1) for the statement used here) Let $\mathbf{g}_k$ be as defined in Algorithm 4. Then, under Assumptions 1 and 3, we have*

$$
\begin{aligned}
\frac{\alpha_k}{2}\mathbb{E}\|\mathbf{g}_k - G(\mathbf{z}_k)\|^2 \leq &\left(1 - \frac{\alpha_k}{2}\right)\mathbb{E}\|\mathbf{g}_k - G(\mathbf{z}_k)\|^2 - \mathbb{E}\|\mathbf{g}_{k+1} - G(\mathbf{z}_{k+1})\|^2 \\
&+ 2L^2\mathbb{E}\|\mathbf{z}_{k+1} - \mathbf{z}_k\|^2 + 2\alpha_k^2(B^2\mathbb{E}\|\mathbf{z}_k - \mathbf{z}_0\|^2 + \sigma^2).
\end{aligned}
$$

We now include the restatement of Theorem 4.1 that includes the parameter details and then provide its proof.

**Theorem D.4.** *(Detailed restatement of Theorem 4.1) Let Assumptions 1, 2, 3 hold and suppose that $\rho > 0$ is sufficiently small with*

$$
\rho \leq \min\left\{\frac{L}{55}\left(\frac{83}{24L^2} - \frac{9\bar{\tau}}{\sqrt{3}L^2}\right), \quad \frac{1}{12L}, \quad \frac{1}{16L(1+\bar{\tau})} - \frac{2\bar{\tau}}{17\sqrt{3}L}\left(\frac{9B^2}{L^2} + 3\right)\right\} =: f(\bar{\tau}).
\tag{D.14}
$$

*Then, for the output of Algorithm 4 with*

$$
\beta_k = \frac{1}{k+3}, \quad \alpha_k = \frac{2}{\sqrt{k+3}}, \quad \gamma_k = \frac{1}{4L}, \quad \tau_k = \frac{\bar{\tau}}{\sqrt{k+3}},
$$

*where $\bar{\tau} \leq \min\left\{\frac{L^2}{219B^2}, \frac{L^2}{50(3B^2+L^2)}\right\}$. Then, we have*

$$
\mathbb{E}[\mathrm{res}(\mathbf{z}^{out})] \leq \varepsilon \text{ with stochastic oracle complexity } \widetilde{O}(\varepsilon^{-4}),
$$

*where $\Pr(\hat{k} = k) = \frac{\tau_k(k+3)}{\sum_{i=0}^{K-1}\tau_i(i+3)}$ and $\mathbf{z}^{out} = \mathbf{z}_{\hat{k}+1/2}$.*

**Remark D.5.** As the result in Section 2, the range of $\rho$ is quite restrictive. In particular, as $\bar{\tau} \to 0$ the dominant term in the upper bound, $\frac{1}{16L(1+\bar{\tau})} - \frac{2\bar{\tau}}{17\sqrt{3}L}\left(\frac{9B^2}{L^2} + 3\right)$, becomes $\approx \frac{1}{16L}$.

To show that all the terms in the upper bound of $\rho$ are strictly positive, we require,

$$\frac{1}{16(1+\bar{\tau})} - \frac{2\bar{\tau}}{17\sqrt{3}}\left(\frac{9B^2}{L^2} + 3\right) > 0 \iff \frac{17\sqrt{3}L^2}{288B^2 + 96L^2} > \bar{\tau} + \bar{\tau}^2$$

Because $\frac{L^2}{50(3B^2+L^2)} \geq \bar{\tau}$, this is satisfied. It is easy to show that the other arguments in the definition of $\rho$, in (D.14), are strictly positive.

To get the best dependence, one would also have to use the idea of (Pethick et al., 2023) to include another term in the potential function to have a bound for $\rho$ that approaches $\frac{1}{2L}$ as $\bar{\tau} \to 0$. We do not pursue this here for brevity since the best-known upper bound for $\rho$ (which is $1/L$) is attained in Section 3 for another algorithm.

*Proof of Theorem D.4.* To upper bound the last term on the right-hand side of the assertion of Lemma D.1, we use the result of Lemma D.3. In particular, in Lemma D.3, we let $\alpha_k = \frac{2}{\sqrt{k+3}}$ and multiply both sides by $\frac{\bar{\tau}(1-\beta_k)c_3}{L^2}$ to get

$$\frac{\bar{\tau}(1-\beta_k)c_3}{L^2}\mathbb{E}\|\mathbf{g}_{k+1} - G(\mathbf{z}_{k+1})\|^2$$
$$\leq \frac{\bar{\tau}(1-\beta_k)c_3}{L^2}\left(1 - \frac{1}{\sqrt{k+3}}\right)\mathbb{E}\|\mathbf{g}_k - G(\mathbf{z}_k)\|^2 - \frac{\bar{\tau}(1-\beta_k)c_3\alpha_k}{2L^2}\mathbb{E}\|\mathbf{g}_k - G(\mathbf{z}_k)\|^2$$
$$+ 2\bar{\tau}(1-\beta_k)c_3\mathbb{E}\|\mathbf{z}_{k+1} - \mathbf{z}_k\|^2$$
$$+ \frac{2\bar{\tau}(1-\beta_k)c_3\alpha_k^2 B^2}{L^2}\mathbb{E}\|\mathbf{z}_k - \mathbf{z}_0\|^2 + \frac{2\bar{\tau}(1-\beta_k)c_3\alpha_k^2\sigma^2}{L^2}.$$

Using the identity $(1-\beta_k)\left(1 - \frac{1}{\sqrt{k+3}}\right) \leq \frac{k+1}{k+3}$ and Young's inequality on the last inequality gives us that

$$\frac{\bar{\tau}(1-\beta_k)c_3}{L^2}\mathbb{E}\|\mathbf{g}_{k+1} - G(\mathbf{z}_{k+1})\|^2$$
$$\leq \frac{\bar{\tau}c_3(k+1)}{L^2(k+3)}\mathbb{E}\|\mathbf{g}_k - G(\mathbf{z}_k)\|^2 - \frac{\bar{\tau}(1-\beta_k)c_3\alpha_k}{2L^2}\mathbb{E}\|\mathbf{g}_k - G(\mathbf{z}_k)\|^2$$
$$+ 2\bar{\tau}(1-\beta_k)c_3\mathbb{E}\|\mathbf{z}_{k+1} - \mathbf{z}_k\|^2$$
$$+ \frac{4\bar{\tau}(1-\beta_k)c_3\alpha_k^2 B^2}{L^2}\left(\mathbb{E}\|\mathbf{z}_{k+1} - \mathbf{z}_0\|^2 + \mathbb{E}\|\mathbf{z}_k - \mathbf{z}_{k+1}\|^2\right) + \frac{2\bar{\tau}(1-\beta_k)c_3\alpha_k^2\sigma^2}{L^2}. \quad \text{(D.15)}$$

Let us also note for the sixth term on the right-hand side of (D.1) that

$$\mathcal{C}_{3,k}\mathbb{E}\|\mathbf{z}_0 - \mathbf{z}_{k+1/2}\|^2 = \left(\frac{5\rho\tau_k\beta_k}{2\gamma_k} + \frac{2\tau_k(36\tau_k\gamma_k^2 B^2 - \beta_k)}{1+\tau_k}\right)\mathbb{E}\|\mathbf{z}_0 - \mathbf{z}_{k+1/2}\|^2$$
$$\leq \left(\frac{5\rho\tau_k\beta_k}{2\gamma_k} - \frac{2\tau_k\beta_k}{1+\tau_k}\right)\mathbb{E}\|\mathbf{z}_0 - \mathbf{z}_{k+1/2}\|^2$$
$$+ \frac{3 \times 72\tau_k^2\gamma_k^2 B^2}{1+\tau_k}\mathbb{E}\left(\|\mathbf{z}_0 - \mathbf{z}_{k+1}\|^2 + \|\mathbf{z}_{k+1} - \mathbf{z}_k\|^2 + \|\mathbf{z}_k - \mathbf{z}_{k+1/2}\|^2\right). \quad \text{(D.16)}$$

Adding (D.15) to the result of Lemma D.1 and using (D.16), we obtain

$$\mathbb{E}\|\mathbf{z}^\star - \mathbf{z}_{k+1}\|^2 + \frac{\bar{\tau}(1-\beta_k)c_3}{L^2}\mathbb{E}\|\mathbf{g}_{k+1} - G(\mathbf{z}_{k+1})\|^2$$
$$\leq (1-\beta_k)\mathbb{E}\|\mathbf{z}^\star - \mathbf{z}_k\|^2 + \frac{\bar{\tau}c_3(k+1)}{L^2(k+3)}\mathbb{E}\|\mathbf{g}_k - G(\mathbf{z}_k)\|^2 + \beta_k\|\mathbf{z}^\star - \mathbf{z}_0\|^2 + \mathcal{R}_k$$
$$+ \mathcal{D}_{1,k}\mathbb{E}\|\mathbf{z}_k - \mathbf{z}_{k+1/2}\|^2 + \mathcal{D}_{2,k}\mathbb{E}\|\mathbf{z}_k - \mathbf{z}_{k+1}\|^2 + \mathcal{D}_{3,k}\mathbb{E}\|\mathbf{z}_0 - \mathbf{z}_{k+1/2}\|^2$$
$$+ \mathcal{D}_{4,k}\mathbb{E}\|\mathbf{g}_k - G(\mathbf{z}_k)\|^2 + \mathcal{D}_{5,k}\mathbb{E}\|\mathbf{z}_0 - \mathbf{z}_{k+1}\|^2, \quad \text{(D.17)}$$

where the coefficients are defined as (where we assigned $c_3 = 6$ for the free variable)

$$\mathcal{R}_k = \mathcal{E}_k + \frac{12\bar{\tau}(1 - \beta_k)\alpha_k^2\sigma^2}{L^2},$$

$$\mathcal{D}_{1,k} = \mathcal{C}_{1,k} + \frac{216\tau_k^2\gamma_k^2 B^2}{1 + \tau_k},$$

$$\mathcal{D}_{2,k} = \mathcal{C}_{2,k} + 12\bar{\tau}(1 - \beta_k) + \frac{24\bar{\tau}(1 - \beta_k)\alpha_k^2 B^2}{L^2} + \frac{216\tau_k^2\gamma_k^2 B^2}{1 + \tau_k},$$

$$\mathcal{D}_{3,k} = \mathcal{C}_{3,k} - \frac{72\tau_k^2\gamma_k^2 B^2}{1 + \tau_k},$$

$$\mathcal{D}_{4,k} = \mathcal{C}_{4,k} - \frac{3\bar{\tau}(1 - \beta_k)\alpha_k}{L^2},$$

$$\mathcal{D}_{5,k} = \frac{24\bar{\tau}(1 - \beta_k)\alpha_k^2 B^2}{L^2} + \frac{216\tau_k^2\gamma_k^2 B^2}{1 + \tau_k} - \frac{\beta_k(1 - \tau_k)}{1 + \tau_k}.$$

This is, unfortunately, an even more complicated bound than Lemma D.1. One can see that the first two lines of (D.17) contain terms that will telescope after minor manipulations. The third line of (D.17) contains the term $\mathcal{R}_k$, which scales as $O(\tau_k^2 + \alpha_k^2) = O(1/k)$. What remains is to select the parameters such that the last five terms of (D.17) will be nonpositive.

We wish to upper bound the sum of the last five terms in (D.17) with

$$\Theta(-\tau_k\mathbb{E}[(1 - \beta_k)\|\mathbf{z}_k - \mathbf{z}_{k+1/2}\|^2 + (1 - \beta_k)\|\mathbf{g}_k - G(\mathbf{z}_k)\|^2 + \beta_k\|\mathbf{z}_0 - \mathbf{z}_{k+1/2}\|^2]).$$

Let us recall

$$\alpha_k = \frac{2}{\sqrt{k+3}}, \quad \gamma_k = \frac{1}{4L} \quad \tau_k = \frac{\bar{\tau}}{\sqrt{k+3}}.$$

Then the last term of (D.17) will be nonpositive, that is, $\mathcal{D}_{5,k} \leq 0$, if,

$$\frac{96\bar{\tau}(1 - \beta_k)B^2}{(k+3)L^2} + \frac{27\bar{\tau}^2 B^2}{2(k+3)(1 + \tau_k)L^2} \leq \frac{\beta_k(1 - \tau_k)}{1 + \tau_k},$$

First, upper bound the left hand side. Since $1 - \beta_k \leq 1$, $\bar{\tau}^2 \leq \bar{\tau}$ because $\bar{\tau} \leq 1$ and $\frac{1}{1+\tau_k} \leq 1$, we get

$$\frac{96\bar{\tau}(1 - \beta_k)B^2}{(k+3)L^2} + \frac{27\bar{\tau}^2 B^2}{2(k+3)(1 + \tau_k)L^2} \leq \frac{96\bar{\tau}B^2}{(k+3)L^2} + \frac{27\bar{\tau}B^2}{2(k+3)L^2} \leq \frac{219\bar{\tau}B^2}{2(k+3)L^2}.$$

Next, lower bound the right hand side,

$$\frac{\beta_k(1 - \tau_k)}{1 + \tau_k} = \frac{1}{k+3} \cdot \frac{1 - \tau_k}{1 + \tau_k} \geq \frac{1}{2(k+3)}$$

where $\beta_k = \frac{1}{k+3}$, and $\frac{1-\tau_k}{1+\tau_k} \geq \frac{1-1/3}{1+1/3} = 1/2$ since $\tau_k = \frac{\bar{\tau}}{\sqrt{k+3}} \leq \bar{\tau} \leq 1/3$. Therefore it is enough to require

$$\frac{219\bar{\tau}B^2}{2(k+3)L^2} \leq \frac{1}{2(k+3)}.$$

This will be implied by,

$$\bar{\tau} \leq \frac{L^2}{219B^2}.$$

We estimate the second from last term of (D.17). By using $\mathcal{C}_{4,k}$ from (D.2) and $\bar{\tau}\alpha_k = 2\tau_k$, we have

$$\mathcal{D}_{4,k} = 110\rho\tau_k\gamma_k + \frac{4\tau_k\gamma_k^2(9 + 18\tau_k)}{1 + \tau_k} - \frac{6\tau_k(1 - \beta_k)}{L^2}.$$

We will upper bound this term by $-\frac{\tau_k(1-\beta_k)}{32L^2}$. That is, we have

$$\mathcal{D}_{4,k} \leq -\frac{\tau_k(1-\beta_k)}{32L^2} \iff \frac{55\rho}{2L} + \frac{9}{4L^2(1+\tau_k)} + \frac{9\bar{\tau}}{2L^2(1+\tau_k)\sqrt{k+3}} \leq \frac{191(1-\beta_k)}{32L^2},$$

where for the equivalence, we divided both sides of the inequality by $\tau_k$ and plugged in $\gamma_k$.

Using $\frac{1}{1+\tau_k} \leq 1$, $1-\beta_k \geq 2/3$ since $k \geq 0$, and $\frac{1}{\sqrt{k+3}} \leq 1/\sqrt{3}$, it is enough that

$$\frac{55\rho}{2L} + \frac{9}{4L^2} + \frac{9\bar{\tau}}{2\sqrt{3}L^2} \leq \frac{191}{48L^2} \iff \frac{55\rho}{2L} \leq \frac{83}{48L^2} - \frac{9\bar{\tau}}{2\sqrt{3}L^2}.$$

Therefore, a sufficient condition is

$$\rho \leq \frac{L}{55}\left(\frac{83}{24L^2} - \frac{9\bar{\tau}}{\sqrt{3}L^2}\right). \tag{D.18}$$

We estimate the third from last term of (D.17). We wish to show that $\mathcal{D}_{3,k} \leq -\frac{\tau_k\beta_k}{3(1+\tau_k)}$. By using $\mathcal{C}_{3,k}$ from (D.2), we have

$$\mathcal{D}_{3,k} = \frac{5\rho\tau_k\beta_k}{2\gamma_k} - \frac{2\tau_k\beta_k}{1+\tau_k} \leq -\frac{\tau_k\beta_k}{3(1+\tau_k)} \iff \frac{\rho}{2\gamma_k} \leq \frac{1}{3(1+\tau_k)},$$

when $\rho \leq \frac{1}{12L}$. We used here that $1+\tau_k \leq 2$ since $\tau_k \leq \bar{\tau} \leq 1$ and $\gamma_k = \frac{1}{4L}$.

We estimate the sixth term in (D.17). We wish to show that $\mathcal{D}_{2,k} \leq 0$. Let us use the definition of $\mathcal{C}_{2,k}$ from (D.2) to write

$$\mathcal{D}_{2,k} = \frac{1}{24(1+\tau_k)} + 12\bar{\tau}(1-\beta_k) + \frac{24\bar{\tau}(1-\beta_k)\alpha_k^2 B^2}{L^2} + \frac{216\tau_k^2\gamma_k^2 B^2}{1+\tau_k} - \frac{(1-\tau_k)(1-\beta_k)}{1+\tau_k}.$$

Using $\alpha_k^2 = \frac{4}{k+3}$, $\gamma_k^2 = \frac{1}{16L^2}$ and $\tau_k^2 = \frac{\bar{\tau}^2}{k+3}$, the terms that are the second and third from the last become

$$\frac{24\bar{\tau}(1-\beta_k)\alpha_k^2 B^2}{L^2} = \frac{96\bar{\tau}(1-\beta_k)B^2}{(k+3)L^2}, \quad \text{and} \quad \frac{216\tau_k^2\gamma_k^2 B^2}{1+\tau_k} = \frac{27\bar{\tau}^2 B^2}{2(k+3)(1+\tau_k)L^2}.$$

Now the lower bound of the first and last term gives

$$\frac{(1-\tau_k)(1-\beta_k)}{1+\tau_k} - \frac{1}{24(1+\tau_k)} = \frac{(1-\tau_k)(1-\beta_k) - 1/24}{1+\tau_k} \geq \frac{59}{120(1+\tau_k)},$$

because $1-\beta_k \geq 2/3$, and $1-\tau_k \geq 4/5$ since $\tau_k \leq \frac{1}{3\sqrt{3}} < \frac{1}{5}$. So it is enough to show,

$$\mathcal{D}_{2,k} \leq 0 \impliedby 12\bar{\tau}(1-\beta_k) + \frac{96\bar{\tau}(1-\beta_k)B^2}{(k+3)L^2} + \frac{27\bar{\tau}^2 B^2}{2(k+3)(1+\tau_k)L^2} \leq \frac{59}{120(1+\tau_k)}.$$

For the right hand side, since $\tau_k \leq 1$ gives, $\frac{1}{1+\tau_k} \geq \frac{1}{2}$, so

$$\frac{59}{120(1+\tau_k)} \geq \frac{59}{240}.$$

Using $1-\beta_k \leq 1$, $\frac{1}{k+3} \leq \frac{1}{3}$, $\bar{\tau} \leq 1/3$ (so, $\bar{\tau}^2 \leq \bar{\tau}$) and $\frac{1}{1+\tau_k} \leq 1$, we get upper bound of left hand side,

$$12\bar{\tau}(1-\beta_k) + \frac{96\bar{\tau}(1-\beta_k)B^2}{(k+3)L^2} + \frac{27\bar{\tau}^2 B^2}{2(k+3)(1+\tau_k)L^2} \leq 12\bar{\tau} + \frac{32\bar{\tau}B^2}{L^2} + \frac{9\bar{\tau}B^2}{2L^2} = \bar{\tau}\left(12 + \frac{73B^2}{2L^2}\right)$$

Therefore it is sufficient that

$$\bar{\tau}\left(12 + \frac{73B^2}{2L^2}\right) \leq \frac{59}{240}.$$

This is implied by

$$\bar{\tau} \leq \frac{59}{120} \left( \frac{L^2}{73B^2 + 24L^2} \right).$$

For the cleaner statement, we use the simpler sufficient condition,

$$\bar{\tau} \leq \frac{L^2}{50(3B^2 + L^2)},$$

because,

$$150B^2 + 50L^2 \geq \frac{120}{59} \left( 73B^2 + 24L^2 \right) \quad \text{and hence} \quad \frac{L^2}{50(3B^2 + L^2)} \leq \frac{59}{120} \left( \frac{L^2}{73B^2 + 24L^2} \right).$$

We next estimate the fifth term in (D.17). Let us use the definition of $\mathcal{C}_{1,k}$ from (D.2) to write

$$\mathcal{D}_{1,k} = \frac{\tau_k(1/4 + 9/2\gamma_k^2 L^2 + 72\tau_k\gamma_k^2 L^2 - 2(1 - \beta_k))}{1 + \tau_k} + \frac{5\rho\tau_k(1 - \beta_k)}{2\gamma_k} + 11\rho\tau_k L^2\gamma_k + \frac{216\tau_k^2\gamma_k^2 B^2}{1 + \tau_k}.$$

We then have that

$$\mathcal{D}_{1,k} \leq -\frac{\tau_k(1 - \beta_k)}{128(1 + \tau_k)} \iff \frac{1}{4(1 + \tau_k)} + 10L\rho(1 - \beta_k) + \frac{11\rho L}{4} + \frac{9}{32(1 + \tau_k)}$$

$$+ \frac{\bar{\tau}}{\sqrt{k+3}(1 + \tau_k)} \left( \frac{27B^2}{2L^2} + \frac{9}{2} \right) \leq \frac{255(1 - \beta_k)}{128(1 + \tau_k)},$$

where we divided both sides by $\tau_k$ and used $\gamma_k = \frac{1}{4L}$ in the equivalence step.

Because $1 - \beta_k \geq 2/3$, the previous inequality will be implied by

$$10L\rho(1 - \beta_k) + \frac{11\rho L}{4} + \frac{\bar{\tau}}{\sqrt{k+3}(1 + \tau_k)} \left( \frac{27B^2}{2L^2} + \frac{9}{2} \right) \leq \frac{51}{64(1 + \tau_k)}.$$

Now upper bound the left hand side. By using $1 - \beta_k \leq 1$,

$$10L\rho(1 - \beta_k) + \frac{11\rho L}{4} \leq \frac{51}{4}L\rho \quad \text{and} \quad \frac{\bar{\tau}}{\sqrt{k+3}(1 + \tau_k)} \leq \frac{\bar{\tau}}{\sqrt{3}},$$

because $\sqrt{k+3} \geq \sqrt{3}$, and $\frac{1}{1+\tau_k} \leq 1$. Therefore it is enough that

$$\frac{51}{4}L\rho + \frac{\bar{\tau}}{\sqrt{3}} \left( \frac{27B^2}{2L^2} + \frac{9}{2} \right) \leq \frac{51}{64(1 + \tau_k)}.$$

This is true as long as

$$\rho \leq \frac{1}{16L(1 + \tau_k)} - \frac{2\bar{\tau}}{17\sqrt{3}L} \left( \frac{9B^2}{L^2} + 3 \right),$$

which can be easily made independent of $k$ since $\tau_k$ is nonincreasing, that is, this bound of $\rho$ is implied by

$$\rho \leq \frac{1}{16L(1 + \bar{\tau})} - \frac{2\bar{\tau}}{17\sqrt{3}L} \left( \frac{9B^2}{L^2} + 3 \right), \tag{D.19}$$

where the upper bound is guaranteed to be positive due to the definition of $\bar{\tau}$.

Summarizing the constraints derived from the nonpositivity of $\mathcal{D}_{4,k}$, $\mathcal{D}_{3,k}$, and $\mathcal{D}_{1,k}$, we require the parameter $\rho$ to satisfy the following condition,

$$\rho \leq \min \left\{ \frac{L}{55} \left( \frac{83}{24L^2} - \frac{9\bar{\tau}}{\sqrt{3}L^2} \right), \ \frac{1}{12L}, \ \frac{1}{16L(1 + \bar{\tau})} - \frac{2\bar{\tau}}{17\sqrt{3}L} \left( \frac{9B^2}{L^2} + 3 \right) \right\}.$$

where $\bar{\tau} \leq \min\left\{\frac{L^2}{219B^2}, \frac{L^2}{50(3B^2+L^2)}\right\}$.

As $\bar{\tau} \to 0$, the bound in (D.18) approaches $\frac{1}{15.9L}$, while the bound in (D.19) approaches $\frac{1}{16L}$, so this is the bound we have in our theorem statement for $\bar{\tau} \to 0$.

With these, we then estimate (D.17) as

$$\mathbb{E}\|\mathbf{z}^\star - \mathbf{z}_{k+1}\|^2 + \frac{6\bar{\tau}(1-\beta_k)}{L^2}\mathbb{E}\|\mathbf{g}_{k+1} - G(\mathbf{z}_{k+1})\|^2$$

$$\leq (1-\beta_k)\mathbb{E}\|\mathbf{z}^\star - \mathbf{z}_k\|^2 + \frac{6\bar{\tau}(k+1)}{L^2(k+3)}\mathbb{E}\|\mathbf{g}_k - G(\mathbf{z}_k)\|^2 + \beta_k\|\mathbf{z}^\star - \mathbf{z}_0\|^2$$

$$+ \frac{72\tau_k^2\gamma_k^2\sigma^2}{(1+\tau_k)} + \frac{12\bar{\tau}(1-\beta_k)\alpha_k^2\sigma^2}{L^2}$$

$$- \frac{\tau_k(1-\beta_k)}{128(1+\tau_k)}\mathbb{E}\|\mathbf{z}_k - \mathbf{z}_{k+1/2}\|^2 - \frac{\tau_k\beta_k}{3(1+\tau_k)}\mathbb{E}\|\mathbf{z}_0 - \mathbf{z}_{k+1/2}\|^2 - \frac{\tau_k(1-\beta_k)}{32L^2}\mathbb{E}\|\mathbf{g}_k - G(\mathbf{z}_k)\|^2.$$

We now multiply both sides by $k+3$ to obtain

$$(k+3)\mathbb{E}\|\mathbf{z}^\star - \mathbf{z}_{k+1}\|^2 + \frac{6\bar{\tau}(k+2)}{L^2}\mathbb{E}\|\mathbf{g}_{k+1} - G(\mathbf{z}_{k+1})\|^2$$

$$\leq (k+2)\mathbb{E}\|\mathbf{z}^\star - \mathbf{z}_k\|^2 + \frac{6\bar{\tau}(k+1)}{L^2}\mathbb{E}\|\mathbf{g}_k - G(\mathbf{z}_k)\|^2 + \|\mathbf{z}^\star - \mathbf{z}_0\|^2$$

$$+ (k+3)\left(\frac{72\tau_k^2\gamma_k^2\sigma^2}{(1+\tau_k)} + \frac{12\bar{\tau}(1-\beta_k)\alpha_k^2\sigma^2}{L^2}\right)$$

$$- \frac{\tau_k(k+3)(1-\beta_k)}{128(1+\tau_k)}\mathbb{E}\|\mathbf{z}_k - \mathbf{z}_{k+1/2}\|^2 - \frac{\tau_k(k+3)\beta_k}{3(1+\tau_k)}\mathbb{E}\|\mathbf{z}_0 - \mathbf{z}_{k+1/2}\|^2$$

$$- \frac{\tau_k(k+3)(1-\beta_k)}{32L^2}\mathbb{E}\|\mathbf{g}_k - G(\mathbf{z}_k)\|^2.$$

That is, we have for some $\delta \geq \frac{1}{128}$ that

$$\delta\tau_k(k+3)\mathbb{E}\left[\frac{1-\beta_k}{L^2}\|\mathbf{g}_k - G(\mathbf{z}_k)\|^2 + (1-\beta_k)\|\mathbf{z}_k - \mathbf{z}_{k+1/2}\|^2 + \beta_k\|\mathbf{z}_0 - \mathbf{z}_{k+1/2}\|^2\right]$$

$$\leq -(k+3)\mathbb{E}\|\mathbf{z}^\star - \mathbf{z}_{k+1}\|^2 - \frac{6\bar{\tau}(k+2)}{L^2}\mathbb{E}\|\mathbf{g}_{k+1} - G(\mathbf{z}_{k+1})\|^2$$

$$+ (k+2)\mathbb{E}\|\mathbf{z}^\star - \mathbf{z}_k\|^2 + \frac{6\bar{\tau}(k+1)}{L^2}\mathbb{E}\|\mathbf{g}_k - G(\mathbf{z}_k)\|^2 + O(1).$$

Summing up, using $\sum_{k=0}^K \tau_k(k+3) = \Omega(K^{3/2})$, since $\tau_k = \Theta(1/\sqrt{k})$, we obtain that

$$\frac{1}{\sum_{k=0}^K \tau_k(k+3)}\sum_{k=0}^{K-1}\tau_k(k+3)\mathbb{E}\left[\frac{1-\beta_k}{L^2}\|\mathbf{g}_k - G(\mathbf{z}_k)\|^2 + (1-\beta_k)\|\mathbf{z}_k - \mathbf{z}_{k+1/2}\|^2\right.$$

$$\left. + \beta_k\|\mathbf{z}_0 - \mathbf{z}_{k+1/2}\|^2\right] = O(K^{-1/2}),$$

that is, the left-hand side is smaller than $\varepsilon^2$ after $O(\varepsilon^{-4})$ iterations. The left-hand side of the last inequality can be converted to $\mathrm{res}(\mathbf{z}_{k+1/2})$ and after using (Alacaoglu et al., 2025, Lemma 6.2). With the same idea as Section 2 and (Alacaoglu et al., 2025, Theorem 4.6), we convert this to a guarantee on a randomly selected iterate. $\square$

# E. Details for Section 5 and Further Numerical Results

## E.1. Additional experiment

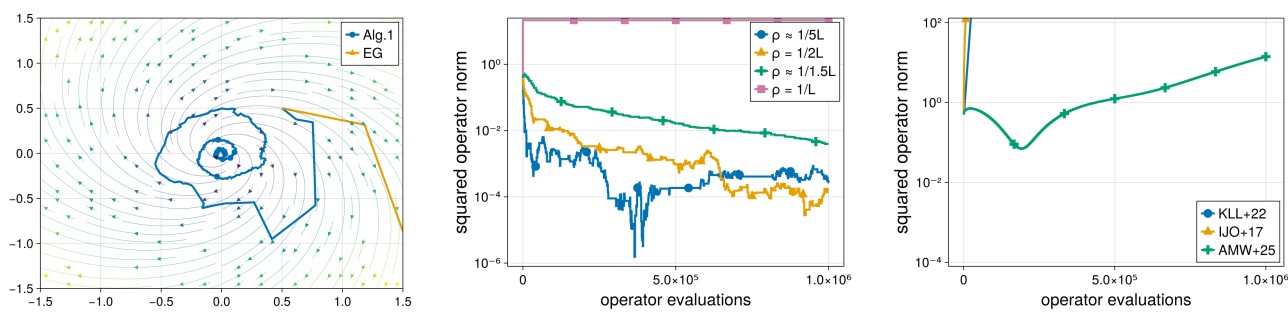

*Figure 2.* Left: Trajectories of Alg.1 and EG for the LAx counter-example (see (E.1)). Middle: Alg.1 for counter-example with varying $\rho$. Right: Methods from (Kotsalis et al., 2022), (Iusem et al., 2017), (Alacaoglu et al., 2025) in Table 1 for the LAx counter-example. *Middle and Right panel has log-scaled y-axis.*

Throughout this section, the numerical examples are unconstrained, i.e., we take $r \equiv 0$. Hence $\partial r = 0$ and the residual $\text{res}(\mathbf{z}_k) := \text{dist}(0, (G + \partial r)\mathbf{z}_k)$ reduces to $\|G(z)\|$. Therefore, we report the squared operator norm $\|G(z)\|^2$.

We use the LAx counter-example instance from (Gorbunov et al., 2023, Theorem 4.3). For $L > 0$ define

$$G(x) = LAx, \qquad x \in \mathbb{R}^2,$$

where $A$ is the rotation matrix

$$A = \begin{pmatrix} \cos\theta & -\sin\theta \\ \sin\theta & \cos\theta \end{pmatrix}, \qquad \rho = -\frac{\cos\theta}{L}. \tag{E.1}$$

(Fig. 2, *Left*) We use the same setup as the left panel of Fig. 1 fixing $\rho = \frac{1}{2L}$ to match the residual plot and keep the initialization and stopping criterion identical. In this setting, EG spirals outward and diverges as predicted by theory (Gorbunov et al., 2023), whereas Algorithm 1 converges to the solution and stabilizes.

(Fig. 2, *Middle*) On the same LAx instance, we examine stability as a function of the rotation angle $\theta$. We report the operator norm $\|F(z)\|^2$ on a log-scaled $y$-axis. Over the tested grid of $\theta$, Algorithm 1 is stable for $\rho < \frac{1}{L}$ and becomes unstable at $\rho = \frac{1}{L}$, confirming a stability boundary near $\frac{1}{L}$ for this instance, confirming our theoretical results.

(Fig. 2, *Right*) For a fixed instance with $\rho = \frac{1}{2L}$, we also run the three methods in Table 1 that require $\rho = 0$ ((Kotsalis et al., 2022), (Iusem et al., 2017), (Alacaoglu et al., 2025)) using the same initialization where the noise in the stochastic gradient have the Gaussian distribution. In this setting, all three baselines become unstable and diverge since their theory only covers $\rho = 0$.

In Fig. 3, and Fig. 1 (middle/right), we use the unconstrained quadratic problem from (Pethick et al., 2023, Example 2),

$$\min_{x \in \mathbb{R}} \max_{y \in \mathbb{R}} \varphi(x, y) := axy + \frac{b}{2}x^2 - \frac{b}{2}y^2 \tag{E.2}$$

To avoid confusion with our wMVI parameter $\rho$, we denote by $\bar{\rho}$ the parameter used in (Pethick et al., 2023, Example 2). Under their convention

$$a = \sqrt{L^2 - L^4\bar{\rho}^2}, \qquad b = L^2\bar{\rho}.$$

For this example, the corresponding wMVI parameter in wMVI is $\rho = -\bar{\rho}$. Thus, in our experiments, we set $L = 1$ and $\rho = \frac{1}{10L}$, equivalently $\bar{\rho} = -\frac{1}{10L}$.

For Fig. 3, we replace the heavy-tailed noise (from Fig. 1 (middle/right)) with zero-mean Gaussian noise and keep the same $\gamma$ grid and seven-seed, reporting the mean of the last iterate. We reuse the Algorithm 4 schedule fixed in the main text. Both methods behave similarly for moderate $\gamma$; as it decreases, the method of (Pethick et al., 2023) loses stability and diverges, whereas Algorithm 4 remains stable.

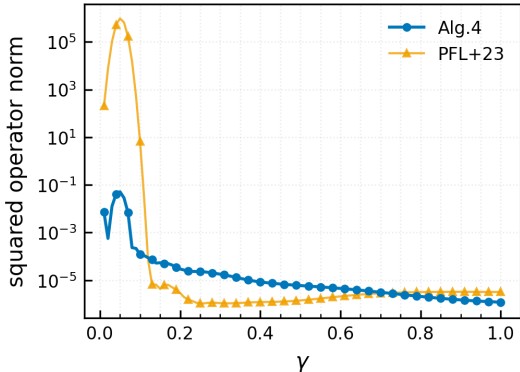

*Figure 3.* Alg. 4 and the algorithm of (Pethick et al., 2023) for the unconstrained problem in (E.2) with Gaussian noise. *Panel has log-scaled y-axis.*

### E.2. Experiment Details and Hyperparameters

For the LAx counter-example with Algorithm 1 we tune the $\alpha_k$ schedule by sweeping the multiplier $c$ in $\alpha_k = \frac{c\,\alpha}{\sqrt{k+2}\,\log(k+3)}$, increasing $c$ from 1.0 and enforcing $\alpha_0 < 1$. For the Algorithm 2 inner loop, we run a coarse-to-fine search over a single scaling coefficient that determines the per-iteration budgets $(N_k, M_k)$, targeting the smallest sample sizes that preserve the observed accuracy. Fig. 1 (left) and Fig. 2 (left) correspond to the same run.

For Fig. 2 (middle) we reuse the $N_k, M_k$ value from the example above, for Algorithm 1. We retune only the $\alpha_k$ multiplier $c$ using the same sweep and selection rule. At the boundary $\rho = \frac{1}{L}, \eta = \frac{1}{L}$, we need $\alpha = 1 - \frac{\rho}{\eta} = 0$, so we set $\alpha = 10^{-3}$ to initialize the $(N_k, M_k)$ budgets (since they depend on $\alpha$), but Algorithm 1 still diverges at this threshold. Fig. 2 (right) For each method, we use the hyperparameter setting prescribed by its theory and show the method becomes unstable since $\rho \neq 0$.

Similar to Algorithm 1, we tune the $\alpha_k$ schedule for Algorithm 4 with a coarse-to-fine grid over the initialization $\alpha_0$ and the decrease factor $c$, that is $\alpha_k = \frac{\alpha_0}{\sqrt{k/c+1}}$ in Algorithm 4. We first run a broad scan over $\alpha_0$ to identify a stable region, then perform a focused search around the best area to select the final setting. The chosen $\alpha_0$ and $c$ are frozen for the full $\gamma$ sweep and reused across noise models. With this schedule, Algorithm 4 remains stable as $\gamma$ decreases, while (Pethick et al., 2023) diverges.

### E.3. Computing infrastructure

All experiments are ran locally on a MacBook Pro (Apple M2 Pro, 10-core CPU; macOS, arm64). We used Julia 1.10.5 and Python 3.8.20. No GPU acceleration was used; all results are CPU-only. We fixed pseudorandom seeds for each run and logged hyperparameters and metrics for reproducibility. Environment files are included in the supplement.

### E.4. Code bases and modifications

*Julia (LAx / Alg. 1)* We build on an open-source Julia package from (Alacaoglu et al., 2023) available at `https://github.com/AxelBohm/beyond_golden_ratio.git`. While keeping its original structure, we implemented LAx counter-example problem and our Algorithm 1

*Python (Example 2 / Alg. 4)* We adapt our code-base from (Pethick et al., 2023) and its repository at `https://github.com/LIONS-EPFL/stochastic-weak-minty-code.git`. We use their method, labeled as PFL+23 with no change, and implemented Algorithm 4 using their primitives in the codebase.

