# OpenReview forum: "Solving Stochastic Variational Inequalities without the Bounded Variance Assumption"
_ICML.cc/2026/Conference — ICML 2026 regular_

### Official Review · Reviewer_osaY · 2026-03-06

**Soundness:** 3
**Presentation:** 3
**Significance:** 2
**Originality:** 2
**Overall Recommendation:** 4
**Confidence:** 3

**Summary:**

In this work, the authors considered the stochastic variational inequalities with a focus on constrained min-max optimisation. They assumed that a solution to the weak minty variational inequality (wMVI) exists, and they relaxed the bounded variance assumption to a weaker one, where the variance of the stochastic oracle can grow as fast as the squared norm of the optimisation variable. They considered three different methods and, assuming some technical assumptions, they proved that the generated sequence by the considered schemes will converge to an $\epsilon$-optimal residual point with an oracle complexity bound that matches the previous best results. For one of the given schemes (Algorithm 1), they achieved the mentioned result for a larger bound on the wMVI parameter, which makes it work for the previously known counterexample of the extragradient algorithm. However, the results for the other two schemes are much more conservative with respect to the bound on the wMVI constant. They have proposed a single-loop algorithm (Algorithm 4) to solve the mentioned problem. However, the results given for this algorithm rely on a more restrictive assumption.

**Compliance With Llm Reviewing Policy:**

Affirmed.

**Final Justification:**

I thank the authors for their efforts during the rebuttal process.

After considering the other reviewers’ comments and all rebuttals, I maintain my current recommendation.

**Key Questions For Authors:**

1. The authors in [3] (references are given in Strengths and Weaknesses) defined the proximal version of wMVI and addressed the constrained minimax problem through the lens of proximal wMVI. What is the connection between your formulation and the one given [3]?
2. In section 2.1, what will change if you add a prox operator for obtaining the iterates $z_{k+1}$ too?
3. In the same experiment settings, how is the comparison between the studied methods and the ones considered from the literature with respect to wall-clock time (for converging to an $\epsilon$-optimal residual point) and memory usage?

**Limitations:**

yes

**Strengths And Weaknesses:**

Strengths:
1. The draft is well-written and presents a thorough introduction section.
2. Relaxing the bound on the variance of the stochastic oracle from a constant to a term that can growas fast as the squared norm of the optimisation variable.
3. Analysing the previously known forward-backward-forward method with mini batch technique and obtain the best previous complexity bound for converging to an $\epsilon$-optimal residual point under a weaker variance bound assumption (tighter bound on choice of wMVI parameter $\rho$).
4. Analysing Algorithm 1 and obtaining the same results but for a more relaxed bound on the choice of $rho,$ which makes the algorithm work for the previously known counterexample of the Extragradient algorithm.

Weaknesses:
1. It was stated that (BV) fails for biliniear cost function. I think it only fails when the stochastic oracle has multiplicative noise and not additive noise. In the case of additive noise, like additive white noise, it still holds for the bilinear cost function.
2. Assumption 2 seems restrictive and needs to be justified better.
3. In Section 2.1, the prox operator is only used for obtaining $z_{k+\tfrac{1}{2}}$ and not for obtaining $z_{k+1}.$ It is restrictive as when $r$ is the indicator function of a constraint set, then prox will be relaxed to the projection operator. While the iterates $z_{k+\tfrac{1}{2}}$ are guaranteed to be in the constraint set, the iterates $z_{k+1}$ are not. In many applications, you can not even evaluate the cost function or any oracle outside of the constraint set.
4. In Theorem 2.1, $z^{\text{out}}$ is not defined.
5. It is not clear whether, for a choice of $\rho$ which satisfies both Theorem 2.1 and Theorem 3.1 conditions, there is any advantage in using Algorithm 1 instead of FBF with mini batch. It would be helpful if you could compare the wall-clock time for solving the same problem with both algorithms in the numerical example section.
6. The main advantage of Algorithm 4 is not clear. While it is a single loop algorithm, it does not improve the oracle complexity bound, it has the most restrictive bound on the choice of $\rho,$ and it requires the most restrictive assumption (Assumption 2).
7. The paper lacks discussions in terms of other optimality criteria (second-order optimality) discussed in minimax optimisation and multi-player game literature (local Nash [1,2]).
8. The numerical examples only include two toy problems and are not very informative. It could include more realistic examples and better comparisons between the proposed methods and the ones considered from the literature.

-------------------------------------
[1] Farzin, A. A., Pun, Y. M., Braun, P., & Shames, I. (2025). Properties of fixed points of generalised extra gradient methods applied to min-max problems. IEEE Control Systems Letters.

[2] Daskalakis, C., & Panageas, I. (2018). The limit points of (optimistic) gradient descent in min-max optimization. Advances in neural information processing systems, 31.

[3] Farzin, A. A., Pun, Y. M., Lesage-Landry, A., Diouane, Y., & Shames, I. (2025). Min-max optimisation for nonconvex-nonconcave functions using a random zeroth-order extragradient algorithm. Transactions on Machine Learning Research.

---

> ### Author Rebuttal · Authors · 2026-03-31
>
> Thank you for the constructive feedback!
>
> For brevity, at some points we refer to our rebuttal addressed to other reviewers who asked the same question. Sorry for the inconvenience!
>
> > BV for bilinear problems
>
> We refer to our last response to **Reviewer Ytr7** and our response to Q1 of **Reviewer LMTF**. Hence, we mean that BV fails when we construct stochastic oracles by sampling rows and columns of the data matrix, which to us, is the most natural. We will clarify this in our revision.
>
> > Justification of Assumption 2
>
> We refer to our response to Q1 of **Reviewer LMTF**. We argue that for finite-sum problems, this assumption is not restrictive and we also emphasize that it is only used for 1 out of 3 of our main results.
>
> > No prox for $z_{k+1}$
>
> Our algorithm outputs an iterate selected randomly from the history of $z_{k+1/2}$ and so **we do output feasible iterates**. $z_{k+1}$ being infeasible does not affect our guarantees. Indeed, it is often preferable to have one prox instead of two for applications when this step may be expensive. We refer to the literature survey in arxiv:1908.08465 (see for example their pg 5) for this point.
>
> > Definition of $z^{out}$
>
> We apologize for the suboptimal exposition. We had mentioned that $z^{out}$ is defined in the notation paragraph on page 2, as an output selected randomly from the iterates. We will now include at every theorem statement how the output is defined.
>
> > Benefit of Alg. 1 when $\rho$ is small in terms of wall-clock time
>
> In fact, when $\rho$ is small enough, we do not claim that there is any advantage in using Alg. 1. Indeed **Alg. 1 is preferable when $\rho$ is large to solve problems that are more nonmonotone, that cannot be solved with other methods**. Because of its more complicated structure, Alg. 1 may not be favorable with wall-clock time comparisons unless an engineering effort is made to optimize it. We emphasize that we introduce Alg. 1 to handle regimes of $\rho$ that cannot be handled otherwise. We will clarify this in our revision.
>
> > Advantage of Alg. 4
>
> The advantage is that it is single-loop (simpler to implement) and requires one sample at a time without any mini batching. These often lead to a better practical performance and easier implementation. We will clarify this in our revision
>
> > Second order optimality
>
> We will clarify that this is outside the scope of our paper. In general, one needs further assumptions for second order stationarity. This is an interesting direction for future work that we will include in our revision.
>
> > The numerical examples only include two toy problems and are not very informative
>
> We wish to emphasize that **our main contribution is theoretical and this is the reason why our experimental results are on simple problems**. In fact, **in the literature of our paper, many papers either use similar  problems (see Pethick et al., 2023) or include no numerical experiments at all: see for example Gorbunov et al. 2023 or Alacaoglu et al. 2025.**
>
> > Comparison to Farzin et al., TMLR 2025
>
> This work requires weak MVI in the prox-mapping defined in their Eq. (24) which is a different formulation than the literature, see for example Pethick et al., 2023 Alacaoglu et al. 2025 and others. In fact Remark 4 in this paper compare to the setup of Pethick et al., 2023 which is the same setup as ours and mention that they are **not directly comparable**. **Moreover, [3]  assumes bounded variance.** In their Assumption 1, eq (8). We will cite and discuss this reference. Thanks for pointing it out!
>
> > Adding prox for both steps
>
> In that case, we cannot handle nonmonotonicity due to nonlinearity that prox brings. As mentioned before, we output feasible iterates, so we do not believe that there is a benefit in including two prox operators.
>
> > Wall-clock time and memory usage
>
> When we compare Alg 4 and PFL+23 (the alg from Pethick et al., 2023) we observed that the trends we observe with respect to the number of iterations is the same when we compare wrt wall-clock time and memory. For other methods, it will depend on their implementation, which is an engineering issue, so it is outside of the scope of our paper. For our other empirical result, we are showing that Alg. 1 converges for a problem that all the other methods diverge. Since Alg 1 is the only convergent method in this case, we do not think wall-clock time or memory comparisons will add to our message here.
>
> We will include this explanation in our revision. If you wish, we can also share with you detailed comparison results wrt time and memory.

---

> > ### Author Rebuttal · Reviewer_osaY · 2026-04-01
> >
> > I thank the authors for their responses; many of my comments have been addressed. I believe that several of the discussed points can be clarified and incorporated into the manuscript through additional remarks and a substantive revision.
> >
> > However, my main unresolved concern remains related to Weakness 3 and Question 2. As a theoretical optimisation paper, broader applicability should be taken into account. In many practical settings, such as autonomous driving or other engineering applications, the feasibility of not only the final output but also all intermediate iterates is crucial. In such cases, evaluating the function or its gradients outside the constraint set is often not possible. As acknowledged by the authors, incorporating a second $prox$ operator into the algorithm necessitates further theoretical analysis to extend the results, which falls outside the scope of a short rebuttal.
> >
> > Therefore, I maintain my current score.

---

> > > ### Author Response · Authors · 2026-04-01
> > >
> > > Dear Reviewer osaY,
> > >
> > > Thank you for following up! We really appreciate that you are engaging with us. We are sorry to hear that our rebuttal did not completely address your concern. We would like to make two points. We start with the shorter one:
> > >
> > > 1. **Handling the setup that you describe is actually trivial for our results in Section 3**. Sorry for not making this point in our rebuttal, we had focused on the methods of Section 2 and 4. In particular, in Section 3, the only part we use FBF is Alg. 3 and the only behaviour we need from Alg. 3 in our work is for it to solve strongly monotone problems with optimal rate. We can easily replace this with extragradient method that utilizes proximal operators in both steps to handle the case when $G$ is not defined over the whole space but only over the feasible set. We would have to use the analysis of [Iusem et al., 2017] (reference in our paper) who can analyze this method under unbounded variance, for monotone (or more precisely star monotone problems) as our subsolver instead of Alg. 3.
> > >
> > > We can also alternatively use the forward-reflected-backward method of https://arxiv.org/abs/1808.04162. This method is analyzed for stochastic strongly monotone problems in Kotsalis et al., 2022 (citation in our work). So **this is definitely not a substantial revision to allow the setting you mention in Section 3**. If the reviewer wishes, we are happy to sketch the analysis of this simple extension and its incorporation into Section 3 in a future comment.
> > >
> > > 2. For Sections 2 and 4, we would like to emphasize that **all the intermediate iterates $z_{k+1/2}$ are feasible for all $k$.** For the setup that you described, **one can evaluate any objective function or gradient at any $z_{k+1/2}$ for any $k$ with no issues.**
> > >
> > > On the other hand, we understand your concern that the first step evaluates $G$ at $z_k$ and $z_k$ may be infeasible. However, we respectfully disagree that the fact that the auxiliary iterates $z_k$ are not feasible concern a "core tenet" of our work. The operator $G$ is defined over the whole space in our work and we only use the iterates $z_k$ to evaluate $G$ which is allowed. Of course, there may be applications where $G$ is not defined over the whole space, but this is not our setting.
> > >
> > > Moreover, we wish to emphasize that **existing, published work in our area has the same algorithmic structure and the same setting with $G$ being defined over the whole space**, see for example Pethick et al., 2023 or Pethick et al., 2022. The work you mention by Farzin et al utilizes the extragradient method but **they have a different assumption and hence are not directly comparable to our work or the work of Pethick et al., 2023 (this is according to Remark 4 of reference [3] in your review**. Moreover, neither the works of Farzin and coauthors nor Pethick et al., 2023 can handle unbounded variance. Indeed **allowing the unbounded variance is the core tenet of our work.**
> > >
> > > Moreover, many works in the area of our submission cannot even handle constraints, see for example Choudhury et al., 2023 or Gorbunov et al., 2023. As a result, we believe that we improve the state-of-the-art under this standard setup in the area of our paper (operator being defined over the whole space) by allowing unbounded variance. We are happy to mention the works of Farzin et al and mention it as an open question to derive a method where the operator may not be defined over the whole space and one can have unbounded variance for methods in Sections 2 and 4.
> > >
> > > In summary, we believe that we are improving the state-of-the-art in the sub-area that our work belongs to, under the exact same setup. We agree that the setting that the reviewer mentions can be important but **we are not focusing on any autonomous driving or engineering applications and we are more than happy to state this explicitly in our work.** Unfortunately we believe it may be unfair to our efforts to evaluate our work in a specific setting that our work is not designed for. We sincerely hope that the reviewer can evaluate our work for its main contribution -- removing the bounded variance assumption that is common in all the literature for constrained nonmonotone problems prior to our work.
> > >
> > > We wish to emphasize again that we can handle the specific setup the reviewer has in mind in Section 3 of our work **without a substantial revision** since Alg. 3 can be exchanged with extragradient in a plug-and-play fashion.
> > >
> > > Could you please let us know if there is anything unclear in our response or if you have any further questions?
> > >
> > > Best regards,
> > > Authors

---

### Official Review · Reviewer_Ytr7 · 2026-03-10

**Soundness:** 3
**Presentation:** 2
**Significance:** 2
**Originality:** 2
**Overall Recommendation:** 4
**Confidence:** 3

**Summary:**

In this paper, the authors aim to solve the Variational Inequality of the form: find $z*$ so that $0 \in (G+\partial r)(z*)$ where $r$ is convex and $G$ is an operator satisfying the $\rho$-weak Minty Variational Inequality (wMVI), here $\rho \geq 0$. The wMVI is a relaxation of the monotone condition. The operator is assumed to be given in an expectation form $G(z) = \mathbb{E}(\tilde{G}(z,\xi))$ -- $G$ is only accessed via stochastic approximations.

The main contribution is to relax the commonly-used variance bounded condition "$\mathbb{E}\Vert \tilde{G}(z,\xi) - G(z) \Vert^2 \leq \sigma^2$ for all $z$" to a more relaxed condition, $\mathbb{E} \Vert \tilde{G}(z,\xi) - G(z) \Vert^2 \leq B^2 \Vert z-z^*\Vert^2 + \sigma^2$ while still deriving SOTA convergence results for some existing algorithms (Forward-Backward-Forward variants). Specifically, they establish the complexity bound $\tilde{O}(\epsilon^{-4})$ for obtaining a point $z_k$ satisfying $\mathbb{E} dist(0, (G+\partial r) z_k) \leq \epsilon$ which is a first-order convergence guarantee.

**Compliance With Llm Reviewing Policy:**

Affirmed.

**Final Justification:**

The authors sufficiently addressed my concerns. I think the relaxation of the variance condition has theoretical interest.
I gave 4 (weak accept) because of the limited numerical experiments. I acknowledge that this paper is theoretical. However, since the authors argued that the variance condition is important and supported the claim with some real-world RL problems, I think some "serious" experiments would be very appreciated.

**Key Questions For Authors:**

- What is the convex function $r$ in the experiment?
- Lines 420-421, right column: the sentence "As $\gamma$ decreases, Alg. 4 remains stable..." sounds a bit counter-intuitive to me, since small $\gamma$ is expected to make algorithms stable. Can you explain more about that?
- Can you give some (real-world) examples on how to estimate $B$? How large is it?

**Limitations:**

yes

**Strengths And Weaknesses:**

# Strength
- The paper advances the current state of the art along several dimensions, including:  (1) relaxing the commonly used bounded variance assumption, (2) handling operators satisfying the weak Minty variational inequality (wMVI) condition, (3) achieving optimal complexity guarantees, and (4) allowing a known largest upper bound for the parameter $\rho$. It is worth noting that the paper does not simultaneously improve all of these aspects within a single result. Instead, the authors analyze several algorithms and show that different subsets of these directions can be improved for each scheme.
# Weaknesses:
- Although stochastic variational inequalities arise in several important machine learning applications (e.g., GAN training), the practical significance of relaxing the bounded variance assumption is not entirely clear. In particular, it would be helpful if the authors could provide concrete ML examples that satisfy the proposed relaxed condition but do not satisfy the standard bounded variance assumption.
- The presentation and results are somewhat cluttered and lack focus. This is understandable given that the paper analyzes several algorithmic schemes.
- Since the paper is primarily theoretical and analysis-focused, the experimental section is relatively weak.

---

> ### Author Rebuttal · Authors · 2026-03-31
>
> Thank you for the feedback!
>
> > ML examples not satisfying bounded variance but satisfying our assumption
>
> Neu and Okolo, 2024 explicitly studies stochastic saddle-point optimization in a regime where the gradient noise can scale linearly with the size of variables with an application to reinforcement learning. They have the same assumption as us, showing that there are **RL applications** where bounded variance is not satisfied but our assumption is satisfied.
>
> Another example is **distributionally robust learning**. For instance, we can consider the setup of arXiv:2403.10763 who studies penalized DRO objectives of the form $\min_{w}\max_{q} \sum_{i=1}^nq_{i} \ell_{i}(w)-\nu D(q||1/n) +\frac{\mu}{2}\|w\|^2$, for some distance function $D$ and evaluates on regression and classification tasks. In particular, if we take $\ell_{i}(w)=\frac{1}{2}(y_{i}-x_{i}^\top w)^2$, then $\nabla \ell_{i}(w) = (x_{i}x_{i}^\top)w-y_{i}x_{i}$, which is affine in $w$. To see why bounded variance can fail, let us focus on the gradient wrt $w$. We write $A_{i}=x_{i}x_{i}^\top$ and $b_{i}=y_{i}x_{i}$, the oracle error for primal gradient is $(A_{i}-\bar{A})w-(b_{i}- \bar{b})$, where $\bar{A}=\frac{1}{n}\sum_{j}A_{j}$ and $\bar{b}=\frac{1}{n}\sum_{j}b_{j}$. Therefore, in general, the oracle error grows linearly in $\|w\|$, so the bounded variance assumption fails on an unbounded domain. At the same time, because the feature vectors are generally bounded, the oracle satisfies exactly the assumption in our paper. In other words, this is another concrete ML example where bounded variance fails, but our relaxed assumption sill holds.
>
> > *cluttered and lacks focus*
>
> We would be happy to incorporate any suggestion that the reviewer has. In particular, could you give us particular pointers what was cluttered and lacked focus?
>
> > *experimental section is relatively weak*
>
> Indeed, empirical section is only to **supplement our theoretical results since our main focus is our theoretical contributions**. Let us emphasize that in the papers in the area of our submission, **the empirical setups are similar to ours**, see for example Pethick et al., 2023. Other relevant results don’t even have experiments, see for example Gorbunov  et al., 2023 or Alacaoglu et al., 2025.
>
> > *What is $r$ in experiments*
>
> In all current experiments, we take $r \equiv 0$. Thus, these experiments are a special case without an additional convex regularization. This is because some of the competitor algorithms only apply to unconstrained problems. **These experiments show that our results are not only more general because they can handle regularizers,  but they may be even more preferable in the special case of unconstrained problems.**
>
> > Smaller $\gamma$ being more difficult
>
> We did not mean that decreasing $\gamma$ is generically destabilizing. Rather, in weak Minty settings, $\gamma$ is coupled to the weak Minty parameter $\rho$ in the analysis, so smaller $\gamma$ is not automatically more favorable. This is visible both in the comparator and in our analysis. In Pethick et al. (2023), Theorem 7.1 states $\gamma \in (\lfloor -2\rho \rfloor_{+},1/L_{F})$, thus even in their analysis, $\gamma$ cannot be chosen albitraily small independently of the weak Minty parameter. We note that their paper uses a different sign convention for the weak Minty parameter than ours but the message is the same. In our analysis, the same type of tradeoff appears directly in the coefficients. For example, in Appendix D, line 1274, the coefficient $D_{3, k} = \frac{2E_{1}\rho \tau_{k}\beta_{k}}{\gamma_{k}}-\frac{2\tau_{k}\beta_{k}}{1+\tau_{k}}$ contains a term proportional to $\rho/\gamma_{k}$, so decreasing $\gamma_{k}$ may make the required negativity condition impossible to satisfy. In our analysis, we fixed $\gamma$ to make sure this doesn't happen but for an arbitrary $\gamma$, it can cause issues. Empirically, on this benchmark, Algorithm 4 remained stable over a broader tested range of $\gamma$ values than the method of Pethick et al. 2023.
>
> > Estimating $B$
>
> We refer to Neu&Okolo, 2024 where they show how to calculate B for their case (see Sec 2 of their work).
>
> Consider the linear operator $G(x)=Ax$, where $A\in \mathbb{R}^{m\times n}$, and let $a_{i}^\top$ denote the $i$ th row of $A$. If we sample $i \sim Unif [1,\dots,m ]$ a natural unbiased stochastic oracle is $\tilde{G}(x, i) = m(a_{i}^\top x)e_{i}$. Indeed, $\mathbb{E}[\tilde{G}(x, i)]=\frac{1}{m}\sum_{i=1}^m m(a_{i}^\top x)e_{i}= Ax$. Moreover, $\mathbb{E}\| \tilde{G}(x,i)-G(x)\|^2=\mathbb{E}\|\tilde{G}(x,i)\|^2-\|{G}(x)\|^2=(m-1)\|Ax\|^2 \leq (m-1)\|A\|^2\|x\|^2$. Hence Assumption 3 holds with $B^2=(m-1)\|A\|^2$ and $\sigma=0$. On the other hand, if $A\not= 0$, choose $\nu$ such that $A\nu\not=0$, and set $x=t\nu$. Then $\mathbb{E}\| \tilde{G}(x,i)-G(x)\|^2=(m-1)\|Ax\|^2 \rightarrow \infty$ as $t\rightarrow\infty$. Therefore on an unbounded domain, bounded variance fails, while Assumption 3 holds.

---

> > ### Author Rebuttal · Reviewer_Ytr7 · 2026-04-02
> >
> > Thanks for addressing my concerns. I increased my score to 4.

---

> > > ### Author Response · Authors · 2026-04-04
> > >
> > > Dear Reviewer Ytr7,
> > >
> > > Many thanks for your acknowledgement! We are happy to hear that our response addressed your questions and that you had decided to increase your score. Please let us know if you have any further questions. We are happy to clarify.
> > >
> > > Thanks,
> > > Authors

---

### Official Review · Reviewer_nJi1 · 2026-03-11

**Soundness:** 2
**Presentation:** 2
**Significance:** 2
**Originality:** 2
**Overall Recommendation:** 2
**Confidence:** 4

**Summary:**

The authors study stochastic variational inequalities under the assumption that they have access to a stochastic operator with a variance bounded not only by an uniform constant $\sigma$, but also by a term depending on the $z$ point: $B \| z - z^0\|$. In this formulation, the convergence of several classical methods for solving variational inequalities under different conditions on the monotonicity of the target operator is proved.

**Compliance With Llm Reviewing Policy:**

Affirmed.

**Final Justification:**

For me, the contribution of a theoretical paper may be as follows:

1) A new algorithmic idea. Not about this paper

2) Record-breaking and interesting guarantees of convergence. Here I answer that yes, the results do provide new bounds.

But I still do not agree with the presentation of the results. And I don't agree that the authors only need to change "a few lines" to fix it.

- Variational inequalities are a rather complex class of problems and it is impossible to consider a completely non-monotone case. And a monotone case and a weakly monotonous case (non-monotone case) are actually often very close to a strongly monotone case. In particular, adding a regularizer to a monotone operator makes it strongly monotone. This changes the task, but not much. Therefore, I do not believe that the results in a strongly monotone case are not strong competitors for the current paper and they can be put in one line for comparison.

- I continue to discuss, for example, [1] the authors forget to mention an important point. In [1], the assumption about noise is stronger than the authors suggest. There is no constant $B$. In [1], it is assumed that noise is bounded only at the points of optimality.
But if we look at Table 1 of the paper under review, column $L, B$, then for [1] it will turn into $L$, and this is better than in the paper under review. This means that [1] can be considered a strong competitor. Moreover, I am not sure that it is possible to say that the authors were able to generalize all existing results to a monotone and weakly monotone case (since there is no $B$ in [1]). Moreover, there is a monotone case in [1] (yes, with poor results). But I believe that an honest comparison will not fit in one line, as the authors want.

3) Proof technique that is interesting in itself and can be reused. Сonclude that no, I have asked the authors several times to describe it. I even gave an example of how this can be done using the example of the results from [1]. But the authors avoided answering this question. That's why I still think the proof is more technical than groundbreaking.

This is a nice paper with chances of publication. But I need another round of review for acceptance.

**Key Questions For Authors:**

Asked before

**Limitations:**

Yes

**Strengths And Weaknesses:**

__Weaknesses:__

1) The authors do not mention the following papers:

[1] Mishchenko, K., Kovalev, D., Shulgin, E., Richtarik, P., and Malitsky, Y. (2020). Revisiting stochastic extragradient.

[2] Hsieh, Y.-G., Iutzeler, F., Malick, J., and Mertikopoulos, P. (2020). Explore aggressively, update conservatively: Stochastic extragradient methods with variable stepsize scaling

[3] Gorbunov et al. Stochastic Extragradient: General Analysis and Improved Rates

[1] is very important, there is also no assumption of bounded noise. The authors of [1] bound the noise only in $z*$. The authors of the paper under review actually bounded the noise at any point, but with an additional factor $B \| z - z^0\|^2$. Therefore, in this regard, the results of [1] can be regarded as more general in terms of the noise assumption.

Papers [2, 3] also consider the assumption close to that of the paper under review. I recommend look at the results of [3]

A detailed review of these papers is necessary for an honest understanding of the field. I recommend that the authors fully compare the settings: assumptions about monotony, the presence of a regularizer, and so on.

The presence of these papers [1-3] greatly reduces the contribution of the paper under review.

2) The limitations on $\rho$ ($1/16L$) are stronger than, for example, obtained in [4]. Can the authors explain this fact? Are these the rudiments of theoretical analysis?

[4] Gorbunov, E., Taylor, A., Horvath, S., and Gidel, G. Convergence of proximal point and extragradient-based methods beyond monotonicity: the case of negative comonotonicity

3) The algorithms are not new (the authors do not claim that this is their contribution)

4) The theoretical analysis is technical. One need to add $B\| z - z^0\|$ and carefully run it through the analysis, and then choose the right step to kill the influence of this member.

__Strengths:__

1) The formulation with a regularizer and a weak monotone operator and unbounded stochstic noise has not been considered in the literature. It turns out that the authors add a bit of generality.

---

> ### Author Rebuttal · Authors · 2026-03-31
>
> Thanks for pointing out these references! To address your concern, we explain that, **these papers do not reduce our contribution. In fact, these papers do not change our contribution at all because none of them can apply to solve a monotone VI (let alone a nonmonotone VI that we focus on).**
>
> [1] Mishchenko et al., (2020):
>
> This paper can only handle the assumption you mention under **strong convexity** for their Thm 2. Without strong convexity, their result is Thm 3. In this case, the authors actually have on the RHS $\sup_{x \in X} \sigma_x$ where $\sigma_x$ is the variance at $x$, so they need the variance to be bounded at all points in the domain. In our case, the domain is unbounded, so **they assume a uniformly bounded variance** without strong convexity.
>
> Even with bounded variance, Thm 3 does not show guarantees on a valid optimality measure. That is, the authors have on the LHS $\mathbb{E}[g(\hat{x}^t) - g(x) + \langle F(x), \hat x^t- x \rangle]$ which is not an optimality measure, see, e.g., Example 1 of https://link.springer.com/article/10.1007/s10107-025-02247-8, The correct optimality measure is $\mathbb{E}[\sup_{x \in X} g(\hat{x}^t) - g(x) + \langle F(x), \hat x^t- x \rangle]$ which is not implied by the result of [1].
>
> [2] Hsieh et al., 2020
>
> This work also **cannot solve monotone** VIs with rates (let us emphasize that we can even solve **nonmonotone** stochastic VIs with best-known complexity). Please see Sec 5.2, Assumption 5 in this paper where the authors assume an error bound condition which can be seen as a strong monotonicity-type assumption.
>
> For their result for affine operators, in Thm. 5, the authors actually take $\kappa=0$ in (1b), please see page 21 of this paper. Even though they claim they can handle nonzero $\kappa$, this is unclear without error bound condition.
>
> Moreover, **this paper only applies to unconstrained problems**, see Eq (Opt) in this paper whereas **we solve constrained and regularized problems**. **Let us emphasize that in the area of VIs, inclusion of regularizers is very important. Many important VI problems come from nonlinear programming or game theory where one needs to have constraints or regularizers.**
>
> [3] Gorbunov et al., (2022)
>
> This paper also **cannot solve monotone VIs**. All the results require Assumption 1.2 of the paper which is quasi-strong monotonicity which also assumes that the solution is unique. These don't hold even for linear operators. Our result does not require any form of strong monotonicity.
>
> Moreover, this work **can only solve unconstrained problems, see Eq (VIP) in this paper. The same limitation mentioned for [2] apply.**
>
> **We can solve monotone or nonmonotone problems (with weak MVI), with constraints or regularizers.**
>
> In fact, it was explained in Alacaoglu et al., 2025 that handling the unbounded variance-type assumption we have is straightforward with strong-convexity type assumptions. The main difficulty arises when we need to show results for only monotone problems. We go beyond this to show results for even nonmonotone problems.
>
> We can include the above clarification in our revision.
>
> **Now that we clarified that these references do not undermine our contributions, we hope that the reviewer can re-evaluate our work.** Can you please let us know if any further confusions remain?
>
> > $\rho$ bound of $1/(16L)$
>
> Let us emphasize that **only our result in Thm 4.1** has this looser bound on $\rho$ which is because we did not tighten the Young’s inequalities. This is because our result in Thm 3.1 already allows $\rho < 1/L$ which is the best-known range for this parameter, whereas [4] requires $\rho<1/(2L)$. That is, we already have a better dependence on $\rho$ than [4].
>
> > *The algorithms are not new*
>
> Indeed, our contributions are **improved and tight analyses for existing algorithms**. We believe that this is a common type of contribution to the literature and **we do not think that not having new algorithms is a weakness**.
>
> > technicality of the analysis
>
> We also respectfully disagree that technicality of a theoretical analysis is a weakness. Indeed, all the papers in the area of our work have technical analyses, one can see Pethick et al., 2023 or the references that the reviewer mentioned above. We are handling a difficult problem and we also believe that our main text describes all the ideas in a high level to guide the reader on the technicalities. We believe it is an interesting future direction to simplify these analyses.
>
> > *a bit of generality*
>
> As mentioned above, our results are expanding on the existing literature on nontrivial ways. We already made these comparisons in Table 1. Therefore, we respectfully disagree that our results are only adding “a bit of generality”. In addition to improving the state-of-the-art, we contribute technical tools for analyses of three important types of algorithms for solving VIs. We hope that the reviewer takes these into account. We will be happy to clarify further.

---

> > ### Author Rebuttal · Reviewer_nJi1 · 2026-04-04
> >
> > Thanks to the authors for the reply!
> >
> > > This paper can only handle the assumption you mention
> >
> > Explanations of this kind should appear in the paper. But unfortunately, just describing them in a rebuttal is not enough. Even without rebbutal, I understood the differences between [1-3] and the paper under review, but as I wrote in the review, the presence of [1-3] greatly changes the contribution of the paper. The fact that the authors do not make a detailed comparison in the paper is critical for me. I'll explain
> >
> > 1) The title "Solving Stochastic Variational Inequalities without the Bounded Variance Assumption"
> >
> > 2) The highlighted part of the contribution "Our main contribution is the analysis of these methods without the bounded variance or bounded domain assumptions. "
> >
> > But that's not true for me, and it's an oversale. Correctly, not "we are the first", but we "generalize the existing results to the minty case." These are two completely different positions. Moreover, I'm not even sure what the authors really generalized. As I said, [1] uses a weaker assumption compared to the paper under review, so it cannot be assumed that the paper generalizes the existing results.
> >
> > It requires a lot of paper modification (even a title for me). An honest comparison with existing securities will take 0.5-1 new additional page. I cannot accept an paper until I see a new version, and this is prohibited by the rules of the conference. Moreover, I have read other reviews, it seems to me that the reviewers believed that "we are the first" and do not know that the real contribution is "we partly generalize the existing"
> >
> > > We also respectfully disagree that technicality of a theoretical analysis is a weakness.
> >
> > It's not enough for me either. We need a clear explanation of all the technical difficulties. Due to the fact that the paper is theoretical, I would recommend including this in the main part of the paper (one more big modification of the paper). For example, [1] https://arxiv.org/pdf/1905.11373
> >
> > 1) Trick with the same $\xi$ in Algorithms 1 and 2 is new
> >
> > 2) Lemma 2 is a key. This is a new way to prove the convergence of stochastic EG.
> >
> > 3) Lemma 2 gives oppotunity to prove Theorem 2 and find a new result
> >
> > This is a short version, need more accurate and detailed. This is what I recommend to add
> >
> > > we did not tighten the Young’s inequalities
> >
> > I understand and I fact this fact can be highlighted. Is this the problem of proof or the setting? Need the discussion in the paper.
> >
> > The paper should be heavily redone, I do not recommend acceptance. I'm leaving the reject.

---

> > > ### Author Response · Authors · 2026-04-04
> > >
> > > Dear Reviewer nJi1,
> > >
> > > Thank you for your acknowledgment and for further engaging with us.
> > >
> > > **We believe that there are some serious misunderstandings about our contributions. We continue to argue that the works you brought up do not undermine our contributions and do not require a substantial revision.**
> > >
> > > > "Explanations of this kind should appear in the paper."
> > >
> > > **We already cited papers that are more relevant to ours than the papers you suggested in [1-3].** In particular, the work of (Choudhury et al., 2023) can handle unbounded variance for *unconstrained* weak MVIs without bounded variance with guarantees on the **operator norm**, however they cannot handle constraints which are extremely important for VIs and their algorithms require information about the solution for parameters. Others may handle constraints but not nonmonotonicity. See Table 1 for a summary.
> > >
> > > > the presence of [1-3] greatly changes the contribution of the paper
> > >
> > > Let us emphasize again **none of the works you suggested, including [1] can solve monotone VIs**. We provided precise pointers explaining why. These works require either **strongly monotone VIs** or assumptions such as **quasi-strong monotonicity** or **error-bound** conditions to handle unbounded variance, **none of these assumptions hold even for a bilinear problem**.
> > >
> > > >  As I said, [1] uses a weaker assumption compared to the paper under review, so it cannot be assumed that the paper generalizes the existing results.
> > >
> > > **We are indeed generalizing previous result because [1] needs to use strong convexity and we do not need strong convexity**. Moreover, as we explained [1] does not have a valid result without strong convexity.
> > >
> > > > "it seems to me that the reviewers believed that "we are the first" and do not know that the real contribution is "we partly generalize the existing"
> > >
> > > We believe that the other reviewers understood our  contributions very well because we already compared with the most relevant works to ours. As we said **none of the works you brought up can solve nonmonotone VIs or even monotone VIs**, **none of the works you brought up can find guarantees for the residual norm for constrained VIs**. As a result we continue to argue with concrete evidence in this message that **none of the works you brought up change our contribution because we focus on monotone or weak MVIs**. Could you please let us know if any of these statements is not correct in your opinion?
> > >
> > > >  it's an oversale
> > >
> > > **We respectfully disagree that our work has any overselling**. As mentioned above, we already cited all the works most relevant to ours and discussed the differences. Please see Table 1, none of these works require strong convexity or similar assumptions.
> > >
> > > We believe you refer to the statement in the second column of page 3. Let us clarify **this is taken out of the context**. The sentence before this says "we first state the algorithms which are either existing in the literature or are simple modifications over the existing ones" and then we say our contribution is the analysis of these methods without the bounded variance or bounded domain. **So we refer to the methods in the previous sentence that you left out while quoting**. As such, we are behind our statement because none of the works you brought up analyze these methods. We can add here that we focus on **non-strongly monotone case** to incorporate your suggestion, but this is **adding 4 words, a minor modification.**
> > >
> > > > The title
> > >
> > > We are happy to change the title to add the specification at the end "for monotone and weakly Minty VIs"
> > >
> > > > Is this the problem of proof or the setting? Need the discussion in the paper.
> > >
> > > It is because the paper you mention is for *unconstrained* setting and it is easier to tighten these in that case. As we mentioned, we didn't tighten this part in order not to make our paper even longer. The best-known bound for $\rho$ is $1/L$ and we already attain this in Section 3 (the work you brought up cannot handle this bound). We can add this in the paper (another small revision).
> > >
> > > > The paper should be heavily redone, I do not recommend acceptance. I'm leaving the reject.
> > >
> > > **We respectfully disagree**. As we highlighted in our rebuttal and this message, none of the works you brought up can solve **nonmonotone or monotone VIs** and we focus on **nonmonotone and monotone VIs**.
> > >
> > > **Incorporating the works you mention is 1 paragraph, a minor modification**
> > >
> > > where we just would say *they need strong convexity, strong monotonicity or slightly weakened versions of them and cannot even solve bilinear problems*.
> > >
> > > > one more big modification
> > >
> > > We believe this is already partly done in the paper. For novelties in Section 2, please see lines 236-247, first column. Section 3, end of page 6, beginning of page 7. Section 4, lines 334-344 and 351-357, second column. We are happy to expand, which we believe is another **minor modification.**
> > >
> > > Can you please let us know if further confusions remain?
> > >
> > > Thanks,
> > > Authors

---

### Official Review · Reviewer_LMTF · 2026-03-13

**Soundness:** 3
**Presentation:** 3
**Significance:** 3
**Originality:** 3
**Overall Recommendation:** 5
**Confidence:** 3

**Summary:**

Overall, the paper examine a critical issue in stochastic variational inequalities by removing the commonly assumed bounded variance and bounded domain conditions, focusing on both monotone problems and structured nonmonotone problems satisfying a weak Minty variational inequality condition. The work intend to address the challenge of achieving optimal complexity guarantees under a more realistic variance growth assumption that allows the oracle variance to scale with the squared distance to a solution, which covers important cases such as bilinear min-max problems with unbounded domains. They analyze three algorithmic frameworks: mini-batched forward-backward-forward method, an inexact fixed-point method with multilevel Monte Carlo, and a variance-reduced single-loop scheme with Halpern anchoring, and show that each attains the best-known $\tilde O(\varepsilon^{-4})$ stochastic oracle complexity for driving the expected residual below $\varepsilon$, while tolerating varying degrees of nonmonotonicity and offering complementary theoretical and practical trade-offs.

**Compliance With Llm Reviewing Policy:**

Affirmed.

**Final Justification:**

All of my comments have been addressed. Hence, I have increased the score to 5.

**Key Questions For Authors:**

I have the following two questions:

1. I am a bit confused regarding Assumption 2. In the analysis of variational inequalities, one typically uses assumptions on the variance with assumptions on the Lipschitz continuity of the expected mapping. Assumption 2 appears to be somewhat strong. Is there a guarantee that $L_{\text{exp}}$ is always finite for all realizations of $\xi$, and for any $x$ and $y$, i.e., that it exists almost surely? Can this be derived from the main Lipschitz continuity assumption? I would also like to understand what challenges arise if one uses the assumption $||G(x) - G(y)|| \leq L||x-y||$ instead of Assumption 2 when analyzing the method.

2. In the expression of Lemma 3.4, I see the quantity $J_{\eta(G+\partial r)}(z_k)$. Will $z_k$ be $z_T$?

I will increase the score if my concerns are addressed.

**Limitations:**

There are no limitations or potential negative societal impact of this work.

**Strengths And Weaknesses:**

**Strengths:**

The paper is theoretically strong and technically sound.

**Weaknesses:**

The writing and language in some parts of the paper can be tightened.

Overall, the paper looks good.

---

> ### Author Rebuttal · Authors · 2026-03-31
>
> Thank you for your constructive comments!
>
> # Response to Q1:
>
> We first emphasize that **Assumption 2 is only used for analyzing the algorithm in Section  4 and not in Section 2 and Section 3**. As a result, two of our main results do not require this assumption and only use Lipschitzness of $G$.
>
> Moreover, **in the context of Section 4 (single loop stochastic VI algorithms for residual guarantee), this assumption is already used in the state-of-the-art results**, see Assumption III in Pethick et al., 2023 or Assumption 4.5 in Alacaoglu et al., 2025. The former work used a uniformly bounded variance assumption to analyze their algorithms along with the expected Lipschitzness, whereas we could avoid the bounded variance assumption.
>
> For example, **if we take a finite sum problem with $G(z)= \frac{1}{N} \sum_{i=1}^N F_i(z)$ where each $F_i$ is Lipschitz, then this assumption will be satisfied for any $x, y$ and independent of the distribution $\xi$ is drawed from – we can use uniform or nonuniform sampling from $N$ operators.** Indeed, Lipschitzness of $F_i$ is a stronger assumption than only Lipschitzness of $G$. However, for many of the applications, it still holds. Let us explain in detail for a canonical min-max problem (A similar estimation also holds for a VI with an affine operator):
>
> Consider $\min_{x\in X}\max_{y\in Y} y^\top Ax$, where the operator is $G(x,y)=(A^\top y, -Ax)$. Let $a_i^\top$ and $b_j$ denote the rows and columns of A, respectively. We let $\xi = (i, j)$ with sample $i \in \{1,\dots,m\}$ and $j \in \{1, \dots,n\}$ selected uniformly at random, and then we define the stochastic operator $\tilde{G}((x, y), \xi) =(n(b_j^\top y)e_j, -m(a_i^\top x)e_i)$ where $e_i$ is the $i$-th canonical basis vector. Then $\mathbb{E} [\tilde{G}((x, y), \xi)] = G(x, y)$. Moreover, for $z=(x, y)$ and $z’=(x’, y’)$, $\|\tilde{G} (z, \xi) -\tilde{G} (z', \xi)\|^2 = n^2|b_j^\top (y-y’)|^2 + m^2|a_i^\top (x-x’)|^2 \leq (n^2\|b_j\|^2 + m^2 \|a_i\|^2)\|z-z'\|^2$. Taking expectation with respect to the uniform sampling gives, $\mathbb{E} \|\tilde{G} (z, \xi) -\tilde{G} (z', \xi)\|^2 \leq \left( n \sum_{j=1}^n \|b_{j}\|^2 +m\sum_{i=1}^m \|a_{i}\|^2\right)\|z-z'\|^2=(m+n)\|A\|_{F}^2\|z-z'\|^2$. Hence, Assumption 2 holds with constant $(m+n)(\|A\|_F)^2$.
>
> **Let us also note that this assumption (or a stronger version of it) is common in the prior work in our area, see for example**
>
> Assumption 3.1 in https://proceedings.mlr.press/v151/gorbunov22b/gorbunov22b.pdf
>
> Assumption 2 in https://arxiv.org/pdf/2310.02987
>
> Assumption III in https://arxiv.org/pdf/2302.09029
>
> Equation (18) in https://arxiv.org/pdf/1703.00260
>
> Lastly, only Lipschitzness of $G$ is not sufficient because in Section 4 we have to apply **variance reduction idea** by using STORM estimator of Cutkosky and Orabona (https://arxiv.org/abs/1905.10018). For this estimator to be applied, we need the expected Lipschitzness (this is common with variance reduction in general). Note that in the minimization case, they used expected Lipschitzness of the gradient of the function $\nabla f(x)$. Other examples using STORM estimator in the VI case include Pethick et al., 2023 and Alacaoglu et al., 2025 who both needed Assumption 2.
>
> # Response to Q2:
>
> This quantity is actually correctly written. $k$ is the iteration counter in the outer loop in Alg 1 and $t$ is the *inner loop*, that is, Alg. 3. Particularly, we run Alg 3 to estimate $J_{\eta(G+\partial r)}(z_k)$: that is, the quantity $J_{\eta(G+\partial r)}(z_k)$ is the solution for the problem that Alg 3 is trying to solve (See Eq. (3.1) and (3.2)). On the other hand, Alg. 3 uses the index $t=1, \dots,  T$ because the output of Alg 3 ($x_T$) is close in expectation to the solution of the subproblem that Alg 3 is solving ($J_{\eta(G+\partial r)}(z_k)$). Since the subproblem in Eq.(3.1) and (3.2) is strongly convex, we have a guarantee that the last iterate of Alg 3 ($x_T$) is close in expectation to the solution of the subproblem ($J_{\eta(G+\partial r)}(z_k)$), where the solution of the subproblem is independent of $t$.
>
> > I will increase the score if my concerns are addressed.
>
> We hope that our responses fully address your concerns. Please let us know if you have any further questions.

---

> > ### Author Rebuttal · Reviewer_LMTF · 2026-04-03
> >
> > Thanks to the authors for addressing all of my comments. I have increased the score to 5.

---

> > > ### Author Response · Authors · 2026-04-04
> > >
> > > Dear Reviewer LMTF,
> > >
> > > Many thanks for your acknowledgement! We are happy to hear that our response addressed your questions and that you had decided to increase your score. Please let us know if you have any further questions. We are happy to clarify.
> > >
> > > Thanks,
> > > Authors

---

### Decision · Program_Chairs · 2026-04-30

**Decision:**

Accept (regular)

**Comment:**

This paper analyzes algorithms for solving stochastic variational inequalities under monotone and weak Minty settings without the bounded variance assumption, achieving the best-known oracle complexity $\tilde O(\epsilon^{-4})$ for constrained VIs.

No reviewer disputes correctness. Three reviewers (LMTF: 5, Ytr7: 4, osaY: 4) are positive on the contribution. The lowest-scoring reviewer (nJi1: 2) acknowledges that "the results do provide new bounds" and describes the paper as "a nice paper with chances of publication," but argues that prior works [1] Mishchenko et al. (2020), [2] Hsieh et al. (2020), and [3] Gorbunov et al. (2022) diminish the contribution and that the presentation requires substantial revision.

The authors argue (and Reviewer osaY independently confirms in the internal discussion) that the cited prior works all require some form of strong monotonicity/convexity and cannot solve plain monotone or weak Minty VIs. Reviewer osaY further notes that regularization-based reductions come at nontrivial cost and do not naturally extend to the weak Minty setting. nJi1 did not elaborate further on this point in the discussion.

Nonetheless, Reviewers nJi1, osaY and Ytr7 all agree that the paper would benefit from a more explicit discussion of [1]-[3]. Reviewers nJi1 and osaY further note that the scope of the novelty claims could be delineated more clearly. The intermediate-iterate feasibility limitation (raised by osaY) should also be stated explicitly, and the experimental section is acknowledged as limited by both osaY and Ytr7. These concerns appear primarily about presentation and framing and do not affect the technical contributions.

Overall, the paper is technically sound and makes a meaningful contribution by extending stochastic VI complexity results to the monotone and weak Minty constrained setting without bounded variance. The lowest score appears driven by positioning and presentation concerns, instead of significant technical objections. I recommend weak accept.